# MARBLE: interpretable representations of neural population dynamics using geometric deep learning

Adam Gosztolai [1,7] ✉, Robert L. Peach[2,3,7], Alexis Arnaudon[4], Mauricio Barahona [5] & Pierre Vandergheynst[6]

The dynamics of neuron populations commonly evolve on low-dimensional manifolds. Thus, we need methods that learn the dynamical processes over neural manifolds to infer interpretable and consistent latent representations. We introduce a representation learning method, MARBLE, which decomposes on-manifold dynamics into local flow fields and maps them into a common latent space using unsupervised geometric deep learning. In simulated nonlinear dynamical systems, recurrent neural networks and experimental single-neuron recordings from primates and rodents, we discover emergent low-dimensional latent representations that parametrize high-dimensional neural dynamics during gain modulation, decision-making and changes in the internal state. These representations are consistent across neural networks and animals, enabling the robust comparison of cognitive computations. Extensive benchmarking demonstrates state-of-the-art within- and across-animal decoding accuracy of MARBLE compared to current representation learning approaches, with minimal user input. Our results suggest that a manifold structure provides a powerful inductive bias to develop decoding algorithms and assimilate data across experiments.

It is increasingly recognized that the dynamics of neural populations underpin computations in the brain and in artificial neural networks[1–3] and that these dynamics often take place on low-dimensional smooth subspaces, called neural manifolds[4–12]. From this perspective, several works have focused on how the geometry[4,7,13,14] or topology[6,8,9] of neural manifolds relates to the underlying task or computation. By contrast, others have suggested that dynamical flows of neural population activity play an equally prominent role[11,15–17] and that the geometry of the manifold is merely the result of embedding of a latent dynamical activity into neural space that changes over time or across individuals[4,18]. Although recent experimental techniques

provide a means to simultaneously record the activity of large neuron populations[19–21], inferring the underpinning latent dynamical processes from data and interpreting their relevance in computational tasks remains a fundamental challenge[16].

Overcoming this challenge requires machine-learning frameworks that leverage the manifold structure of neural states and represent the dynamical flows over these manifolds. There is a plethora of methods for inferring the manifold structure, including linear methods such as principal-component analysis (PCA), targeted dimensionality reduction (TDR)[22] or nonlinear manifold learning methods such as $t$-distributed stochastic neighbor embedding ($t$-SNE)[23] or Uniform

[1]Institute of Artificial Intelligence, Medical University of Vienna, Vienna, Austria. [2]Department of Neurology, University Hospital Würzburg, Würzburg, Germany. [3]Department of Brain Sciences, Imperial College London, London, UK. [4]Blue Brain Project, EPFL, Campus Biotech, Geneva, Switzerland. [5]Department of Mathematics, Imperial College London, London, UK. [6]Signal Processing Laboratory (LTS2), EPFL, Lausanne, Switzerland. [7]These authors contributed equally: Adam Gosztolai and Robert L. Peach. ✉e-mail: adam.gosztolai@meduniwien.ac.at

Manifold Approximation and Projection (UMAP)[24]. Yet, these methods do not explicitly represent time information, only implicitly to the extent discernible in the density variation in the data. While consistent neural dynamics have been demonstrated using canonical correlation analysis (CCA), which aligns neural trajectories approximated as linear subspaces across sessions and animals[4,18], this is only meaningful when the trial-averaged dynamics closely approximate the single-trial dynamics. Similarly to manifold learning, topological data analysis infers invariant structures, for example, loops[6] and tori[9], from neural states without explicitly learning dynamics. Correspondingly, they can capture qualitative behaviors and changes (for example, bifurcations) but not quantitative changes in dynamics and geometry that can be crucial during representational drift[4] or gain modulation[25].

To learn time information explicitly in single trials, dynamical systems methods have been used[26–31]; however, time information in neural states depends on the particular embedding in neural state space (measured neurons), which typically varies across sessions and animals. The Latent Factor Analysis for Dynamical Systems (LFADS) framework partially overcomes this by aligning latent dynamical processes by linear transformations[30]; however, alignment is not meaningful in general as animals can employ distinct neural 'strategies' to solve a task[32]. Recently, representation learning methods such as Physics Informed Variational Auto-Encoder (pi-VAE)[33] and Consistent EmBeddings of high-dimensional Recordings using Auxiliary variables (CEBRA)[34] have been introduced to infer interpretable latent representations and accurate decoding of neural activity into behavior. While CEBRA can be used with time information only, finding consistent representations across animals requires supervision via behavioral data, which, like LFADS, aligns latent representations but uses nonlinear transformations. For scientific discovery, it would be desirable to circumvent using behavioral information, which can introduce unintended correspondence between experimental conditions, trials or animals, thus hindering the development of an unbiased distance metric to compare neural computations.

Here, we introduce a representation learning method called MARBLE (MAnifold Representation Basis LEarning), which obtains interpretable and decodable latent representations from neural dynamics and provides a well-defined similarity metric between neural population dynamics across conditions and even across different systems. MARBLE takes as input neural firing rates and user-defined labels of experimental conditions under which trials are dynamically consistent, permitting local feature extraction. Then, combining ideas from empirical dynamical modeling[35], differential geometry and the statistical theory of collective systems[36,37], it decomposes the dynamics into local flow fields and maps them into a common latent space using unsupervised geometric deep learning[38–40]. The user-defined labels are not class assignments, rather MARBLE infers similarities between local flow fields across multiple conditions, allowing a global latent space structure relating conditions to emerge. We show that MARBLE representations of single-neuron population recordings of the premotor cortex of macaques during a reaching task and of the hippocampus of rats during a spatial navigation task are substantially more interpretable and decodable than those obtained using current representation learning frameworks[30,34]. Further, MARBLE provides a robust data-driven similarity metric between dynamical systems from a limited number of trials, expressive enough to infer subtle changes in the high-dimensional dynamical flows of recurrent neural networks (RNNs) trained on cognitive tasks, which are not detected by linear subspace alignment[4,13,18], and to relate these changes to task variables such as gain modulation and decision thresholds. Finally, we show that MARBLE can discover consistent latent representations across networks and animals without auxiliary signals, offering a well-defined similarity metric.

Our results suggest that differential geometric notions can reveal unaccounted-for nonlinear variations in neural data that can further our understanding of neural dynamics underpinning neural computations and behavior.

## Unsupervised representation of vector fields over manifolds

To characterize neural computations during a task, for example, decision-making or arm-reaching, a typical experiment involves a set of trials under a stimulus or task condition. These trials produce a set of $d$-dimensional time series $\{\mathbf{x}(t; c)\}$, representing the activity of $d$ neurons (or dimensionally reduced variables) under condition $c$, which we consider to be continuous, such as firing rates. Frequently, one performs recordings under diverse conditions to discover the global latent structure of neural states or the latent variables parametrizing all tasks. For such discoveries, one requires a metric to compare dynamical flow fields across conditions or animals and reveal alterations in neural mechanisms. This is challenging as neural states often trace out complex, but sparsely sampled nonlinear flow fields. Further, across participants and sessions, neural states may be embedded differently due to different neurons recorded[4,30].

To address these challenges, MARBLE takes as input an ensemble of trials $\{\mathbf{x}(t; c)\}$ per condition $c$ and represents the local dynamical flow fields over the underlying unknown manifolds (Fig. 1a showing one manifold) in a shared latent space to reveal dynamical relationships across conditions. To exploit the manifold structure, we assume that $\{\mathbf{x}(t; c)\}$ for fixed $c$ are dynamically consistent, that is, governed by the same but possibly time-varying inputs. This allows describing the dynamics as a vector field $\mathbf{F}_c = (\mathbf{f}_1(c), \ldots, \mathbf{f}_n(c))$ anchored to a point cloud $\mathbf{X}_c = (\mathbf{x}_1(c), \ldots, \mathbf{x}_n(c))$, where $n$ is the number of sampled neural states (Fig. 1b). We approximate the unknown manifold by a proximity graph to $\mathbf{X}_c$ (Fig. 1b) and use it to define a tangent space around each neural state and a notion of smoothness (parallel transport) between nearby vectors (Supplementary Fig. 1 and equation (2)). This construction allows defining a learnable vector diffusion process (equation (3)) to denoise the flow field while preserving its fixed point structure (Fig. 1c). The manifold structure also permits decomposing the vector field into local flow fields (LFFs) defined for each neural state $i$ as the vector field at most a distance $p$ from $i$ over the graph (Fig. 1d), where $p$ can also be thought of as the order of the function that locally approximates the vector field. This lifts $d$-dimensional neural states to a $O(d^{p+1})$-dimensional space to encode the local dynamical context of the neural state, providing information about the short-term dynamical effect of perturbations. Note that time information is also encoded as consecutive neural states are typically adjacent over the manifold. As we will show, this richer information substantially enhances the representational capability of our method.

As LFFs encode local dynamical variation, they are typically shared broadly across dynamical systems. Thus, they do not assign labels to neural states as in supervised learning. Instead, we use an unsupervised geometric deep-learning architecture to map LFFs individually to $E$-dimensional latent vectors (Fig. 1e), which introduces parameter sharing and permits identifying overlapping LFFs across conditions and systems. The architecture consists of three components (Fig. 1f, see Methods for details): (1) $p$ gradient filter layers that give the best $p$-th order approximation of the LFF around $i$ (Supplementary Figs. 1–3 and equation (8)); (2) inner product features with learnable linear transformations that make the latent vectors invariant of different embeddings of neural states manifesting as local rotations in LFFs (Extended Data Figs. 1 and 2 and equation (10)); and (3) a multilayer perceptron that outputs the latent vector $\mathbf{z}_i$ (equation (11)). The architecture has several hyperparameters relating to training and feature extraction (Supplementary Tables 1 and 2). While most were kept at default values throughout and led to convergent training, some were varied, as summarized in Supplementary Table 3, to tune the behavior of the model. Their effect is detailed in the examples below. The network is trained unsupervised, which is possible because the continuity of LFFs over

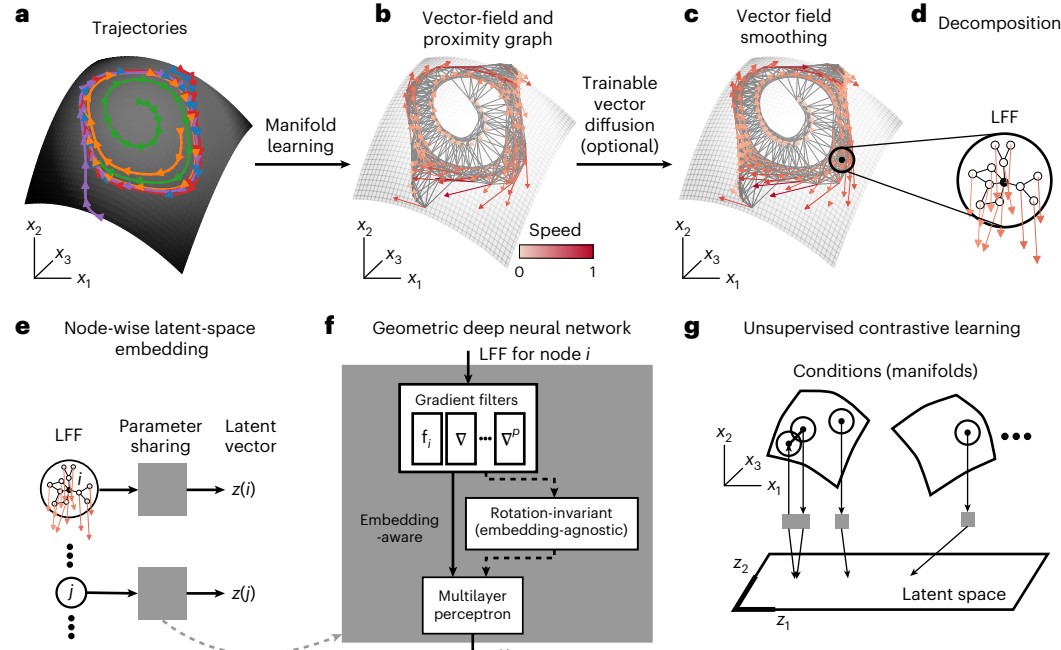

**Fig. 1 | The MARBLE method: unsupervised representation of dynamics over manifolds. a**, Neural activity in different trials (colors) evolving over a latent manifold. **b**, Vector field representation of dynamics. The nearest-neighbor graph between neural states approximates the unknown manifold. **c**, The vector field is optionally denoised by a trainable vector diffusion, which aligns (via parallel transport) nearby vectors while preserving the fixed point structure. **d**, The dynamics are decomposed into LFFs. **e**, LFFs are mapped one by one into latent space by a geometric deep neural network. The model infers dynamical overlaps across datasets based on similar LFFs. **f**, The model has three steps: feature extraction from LFFs using $p$-th order gradient filters; (optional) transformation into rotation-invariant features for embedding-agnostic representations (otherwise, representations are embedding-aware); and mapping features into latent space using a multilayer perceptron. **g**, Using the continuity over the manifold, the network is trained using unsupervised contrastive learning, mapping neighboring LFFs close and non-neighbors (both within and across manifolds) far in latent space.

the manifold (adjacent LFFs being typically more similar (except at a fixed point) than nonadjacent ones) provides a contrastive learning objective (Fig. 1g and equation (12)).

The set of latent vectors $\mathbf{Z}_c = (\mathbf{z}_1(c), ..., \mathbf{z}_n(c))$ represents the flow field under condition $c$ as an empirical distribution $P_c$. Mapping multiple flow fields $c$ and $c'$ simultaneously, which can represent different conditions within a system or different systems altogether, allows defining a distance post hoc $d(P_c, P_{c'})$ between their latent representations $P_c, P_{c'}$, reflecting the dynamical overlap between them. We use the optimal transport distance (equation (13)) because it leverages information of the metric structure in latent space and generally outperforms entropic measures (for example, Kullback–Leibler (KL) divergence) when detecting complex interactions based on overlapping distributions[37].

## Embedding-aware and embedding-agnostic representations

The inner product features (Fig. 1f) allow two operation modes. As an example, consider linear and rotational flow fields over a two-dimensional (2D) plane ($d = 2$, trivial manifold) in Fig. 2a. As shown later, MARBLE can also capture complex nonlinear dynamics and manifolds. We have labeled these flow fields as different conditions (treating them as different manifolds) and used MARBLE to discover a set of latent vectors that generate them.

In embedding-aware mode, the inner product features are disabled (Fig. 2b, left) to learn the orientation of the LFFs, ensuring maximal expressivity and interpretability. Consequently, constant fields are mapped into two distinct clusters, whereas rotational fields are distributed over a ring manifold based on the angular orientation of LFFs (Fig. 2b, left, insets and Extended Data Figs. 1 and 2). This mode is useful when representing dynamics across conditions but within a

given animal or neural network with the same neurons being sampled (Figs. 2c–e and 3a–g) or when a global geometry spanning all conditions is sought after (Fig. 4e).

In embedding-agnostic mode, the inner product features are enabled, making the learned features invariant to rotational transformations of the LFFs. As a result, the latent representation of the vector field will be invariant to different embeddings (conformal maps)[41], which introduce local rotations. Thus, this mode is useful when comparing systems, such as neural networks trained from different initializations (Fig. 3h–j). Our example shows that constant vector fields are no longer distinguishable based on LFF orientation (Fig. 2d, right, insets and Extended Data Figs. 1 and 2); however, we still capture expansion and contraction in LFFs over a one-dimensional manifold (Extended Data Fig. 3). In both embedding-aware and -agnostic examples, note that LFFs from different manifolds (defined by user labels) are mapped close or far away, depending on their dynamical information, corroborating that labels are used for feature extraction and not for supervision.

To demonstrate embedding-agnostic mode for nonlinear dynamics on a nonlinear manifold, consider the Van der Pol oscillator mapped to a paraboloid while varying the damping parameter $\mu$ and the manifold curvature (Fig. 2c and Methods). Using short, randomly initialized simulated trajectories for 20 values of $\mu$, labeled as different conditions, we used embedding-agnostic MARBLE to embed the corresponding vector fields into a shared latent space ($E = 5$). Despite the sparse sampling, we detected robust dynamical variation across conditions as $\mu$ was varied. Specifically, the similarity matrix between conditions $D_{cc'} = d(P_c, P'_c)$ displays a two-partition structure, indicating two dynamical regimes (Fig. 2d). These correspond to the stable and unstable limit cycles separated by the Hopf bifurcation at $\mu = 0$. This result is observed independently of manifold curvature (Extended Data Fig. 4). Furthermore, and despite the sparse sampling across

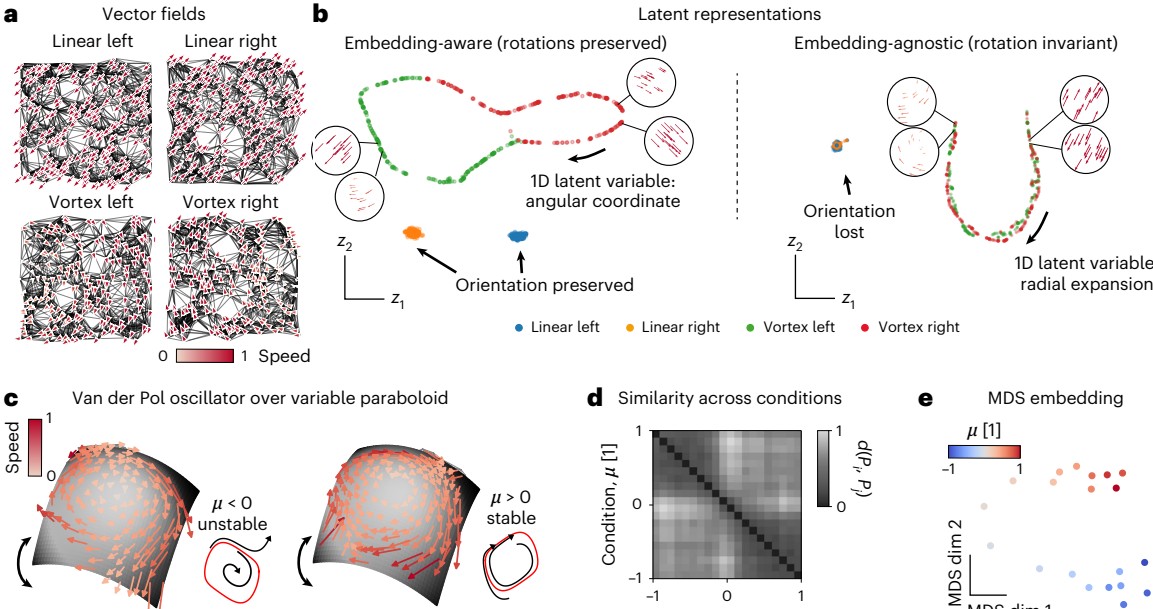

**Fig. 2 | Illustrative examples of joint MARBLE latent representations of dynamics across conditions and manifolds. a**, Four toy vector fields sampled uniformly at random over a flat (trivial) manifold approximated by a graph (black lines). Two constant (top) and two rotational fields (bottom). **b**, Embedding-aware representations distinguish rotational information in LFFs (left). The ring manifold in latent space parametrizes the angular variation (insets). Embedding-agnostic latent representations learn only vector field expansion and contraction (right). The one-dimensional (1D) manifold in latent space parametrizes the radial variation of LFFs (insets). **c**, Vector fields of the Van der Pol oscillator over a variable-curvature paraboloid in the unstable ($\mu = -0.25$) and stable ($\mu = 0.25$) regimes sampled from randomly initialized trajectories. Insets show the limit cycle in red and representative trajectories from a vertical projected view. **d**, Similarity across conditions based on optimal transport distance between respective MARBLE representations. Clustering indicates an abrupt dynamical change at $\mu = 0$. **e**, 2D MDS embedding of the distribution distance matrix recovers the ordering of parameter $\mu$ over two weakly connected one-dimensional manifolds.

conditions, the two-dimensional embedding of $D_{cc'}$ using multidimensional scaling revealed a one-dimensional manifold capturing the continuous variation of $\mu$ (Fig. 2e). Notably, manifold variation was not captured when training an embedding-aware network (Extended Data Fig. 4), confirming that embedding-agnostic representations are invariant to manifold embedding with only marginal loss of expressivity compared to the embedding-aware mode.

## Comparing dynamics across recurrent neural networks

There has been a surge of recent interest in RNNs as surrogate models for neural computations[15,42–44]. Previous approaches for comparing RNN computations for a given task relied on aligning linear subspace representations of neural states[14,42,45], which requires a point-by-point alignment of trials across conditions. While this is valid when the trial-averaged trajectory well approximates the single-trial dynamics[4,18], it does not hold when flow fields are governed by complex fixed point structures. Thus, systematically comparing computations across RNNs requires an accurate representation of the nonlinear flow fields.

To showcase MARBLE on a complex nonlinear dynamical system, we simulated the delayed-match-to-sample task[46] using RNNs with a rank-two connectivity matrix (Fig. 3a and Methods), which were previously shown to be sufficiently expressive to learn this task[47]. This common contextual decision-making task comprises two distinct stimuli with variable gain and two stimulus epochs of variable duration interspersed by a delay (Fig. 3b). At unit gain, we trained the RNNs to converge to output 1 if a stimulus was present during both epochs and −1 otherwise (Fig. 3b). As expected[48], the neural dynamics of trained networks evolve on a randomly oriented plane (Fig. 3c). Yet we found that differently initialized networks produce two classes of solutions: in solution I the neurons specialize in sensing the two stimuli,

characterized by the clustering of their input weights $\mathbf{w}_1^{in}$, $\mathbf{w}_2^{in}$ (Fig. 3d), whereas in solution II the neurons generalize across the two stimuli (Extended Data Fig. 5d). These two solutions exhibit qualitatively different fixed point landscapes (three fixed points for zero gain and one limit cycle for large enough gain), which cannot be aligned via continuous (linear or nonlinear) transformations (Fig. 3e and Extended Data Figs. 5c,e and 6).

We first asked whether MARBLE could infer dynamical neural correlates of loss of task performance as the stimulus gain is decreased beyond the decision threshold. For a given gain, we simulated 200 trials of different durations, sampling different portions of the nonlinear flow field due to randomness (Extended Data Fig. 6). We formed dynamically consistent datasets by subdividing trials at the stimulus onset and end to obtain four epochs. We then formed two groups, one from epochs where the stimulus was on and another where the stimulus was off (Fig. 3b). Repeating for different gains, we obtained 20 groups of stimulated epochs at different gains and 20 additional groups from unstimulated (no gain) epochs. We used the latter as negative controls because the flow fields vary across them due to sampling variability and not gain modulation. We then trained embedding-aware MARBLE to map all 40 groups, labeled as distinct conditions, into a common latent space. The resulting representational similarity matrix between conditions exhibits a block-diagonal structure (Fig. 3f). The top left block denotes distances between unstimulated controls and contains vanishing entries, demonstrating the robustness of MARBLE to sampling variability. The two bottom-right sub-blocks identified by hierarchical clustering indicate a quantitative change in dynamics, which notably corresponds to a sudden drop in task performance (Fig. 2g) from 1 to 0.5 (random). Thus, MARBLE enables the detection of dynamical events that are interpretable in terms of global decision variables.

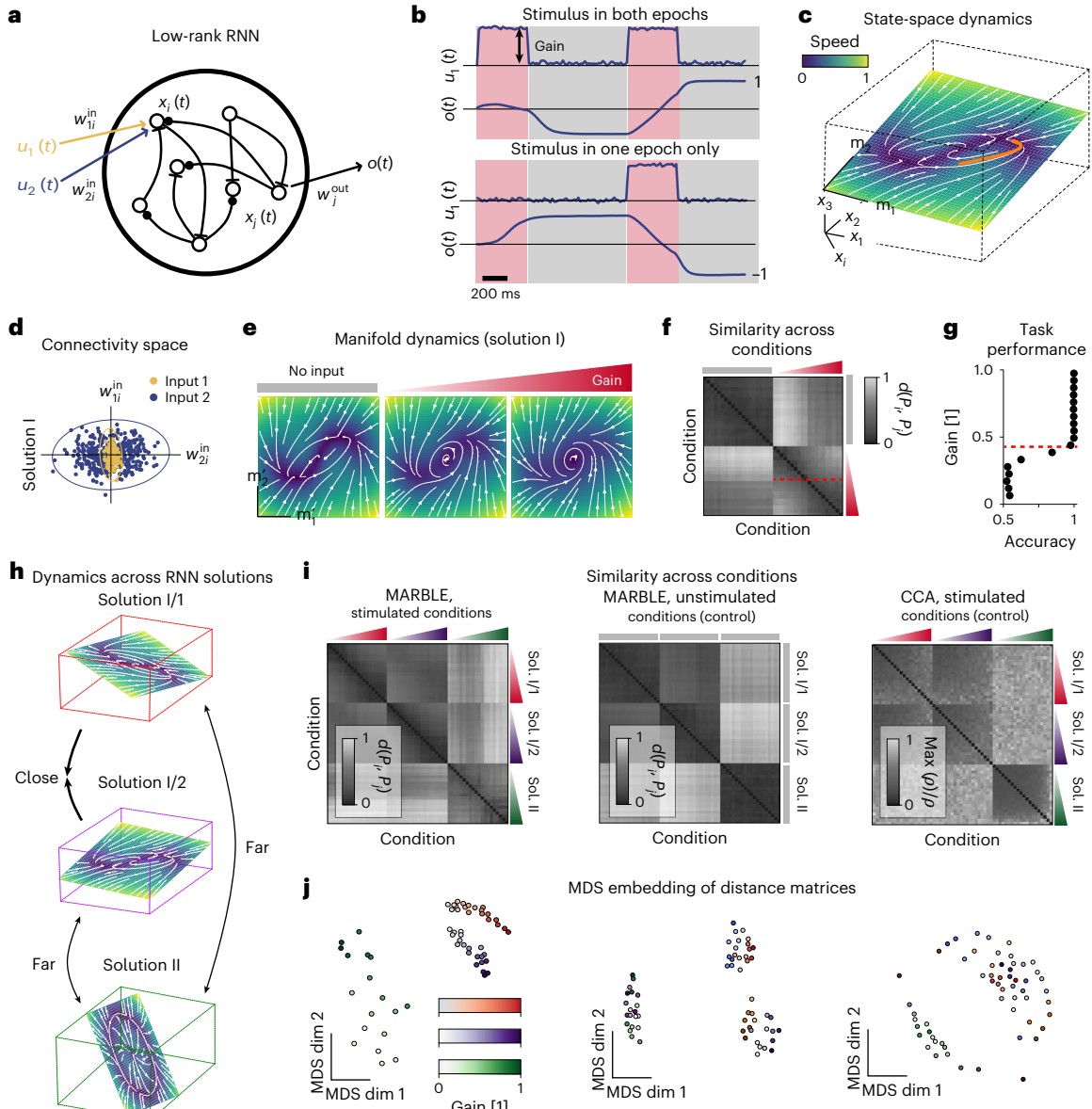

**Fig. 3 | Comparing dynamical processes across recurrent neural networks.**
**a**, Low-rank RNN takes two stimuli as input and produces a decision variable as a read-out. **b**, Two representative stimulus patterns (only one stimulus is shown) and decision outcomes for the delayed-match-to-sample task. Input amplitude is controlled by the 'gain' during stimulus epochs (red) and is zero otherwise (gray). **c**, Neural dynamics of a trained rank-two RNN evolves on a randomly oriented plane. Phase portrait of field dynamics superimposed with a trajectory during a trial (orange). **d**, Space of input weights in a trained RNN. Colors indicate $k$-means clustering with two subpopulations specialized in one or the other input. Ellipses represent $3 \times$ s.d. of fitted Gaussian distributions. **e**, Mean-field dynamics for gains 0, 0.32 and 1.0. **f**, Similarity across gain conditions (shading represents gains from 0 to 1) based on optimal transport distance between the respective embedding-aware MARBLE representations. Hierarchical clustering indicates

two clusters in the nonzero gain sub-block, indicating a qualitative change point (red dashed line). **g**, The predicted change point corresponds to the bifurcation, causing the loss in task performance. **h**, Networks trained from different initializations can have different dynamics over differently embedded manifolds (solution I, II). Networks sampled from the same weight distribution produce the same fixed point structure over differently embedded manifolds (solution I/1, I/2). **i**, Similarity across networks and gains (left) or no gain (control, middle) based on embedding-agnostic representations. We compare this to CCA (right). Shading represents gains from 0 to 1. **j**, MDS embedding of the distance matrices shows the continuous variation of latent states during gain modulation (left), no variation across no-gain conditions (middle) and clustering across network solutions. For CCA the same plot shows clustering across network solutions but no variation due to gain modulation.

Next, we define a similarity metric between the dynamical flows across distinct RNNs. Consider network solutions I and II, which have different flow fields (Fig. 3h) and as a negative control, two new networks whose weights are randomly sampled from the Gaussian distribution of solution I whose flow fields provably preserve the fixed point structure[47]. Due to the arbitrary embedding of neural states across networks, we used embedding-agnostic MARBLE to represent data from these three RNNs at different gains ($E = 5$). We found that

latent representations were insensitive to manifold orientation, detecting similar flow fields across control networks (solutions I/1 and I/2) and different ones across solutions I and II (Fig. 3i, left). Further, the multidimensional scaling (MDS) embedding of the similarity matrix shows one-dimensional line manifolds (Fig. 3j, left) parametrizing ordered, continuous variation across gain-modulated conditions. As expected, for unstimulated conditions, we could still discriminate different network solutions but no longer found coherent dynamical

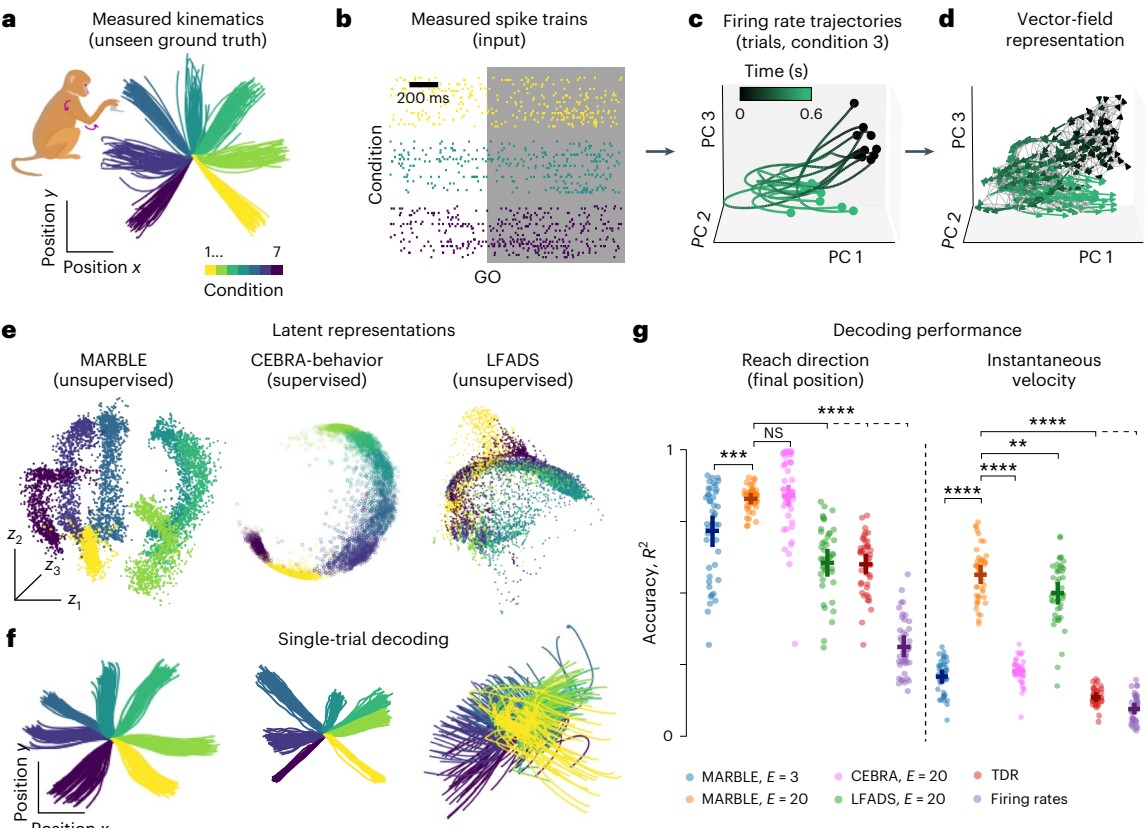

**Fig. 4 | Interpretable representation and decoding of neural activity during arm-reaching. a**, Ground truth hand trajectories of a macaque in seven reach conditions. Monkey image adapted by Andrea Colins Rodriguez from https://www.scidraw.io/drawing/445, CCBY 4.0. **b**, Single-trial spike trains in the premotor cortex for three reach conditions (24 recording channels within each color). The shaded area shows the analyzed traces after the GO cue. **c**, Firing rate trajectories for a reach condition (up) PCA-embedded in three dimensions for visualization. **d**, Vector field obtained from firing rate trajectories. **e**, Latent representations of neural data across conditions in a single session. CEBRA-behavior was used with

reach conditions as labels. The MARBLE representation reveals as an emergent property the latent global geometric arrangement (circular and temporal order) spanning all reaches reflecting physical space. **f**, Linear decoding of hand trajectories from latent representations. **g**, Decoding accuracy measured by $R^2$ between ground truth and decoded trajectories across all sessions for the final position (left) and instantaneous velocity (right). Two-sided Wilcoxon tests (paired samples), $**P < 1 \times 10^{-2}$; $***P < 1 \times 10^{-3}$; $****P < 1 \times 10^{-4}$; NS, not significant. Horizontal and vertical bars show mean and $1 \times$ s.d., respectively ($n = 43$).

changes (Fig. 3i,j, middle). For benchmarking, we used CCA, which quantifies the extent to which the linear subspace representations of data in different conditions can be linearly aligned. While CCA could distinguish solutions I and II having different fixed point structures, it could not detect dynamical variation due to gain modulation (Fig. 3i,j, right). This suggests that finding coherent latent dynamics across animals using CCA[4,18] has likely succeeded in instances where linear subspace alignment is equivalent to dynamical alignment, for example, when trial-averaged dynamics well-approximate the single-trial dynamics. Here we find that MARBLE provides a robust metric between more general nonlinear flow fields possibly generated by different system architectures.

## Representing and decoding neural dynamics during arm-reaches

State-of-the-art representation learning of neural dynamics uses a joint embedding of neural and behavioral signals[34]; however, it would be advantageous to base biological discovery on the post hoc interpretation of neural representations, which do not introduce correlations between neural states and latent representations based on behavioral signals. To demonstrate this, we reanalyzed electrophysiological recordings of a macaque performing a delayed center-out hand-reaching task[30] (Methods). During the task, a trained monkey moved a handle toward seven distinct targets at radial locations from the start position. This dataset comprises simultaneous recordings of

hand kinematics (Fig. 4a) and neural activity via a 24-channel probe from the premotor cortex over 44 recording sessions (Fig. 4b shows one session).

Previous supervised approaches revealed a global geometric structure of latent states spanning different reach conditions[2,49]. We asked whether this structure could emerge from local dynamical features in MARBLE representations. As before, we constructed dynamically consistent manifolds from the firing rates for each reach condition (Fig. 4c,d), which allows for extracting LFFs. Yet, the latent representation across conditions remains emergent because the LFFs are local and shared across conditions. Using these labels, we trained an embedding-aware MARBLE network and benchmarked it against two other prominent approaches: CEBRA[34], which we used as a supervised model using reach condition as labels (CEBRA-behavior) and LFADS[30], an unsupervised method that uses generative recurrent neural networks.

We found that MARBLE representations ($E = 3$) could simultaneously discover the latent states parametrizing the temporal sequences of positions within reaches and the global circular configuration across reaches (Fig. 4e and Extended Data Fig. 7a). The latter is corroborated by the diagonal and periodic structure of the condition-averaged similarity matrix between latent representations across reach conditions (Extended Data Fig. 7b) and a circular manifold in the associated MDS embedding (Extended Data Fig. 7c). By comparison, although CEBRA-behavior could unfold the global arrangement of the reaches,

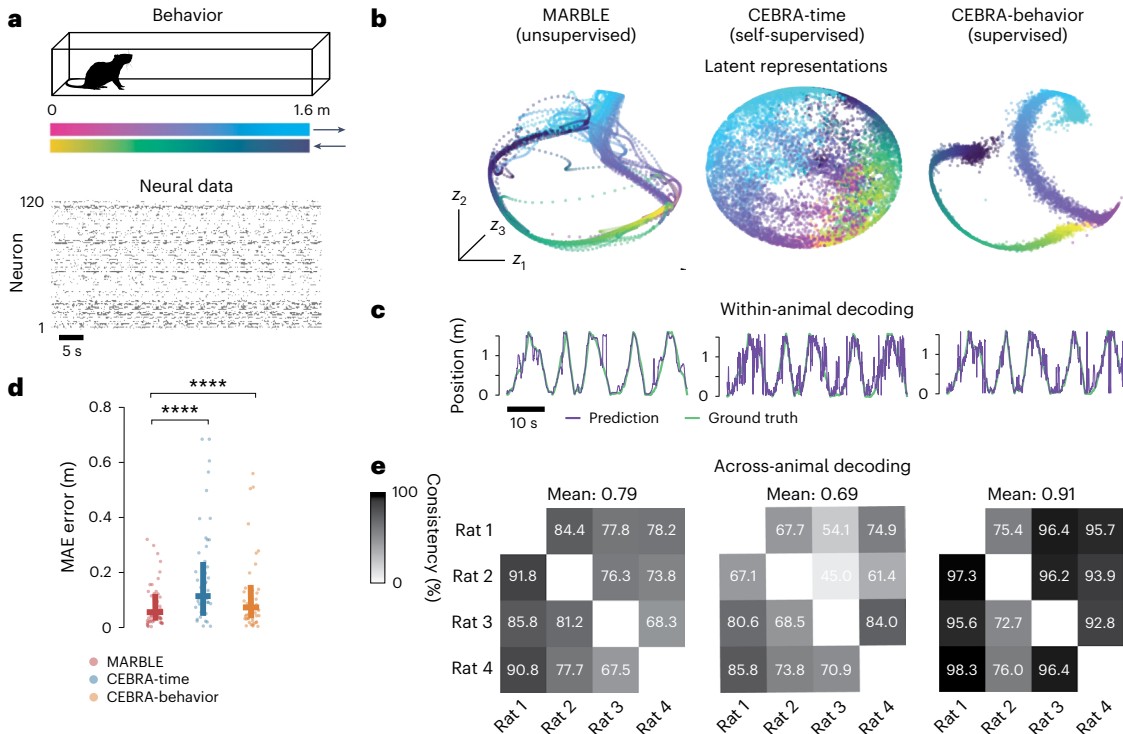

**Fig. 5 | Interpretable representation and cross-animal decoding of neural activity in rat hippocampus during linear maze navigation. a**, Experimental setup of a rat navigating a linear maze with tracked position and direction of motion (top). Raster plot showing spiking activity in 120 neurons in a single session (bottom). Rat image credit: [designer_an]/stock.adobe.com. **b**, Comparison of latent representation ($E = 3$) of unsupervised MARBLE against self-supervised (time-only labels) and supervised (time, position and direction labels) CEBRA.

Color shading is defined in **a. c**, Time traces of linearly decoded animal position within rat 1 ($E = 32$, default settings from CEBRA decoding notebook examples). **d**, Decoding accuracy within the same animal. Two-sided Wilcoxon tests (paired samples), ****$P < 1 \times 10^{-4}$. Horizontal and vertical bars show mean and $1 \times$ s.d., respectively ($n = 2,000$). **e**, Cross-animal consistency as measured by $R^2$ of linear fit between the optimally aligned 3D latent representations of a source animal to a target animal.

the neural states were clustered within a condition due to the supervision, meaning that temporal information was lost. Meanwhile, LFADS representations preserved the temporal information within trials but not the spatial structure (Fig. 4e). As expected[49], TDR also evidences the spatiotemporal structure of reaches; however, to a lower extent than MARBLE or CEBRA, yet with supervision using physical reach directions (Extended Data Fig. 8). Other unsupervised methods such as PCA, *t*-SNE or UMAP, which do not explicitly represent the dynamics, discovered no structure in the data (Extended Data Fig. 9). Hence, MARBLE can discover global geometric information in the neural code as an emergent property of LFFs.

This interpretability, based on a geometric correspondence between neural and behavioral representations, suggests a potent decoder. To show this, we fitted an optimal linear estimator between the latent representations and their corresponding hand positions, which is broadly used in brain–machine interfaces and measures interpretability based on how well the latent states parametrize complex non-linear dynamics. Notably, the decoded kinematics showed excellent visual correspondence to ground truth comparable to CEBRA-behavior and substantially better than LFADS (Fig. 4f and Extended Data Fig. 7d). A tenfold cross-validated classification of final reach direction and instantaneous velocity (Fig. 4g) confirmed that while reach direction could be decoded from a three-dimensional ($E = 3$) latent space, decoding the instantaneous velocity required higher latent space dimensions ($E = 20$; Fig. 4g) due to the variable delay between the GO cue and the onset of the movement. Of note, MARBLE outperformed competing methods in velocity decoding (Fig. 4g), showing that it can represent both the latency and full kinematics. Overall, MARBLE can infer representations of neural dynamics that are simultaneously interpretable and decodable into behavioral variables.

## Consistent latent neural representations across animals

Recent experiments evidence a strong similarity between neural representations across animals in a given task[4,18], with profound implications for brain–machine interfacing; however, as shown above (Fig. 3i,j), linear subspace alignment such as CCA[13] and related shape metrics[13,14,45] do not, in general, capture dynamical variation that otherwise preserves the geometry of the neural manifold. While it is possible to align multiple latent representations through auxiliary linear[30] or nonlinear[34] transformations, this relies on the assumption that the respective populations of neurons encode the same dynamical processes.

Given that MARBLE can produce latent representations that are comparable across RNNs (Fig. 3h,i) and are interpretable within an RNN (Fig. 3e–g) or animal (Fig. 4h–j), we finally asked whether it can produce consistently decodable latent representations across animals. To this end, we reanalyzed electrophysiological recordings from the rat hippocampus during navigation of a linear track[50] (Fig. 5a and Methods). From the neural data alone, MARBLE could infer interpretable representations consisting of a one-dimensional manifold in neural state space representing the animal's position and walking direction (Fig. 5b). Remarkably, unsupervised MARBLE representations were more interpretable than those obtained with CEBRA-time, supervised by time labels over neural states and comparable to CEBRA-behavior using behavior (both position and running direction) as labels (Fig. 5b). This finding was corroborated by significantly higher decoding accuracy (two-sided Wilcoxon tests, $P < 1 \times 10^{-4}$) using a *k*-means decoder (Fig. 5c,d).

When we aligned MARBLE representations post hoc using a linear transformation between animals, we found them to be consistent across animals. This consistency, quantified using the $R^2$ fit from a linear

model trained using one animal as the source and another animal as the target, was higher for MARBLE than for CEBRA-time, although not as good as CEBRA-behavior (Fig. 5e). This is notable given that MARBLE does not rely on behavioral data yet finds consistent representations despite experimental and neurophysiological differences across animals. These findings underscore MARBLE's potential for data-driven discovery and applications such as brain–computer interfaces.

## Discussion

A hallmark of large collective systems such as the brain is the existence of many system realizations that lead to equivalent computations defined by population-level dynamical processes[51,52]. The growing recognition that dynamics in biological and artificial neural networks evolve over low-dimensional manifolds[5,6,8,9] offers an opportunity to reconcile dynamical variability across system realizations with the observed invariance of computations by using manifold geometry as an inductive bias in learning dynamical representations. We have shown that nonlinear dynamical systems can be represented as a decomposition of LFFs that are jointly mapped into latent space. Due to the continuity of the dynamics over the manifold, this mapping can be learned using unsupervised geometric deep learning. Further, latent representations can be made robust to different embeddings of the dynamics by making the extracted LFFs rotation-invariant. These properties enable comparing dynamics across animals and instances of artificial neural networks.

To represent neural states, MARBLE uses condition labels to provide structural knowledge, namely, adjacency information among neural state trajectories that are dynamically consistent (share input patterns). While adjacency information allows extracting features (LFFs), it does not introduce a correlation between neural states (input) and latent states (output) as learning is performed via an unsupervised algorithm and the features impose no condition assignment as they are broadly shared between conditions. Our approach is similar in spirit to spectral clustering, where the user defines adjacency information, which is used to extract features (Laplacian eigenvectors) and then fed into an unsupervised clustering algorithm (k-means). Likewise, aggregating trials within conditions is similar to applying PCA to condition-averaged trials, yet MARBLE does not average across trials. This has profound implications for representing complex nonlinear state spaces where averaging is not meaningful. In contrast, in supervised representation learning[33,34], labels guide the embedding of neural states with similar labels close together and different ones far away. We note, however, that in terms of user input, MARBLE is more demanding than LFADS[30,31], which does not require condition labels. Thus, LFADS can be more suitable for systems where unexpected inputs can occur; however, we have shown that this mild assumption allows MARBLE to leverage the manifold geometry as an inductive bias to obtain more expressive representations.

Our formalism can be framed as a statistical generalization of the convergent cross-mapping framework by Sugihara et al.[35], which tests the causality between two dynamical systems through a one-to-one map between their LFFs. MARBLE generalizes this idea to a distributional comparison of the LFFs to provide a similarity metric between any collection of dynamical systems. In addition, due to the locality of representations, our approach diverges from typical geometric deep learning models that learn vector fields globally[53,54] and are thus unable to consider the manifold embedding and the dynamics separately. Locality also allows assimilating different datasets without additional trainable parameters to increase the statistical power of the model even when individual datasets are poorly sampled. Although our method does not explicitly learn time information[29,55], temporal ordering naturally emerges from our similarity-preserving mapping of LFFs into latent space. Beyond its use in interpreting and decoding neural dynamics, we expect MARBLE to provide powerful representations for general machine-learning tasks.

In summary, MARBLE's LFF learning approach enriches neural states with context information over a neural manifold to provide interpretable and consistent latent representations that were previously only attainable with supervised learning approaches that use additional behavioral information. This suggests that neural flow fields in different animals can be viewed as a projection of common latent dynamics and can be reconstructed as an emergent property of the similarity-preserving embedding of local flow fields.

## Online content

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

## Methods

MARBLE is a representation learning framework that works by decomposing a set of vector fields, representing possibly different dynamical systems or the same dynamical system under different latent parameters, into LFFs and then learning a similarity-preserving mapping from LFFs to a shared latent space. In latent space, the dynamical systems can be compared even if they were originally measured differently.

### Mathematical setup

As input, MARBLE takes a set of discrete vector fields $\mathbf{F}_c = \{\mathbf{f}_1(c), …, \mathbf{f}_n(c)\}$ supported over the point cloud $\mathbf{X}_c = (\mathbf{x}_1(c), …, \mathbf{x}_n(c))$, which describe a set of smooth, compact $m$-dimensional submanifolds $\mathcal{M}_c$ of the state space $\mathbb{R}^d$. Here, $c$ indicates an experimental condition or a dynamical system, $\mathbf{f}_i(c) \in \mathbb{R}^d$ is a vector signal anchored at a point $\mathbf{x}_i(c) \in \mathbb{R}^d$ and represented in absolute (world) coordinates in neural space or some $d$-dimensional representation space that preserves the continuity of the data. Given time series $\mathbf{x}(t; c)$, the vectors $\mathbf{f}_i(c)$ can be obtained, for example, by taking first-order finite differences $\mathbf{f}_i(c) := \mathbf{x}(t + 1; c) − \mathbf{x}(t; c)$. Given this input, MARBLE jointly represents the vector fields as empirical distributions $P(\mathbf{Z}_c) = \sum_{i=0}^n \delta(\mathbf{z}_i(c))$ of latent vectors $\mathbf{Z}_c = (\mathbf{z}_1(c), …, \mathbf{z}_n(c))$. The mapping $\mathbf{F}_c \mapsto \mathbf{Z}_c$ is point-by-point and local in the sense that $\{\mathbf{f}_j; j \in i \cup \mathcal{N}(i)\} \mapsto \mathbf{z}_i$, where $\mathcal{N}(i)$ represents the neighborhood of $i$. This locality means that it suffices to describe MARBLE applied to a single vector field. We will thus drop the subscript $c$ below for concise notation. The generalization of the method to the joint latent representation of multiple vector fields is immediate.

We now introduce an unsupervised geometric deep learning architecture for performing this embedding. The pseudocode for the MARBLE algorithm can be found in Supplementary Note 1.

### Data subsampling

We first subsample the data to ensure that LFFs are not overrepresented in any given region of the vector field due to sampling bias. We use farthest point sampling, a well-established technique in processing point clouds[56] that controls the spacing of the points relative to the diameter of the manifold $\max_{ij}(||\mathbf{x}_i − \mathbf{x}_j||_2) < \alpha \operatorname{diam}(\mathcal{M})$, where $\alpha \in [0, 1]$ is a parameter setting the spacing with 0 being no subsampling.

### Approximating the manifold by a proximity graph

We define locality on $\mathcal{M}$ by fitting a proximity graph to the point cloud $\mathbf{X}$. We use the continuous $k$-nearest neighbor ($ck$-NN) algorithm[57], which, contrary to the classical $k$-NN graph algorithm, can be interpreted as a local kernel density estimate and accounts for sampling density variations over $\mathcal{M}$. The $ck$-NN algorithm connects $i$ and $j$ whenever $||\mathbf{x}_i − \mathbf{x}_j||_2^2 < \delta ||\mathbf{x}_i − \mathbf{x}_u||_2 ||\mathbf{x}_j − \mathbf{x}_v||_2$, where $u, v$ are the $k$-th nearest neighbors of $i, j$, respectively, and $|| \cdot ||_2$ is the Euclidean norm. The scaling parameter $\delta$ can be used to control the number of nearest neighbors and thus the size of the neighborhood.

This proximity graph endows $\mathcal{M}$ with a geodesic structure (for any $i, j \in \mathcal{M}$), there is a shortest path with distance $d(i,j)$. We can then define the LFF at $i$ as the $p$-hop geodesic neighborhood $\mathcal{N}(i, p)$.

### Parametrizing the tangent spaces

To define trainable convolution filters over $\mathcal{M}$, we define tangent spaces $\mathcal{T}_i\mathcal{M}$ at each point $i$ over the manifold that linearly approximate $\mathcal{M}$ within a neighborhood. Specifically, we assume that the tangent space at a point $i$, $\mathcal{T}_i\mathcal{M}$, is spanned by the edge vectors $\mathbf{e}_{ij} \in \mathbb{R}^d$ pointing from $i$ to $K$ nodes $j$ in its neighborhood on the proximity graph. We pick $K > deg(i)$ closest nodes to $i$ on the proximity graph where $K$ is a hyperparameter. Larger $K$ increases the overlaps between the nearby tangent spaces and we find that $K = 1.5|\mathcal{N}(i, 1)|$ is often a good compromise between locality and robustness to noise of the tangent space approximation. The $m$

largest singular values $\mathbf{t}_i^{(\cdot)} \in \mathbb{R}^d$ of the matrix formed by column-stacking $\mathbf{e}_{ij}$ yield the orthonormal basis

$$\mathbb{T}_i \in \mathbb{R}^{d \times m} = \left(\mathbf{t}_i^{(1)}, … \mathbf{t}_i^{(m)}\right) \tag{1}$$

spanning $\mathcal{T}_i\mathcal{M}$. As a result, $\mathbb{T}_i^T \mathbf{f}_i$ acts as a projection of the signal to the tangent space in the $\ell_2$ sense. We perform these computations using a modified parallel transport unfolding package[58]. We illustrate the computed frames on a spherical manifold (Supplementary Fig. 1a−c).

### Connections between tangent spaces

Having the local frames, we next define the parallel transport map $\mathcal{P}_{j \to i}$ aligning the local frame at $j$ to that at $i$, which is necessary to define convolution operations in a common space (Supplementary Fig. 1d). While parallel transport is generally path dependent, we assume that adjacent nodes $i, j$ are close enough to consider the unique smallest rotation, known as the Lévy–Civita connection. Thus, for adjacent edges, $\mathcal{P}_{j \to i}$ can be computed as the matrix $\mathbf{O}_{ji}$ corresponding to $\mathcal{P}_{j \to i}$, as the orthogonal transformation (rotation and reflection)

$$\mathbf{O}_{ji} = \arg \min_{\mathbf{O} \in O(m)} ||\mathbb{T}_i − \mathbb{T}_j \mathbf{O}||_F, \tag{2}$$

where $|| \cdot ||_F$ is the Frobenius norm. The unique solution (up to a change of orientation) to equation (2) is found by the Kabsch algorithm[59]. See Supplementary Note 2 for further details.

### Vector diffusion

We use a vector diffusion layer to denoise the vector field (Fig. 1c), a generalization of the scalar (heat) diffusion, which can be expressed as a kernel associated with the equation[60]

$$\operatorname{vec}(\mathbf{F}(\tau)) = e^{-\tau \mathcal{L}} \operatorname{vec}(\mathbf{F}). \tag{3}$$

Here, $\operatorname{vec}(\mathbf{F}) \in \mathbb{R}^{nm \times 1}$ the row-wise concatenation of vector-valued signals, $\tau$ is a learnable parameter that controls the scale of the LFFs and $\mathcal{L}$ is the random-walk normalized connection Laplacian defined as a block matrix whose nonzero blocks are given by

$$\mathcal{L}(i,j) = \begin{cases} \mathbf{I}_{m \times m} & \text{for } i = j \\ -\deg(i)^{-1} \mathbf{O}_{ij} & \text{for } j \in \mathcal{N}(i, 1). \end{cases} \tag{4}$$

See ref. 61 for further details on vector diffusion. The advantage of vector diffusion over scalar (heat) diffusion is that it uses a notion of smoothness over the manifold defined by parallel transport and thus preserves the fixed point structure. The learnable parameter $\tau$ balances the expressivity of the latent representations with the smoothness of the vector field.

### Approximating local flow fields

We now define convolution kernels on $\mathcal{M}$ that act on the vector field to represent the vector field variation within LFFs. We first project the vector signal to the manifold $\mathbf{f}_i' = \mathbb{T}_i^T \mathbf{f}_i$. This reduces the dimension of $\mathbf{f}_i$ from $d$ to $m$ without loss of information as $\mathbf{f}_i$ was already in the tangent space. We drop the bar in the sequel to understand that all vectors are expressed in local coordinates. In this local frame, the best polynomial approximation of the vector field around $i$ is given by the Taylor-series expansion of each component $f_{i,l}$ of $\mathbf{f}_i$

$$f_{j,l} \approx f_{i,l} + \nabla f_{i,l}(\mathbf{x}_j − \mathbf{x}_i) + \frac{1}{2}(\mathbf{x}_j − \mathbf{x}_i)^T \nabla^2 f_{i,l}(\mathbf{x}_j − \mathbf{x}_i) + …. \tag{5}$$

We construct gradient filters to numerically approximate the gradient operators of increasing order in the Taylor expansion (see Supplementary Note 3 for details). In brief, we implement the first-order gradient

operator as a set of $m$ directional derivative filters $\{\mathcal{D}^{(q)}\}$ acting along unit directions $\{\mathbf{t}_i^{(q)}\}$ of the local coordinate frame,

$$\nabla f_{i,l} \approx \left(\mathcal{D}^{(1)}(f_{i,l}), \dots, \mathcal{D}^{(m)}(f_{i,l})\right)^T. \tag{6}$$

The directional derivative, $\mathcal{D}^{(q)}(f_{i,l})$ is the $l$-th component of

$$\mathcal{D}^{(q)}(\mathbf{f}_i) = \sum_{j=1}^{n} \mathcal{K}_j^{(i,q)} \mathcal{P}_{\to i}(\mathbf{f}_j), \tag{7}$$

where $\mathcal{P}_{\to i} = \mathbf{O}_{ij}$ is the parallel transport operator that takes the vector $\mathbf{f}_j$ from the adjacent frame $j$ to a common frame at $i$. $\mathcal{K}^{(i,q)} \in \mathbb{R}^{n \times n}$ is a directional derivative filter[39] expressed in local coordinates at $i$ and acting along $\mathbf{t}_i^{(q)}$. See Supplementary Note 3 for details on the construction of the directional derivative filter. As a result of the parallel transport, the value of equation (7) is independent of the local curvature of the manifold.

Following this construction, the $p$-th order gradient operators can be defined by the iterated application of equation (6), which aggregates information in the $p$-hop neighborhood of points. Although increasing the order of the differential operators increases the expressiveness of the network (Supplementary Fig. 3), second-order filters ($p = 2$) were sufficient for the application considered in this paper.

The expansion in equation (5) suggests augmenting the vectors $\mathbf{f}_i$ by the derivatives (equation (6)), to obtain a matrix

$$\mathbf{f}_i \mapsto \mathbf{f}_i^{\mathcal{D}} = \left(\mathbf{f}_i, \nabla f_{i,1}, \dots, \nabla f_{i,m}, \nabla(\nabla f_{i,1})_1, \dots, \nabla(\nabla f_{i,m})_m\right), \tag{8}$$

of dimensions $m \times c$ whose columns are gradients of signal components up to order $p$ to give a total of $c = (1 - m^{p+1})/(m(1-m))$ vectorial channels.

## Inner product for embedding invariance

Deformations on the manifold have the effect of introducing rotations into the LFFs. In embedding-agnostic mode, we can achieve invariance to these deformations by making the learned features rotation-invariant. We do so by first transforming the $m \times c$ matrix $\mathbf{f}_i^{\mathcal{D}}$ to a $1 \times c$ vector as

$$\mathbf{f}_i^{\mathcal{D}} \mapsto \mathbf{f}_i^{\text{ip}} = \left(\mathcal{E}^{(1)}(\mathbf{f}_i^{\mathcal{D}}), \dots, \mathcal{E}^{(c)}(\mathbf{f}_i^{\mathcal{D}})\right). \tag{9}$$

Then, by taking for each channel the inner product against all other channels, weighted by a dense learnable matrix $\mathbf{A}^{(r)} \in \mathbb{R}^{m \times m}$ and summing, we obtain

$$\mathcal{E}^{(r)}(\mathbf{f}_i^{\mathcal{D}}) = \mathcal{E}^{(r)}\left(\mathbf{f}_i^{\mathcal{D}}; \mathbf{A}^{(r)}\right) := \sum_{s=1}^{c} \left\langle \mathbf{f}_i^{\mathcal{D}}(\cdot, r), \mathbf{A}^{(r)} \mathbf{f}_i^{\mathcal{D}}(\cdot, s) \right\rangle, \tag{10}$$

for $r = 1, \dots, c$ (Fig. 1f). Taking inner products is valid because the columns of $\mathbf{f}_i^{\mathcal{D}}$ all live in the tangent space at $i$. Intuitively, equation (10) achieves coordinate independence by learning rotation and scaling relationships between pairs of channels.

## Latent space embedding with a multilayer perceptron

To embed each local feature, $\mathbf{f}_i^{\text{ip}}$ or $\mathbf{f}_i^{\mathcal{D}}$, depending on if inner product features are used (equation (9)) we use a multilayer perception (Fig. 1g)

$$\mathbf{z}_i = \text{MLP}\left(\mathbf{f}_i^{\text{ip}}; \omega\right), \tag{11}$$

where $\omega$ are trainable weights. The multilayer perceptron is composed of $L$ linear (fully connected) layers interspersed by rectified linear unit (ReLU) nonlinearities. We used $L = 2$ with a sufficiently high output dimension to encode the variables of interest. The parameters were initialized using the Kaiming method[62].

## Loss function

Unsupervised training of the network is possible due to the continuity in the vector field over $\mathcal{M}$, which causes nearby LFFs to be more similar than distant ones. We implement this via negative sampling[40], which uses random walks sampled at each node to embed neighboring points on the manifold close together while pushing points sampled uniformly at random far away. We use the following unsupervised loss function[40]

$$\mathcal{J}(\mathbf{Z}) = -\log\left(\sigma\left(\mathbf{z}_i^T \mathbf{z}_j\right)\right) - Q\mathbb{E}_{k \sim U(n)} \log\left(\sigma\left(-\mathbf{z}_i^T \mathbf{z}_k\right)\right), \tag{12}$$

where $\sigma(x) = (1 + e^{-x})^{-1}$ is the sigmoid function and $U(n)$ is the uniform distribution over the $n$ nodes. To compute this function, we sample one-step random walks from every node $i$ to obtain 'positive' node samples for which we expect similar LFFs to that at node $i$. The first term in equation (12) seeks to embed these nodes close together. At the same time, we also sample nodes uniformly at random to obtain 'negative' node samples with likely different LFFs from that of node $i$. The second term in equation (12) seeks to embed these nodes far away. We also choose $Q = 1$.

We optimize the loss equation (12) by stochastic gradient descent. For training, the nodes from all manifolds were randomly split into training (80%), validation (10%) and test (10%) sets. The optimizer was run until convergence of the validation set and the final results were tested on the test set with the optimized parameters.

## Distance between latent representations

To test whether shifts in the statistical representation of the dynamical system can predict global phenomena in the dynamics, we define a similarity metric between pairs of vector fields $\mathbf{F}_1$, $\mathbf{F}_2$ with respect to their corresponding latent vectors $\mathbf{Z}_1 = (\mathbf{z}_{1,1}, \dots, \mathbf{z}_{n_1,1})$ and $\mathbf{Z}_2 = (\mathbf{z}_{1,1}, \dots, \mathbf{z}_{n_2,1})$. We use the optimal transport distance between the empirical distributions $P_1 = \sum_i^{n_1} \delta(\mathbf{z}_{i,1})$, $P_2 = \sum_i^{n_2} \delta(\mathbf{z}_{i,2})$

$$d(P_1, P_2) = \min_{\gamma} \sum_{uv} \gamma_{uv} ||\mathbf{z}_{u,1} - \mathbf{z}_{v,2}||_2^2, \tag{13}$$

where $\gamma$ is the transport plan, a joint probability distribution subject to marginality constraints that $\sum_u \gamma_{uv} = P_2$, $\sum_v \gamma_{uv} = P_1$ and $|| \cdot ||_2$ is the Euclidean distance.

## Further details on the implementation of case studies

Below we detail the implementation of the case studies. See Supplementary Table 3 for the training hyperparameters. In each case, we repeated training five times and confirmed that the results were reproducible.

**Van der Pol.** We used the following equations to simulate the Van der Pol system:

$$\begin{aligned} \dot{x} &= y \\ \dot{y} &= \mu(1 - x^2)y - x, \end{aligned} \tag{14}$$

parametrized by $\mu$. If $\mu = 0$, the system reduces to the harmonic oscillator; if $\mu < 0$, the system is unstable, and if $\mu > 0$, the system is stable and converges to a limit cycle. In addition, we map this two-dimensional system to a paraboloid as with the map

$$\begin{aligned} x, y &\mapsto x, y, z = \text{parab}(x, y) \\ \dot{x}, \dot{y} &\mapsto \dot{x}, \dot{y}, \dot{z} = \text{parab}(x + \dot{x}, y + \dot{y}) - \text{parab}(x, y), \end{aligned}$$

where $\text{parab}(x, y) = -(\alpha x)^2 - (\alpha y)^2$.

We sought to distinguish on-manifold dynamical variation due to $\mu$ while being agnostic to geometric variations due to $\alpha$. As conditions, we increased $\mu$ from $-1$, which first caused a continuous deformation in

the limit cycle from asymmetric (corresponding to slow–fast dynamics) to circular and then an abrupt change in crossing the Hopf bifurcation at zero (Fig. 1d).

We trained MARBLE in both embedding-aware and -agnostic modes by forming distinct manifolds from the flow field samples at different values of $\mu$ and manifold curvature. Both the curvature and the sampling of the vector field differed across manifolds.

**Low-rank RNNs.** We consider low-rank RNNs composed of $n = 500$ rate units in which the activation of the $i$-th unit is given by

$$\tau \frac{dx_i}{dt} = -x_i + \sum_{j=1}^{N} J_{ij}\phi(x_j) + \bar{u}_i(t) + \eta_i(t), \quad x_i(0) = 0, \qquad (15)$$

where $\tau = 100$ ms is a time constant, $\phi(x_i) = \tanh(x_i)$ is the firing rate, $J_{ij}$ is the rank-$R$ connectivity matrix, $u_i(t)$ is an input stimulus and $\eta_i(t)$ is a white noise process with zero mean and s.d. of $3 \times 10^{-2}$. The connectivity matrix can be expressed as

$$\mathbf{J} = \frac{1}{N} \sum_{r=1}^{R} \mathbf{m}_r \mathbf{n}_r^T, \qquad (16)$$

for vector pairs $(\mathbf{m}_r, \mathbf{n}_r)$. For the delayed-match-to-sample task, the input is of the form

$$\bar{u}_i(t) = w_{1i}^{in} u_1(t) + w_{2i}^{in} u_2(t), \qquad (17)$$

where $w_{1i}, w_{2i}$ are coefficients controlling the weight of inputs $u_1, u_2$ into node $i$. Finally, the network firing rates are read out to the output as

$$o(t) = \sum_{i=0}^{N} w_i^{out} \phi(x_i). \qquad (18)$$

To train the networks, we followed ref. 47. The experiments consisted of five epochs; a fixation period of 100–500 ms chosen uniformly at random, a 500-ms stimulus period, a delay period of 500–3,000 ms chosen uniformly at random, a 500-ms stimulus period and a 1,000-ms decision period. During training, the networks were subjected to two inputs, whose magnitude (the gain) was positive during stimulus and zero otherwise. We used the following loss function:

$$\mathcal{L} = |o(T) - \hat{o}(T)|, \qquad (19)$$

where $T$ is the length of the trial and $\hat{o}(T) = 1$ when both stimuli were present and $-1$ otherwise. Coefficient vectors were initially drawn from a zero mean, unit s.d. Gaussian and then optimized. For training, we used the ADAM optimizer[63] with hyperparameters shown in Supplementary Tables 2 and 3.

See the main text for details on how MARBLE networks were trained.

**Macaque center-out arm-reaching.** We used single-neuron spike train data as published in ref. 30 recorded using linear multielectrode arrays (V-Probe, 24-channel linear probes) from rhesus macaque motor (M1) and dorsal premotor cortices. See ref. 30 for further details. In brief, each trial began with the hand at the center. After a variable delay, one target, 10 cm from the center position, was highlighted, indicating the GO cue. We analyzed the 700-ms period after the go consisting of a delay followed by the reach. A total of 44 consecutive experimental sessions with a variable number of trials were considered.

We extracted the spike trains using the neo package in Python (http://neuralensemble.org/neo/) and converted them into rates using a Gaussian kernel with a s.d. of 100 ms. We subsampled the rates at 20-ms intervals using the elephant package[64] to match the sampling frequency in the decoded kinematics in ref. 30. Finally, we used PCA

to reduce the dimension from 24-channels to five. In Supplementary Fig. 4, we analyze the sensitivity to preprocessing hyperparameters, showing that our results remain stable for a broad range of Gaussian kernel scales. We also note that decoding accuracy marginally increased for seven and ten principal components at the cost of slower training time.

We trained MARBLE in the embedding-aware mode for each session, treating movement conditions as individual manifolds, which we embedded into a shared latent space.

We benchmarked MARBLE against LFADS, CEBRA (Fig. 4), TDR (Extended Data Fig. 8), PCA, UMAP and $t$-SNE (Extended Data Fig. 9). For LFADS, we took the trained models directly from the authors. For CEBRA, we used the reach directions as labels and trained a supervised model until convergence. We obtained the best results with an initial learning rate of 0.01, Euclidean norm as metric, number of iterations 10,000 and fixed temperature 1. For TDR, we followed the procedure in ref. 22. We represented the condition-averaged firing rate for neuron $i$ at time $t$ as

$$r_{i,c}(t) = \beta_{i,0}(t) + \beta_{i,x}(t)c_x + \beta_{i,y}(t)c_y \qquad (20)$$

where $(c_x, c_y)$ denotes the regressors (the final direction in the physical space of the reaches), and $\beta$ are corresponding time-dependent coefficients to be determined. For the seven reach directions, the regressors $(c_x, c_y)$ takes the following values: 'DownLeft' $(-1, -1)$, 'Left' $(-1, 0)$, 'UpLeft' $(-1, 1)$, 'Up' $(0, 1)$, 'UpRight' $(1, 1)$, 'Right' $(1, 0)$ and 'DownRight' $(1, -1)$. To estimate $(c_x, c_y)$, we constructed the following matrix $M$ with shape conditions× regressors. All but the last column of $M$ contained the condition × condition values of one of the regressors. The last column consisted only of ones to estimate $\beta_0$. Given the conditions × neurons matrix of neural firing rates $R$ and conditions × regressors matrix $M$, the regression model can be written as:

$$R(t) = M[\beta_{i,0}, \beta_{i,x}, \beta_{i,y}]. \qquad (21)$$

We perform least squares projection to estimate the regression coefficients

$$[\beta_{i,0}, \beta_{i,x}, \beta_{i,y}] = (M^T M)^{-1} M^T R. \qquad (22)$$

We projected neural data into the regression subspace by multiplying the pseudoinverse of the coefficient matrix with the neural data matrix $R$. For decoding, we used the estimated regression coefficients to project single-trial firing rates to the appropriate condition-dependent subspaces.

To decode the hand kinematics, we used optimal linear estimation to decode the $x$ and $y$ reaching coordinates and velocities from the latent representation, as in ref. 30. To assess the accuracy of decoded movements, we computed the tenfold cross-validated goodness of fit ($R^2$) between the decoded and measured velocities for $x$ and $y$ before taking the mean across them. We also trained a support vector machine classifier (regularization of 1.0 with a radial basis function) on the measured kinematics against the condition labels.

**Linear maze navigation.** We used single-neuron spiking data from ref. 50, where we refer the reader for experimental details. In brief, neural activity was recorded in CA1 pyramidal single units using two 8- or 6-shank silicon probes in the hippocampus in four rats while walking in alternating directions in a 1.6-m linear track. Each recording had between 48–120 recorded neurons.

We extracted the spike trains using the neo package (http://neuralensemble.org/neo/) and converted them into rates using a Gaussian kernel with an s.d. of 10 ms. We fitted a PCA across all data for a given animal to reduce the variable dimension to five. In Supplementary Fig. 5, we analyze the effect of varying preprocessing hyperparameters,

showing that our results remain stable for a broad range of Gaussian kernel scales and principal components.

We trained MARBLE in embedding-aware mode separately on each animal. The output dimensions were matched to that of the benchmark models.

For benchmarking, we took the CEBRA models unchanged from the publicly available notebooks. To decode the position, we used a 32-dimensional output and 10,000 iterations as per their demo notebook (https://cebra.ai/docs/demo_notebooks/Demo_decoding.html). To decode the rat's position from the neural trajectories, we fit a $k$-NN decoder with 36 neighbors and a cosine distance metric. To assess the decoding accuracy, we computed the mean absolute error between the predicted and true rat positions.

For consistency between animals, we used a three-dimensional ($E = 3$) output and 15,000 iterations, as per the notebook https://cebra.ai/docs/demo_notebooks/Demo_consistency.html. We aligned the latent representations across animals post hoc using Procrustes analysis. We fit a linear decoder to the aligned latent representations between pairs of animals: one animal as the source (independent variables) and another as the target (dependent variables). The $R^2$ from the fitted model describes the variance in the latent representation of one animal that can be explained by another animal (a measure of consistency between their latent representations).

### Reporting summary

Further information on research design is available in the Nature Portfolio Reporting Summary linked to this article.

### Data availability

The data generated during the simulations are available at https://doi.org/10.7910/DVN/KTE4PC (ref. 65).

### Code availability

The code to carry out the simulations and analysis can be found at https://github.com/Dynamics-of-Neural-Systems-Lab/MARBLE under MIT license.

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

### Acknowledgements

We thank N. Aspert for much-needed computing support. We also thank A. Valente, N. Karalias and M. Vinao-Carl for the interesting discussions. A.G. acknowledges support from an Human Frontiers Science Programme (HFSP) Cross-disciplinary Postdoctoral Fellowship (LT000669/2020-C). M.B. acknowledges funding through the Engineering and Physical Sciences Research Council (EPSRC), award number EP/N014529/1 (Center for Mathematics of Precision Healthcare) and EP/W024020/1 (Statistical Physics of Cognition). R.P. acknowledges the Deutsche Forschungsgemeinschaft (German Research Foundation) project ID 424778381-TRR 295. A.A. was supported by funding to the Blue Brain Project, a research center of the École Polytechnique Fédérale de Lausanne, from the Swiss Government's Eidgenössische Technische Hochschule (ETH) Board of the Swiss Federal Institutes of Technology.

### Author contributions

Conceptualization: A.G., R.L.P., A.A., M.B. and P.V.; methodology: A.G., R.L.P., A.A. and P.V.; software: A.G., R.L.P. and A.A.; analysis: A.G., R.L.P. and A.A.; writing - original draft: A.G., R.L.P. and A.A.; and writing - review & editing: M.B. and P.V.

### Funding

### Competing interests

The authors declare no competing interests.

### Additional information

**Extended data** is available for this paper at https://doi.org/10.1038/s41592-024-02582-2.

**Correspondence and requests for materials** should be addressed to Adam Gosztolai.

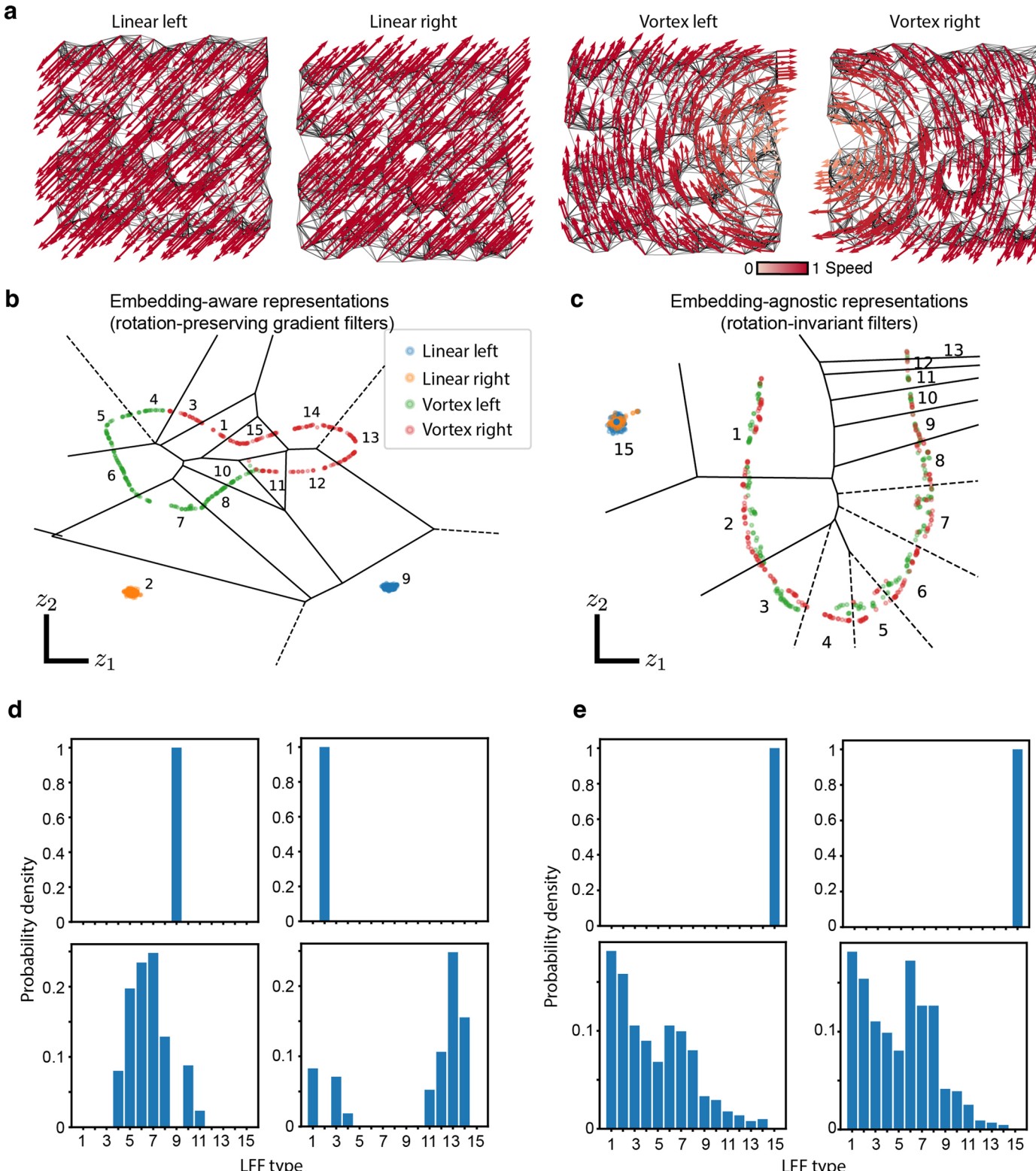

**Extended Data Fig. 1 | Embedding-invariant representations on rotational vector fields. a** Vector fields sampled uniformly at random (n=512) in the interval $[-1, 1]^2$ and fitted with a continuous k-nearest neighbor graph (black lines, k=20). **b** Embedding-aware joint MARBLE representation of the vector fields using second-order (p=2) gradient filters (rotation-preserving). Each point represents an LFF and points close together represent similar LFFs. Features from the linear vector fields aggregate (clusters 2 and 9), while those from the vortex fields fall on separate halves of a one-dimensional circular manifold corresponding to the one-parameter (angle) variation between them. Black lines show k-means clustering (k=15). **c** Embedding-agnostic joint MARBLE representations but with rotation-invariant filters. Features from linear fields can no longer be distinguished (cluster 15) because the filter does not preserve rotational information. Features from vortex fields fall on a linear one-dimensional manifold parametrized by the distance from the center. **d** The histogram of rotation-preserving features can distinguish all fields. **e** The histogram of rotation-invariant features can discriminate linear fields from vortex fields but not the orientation.

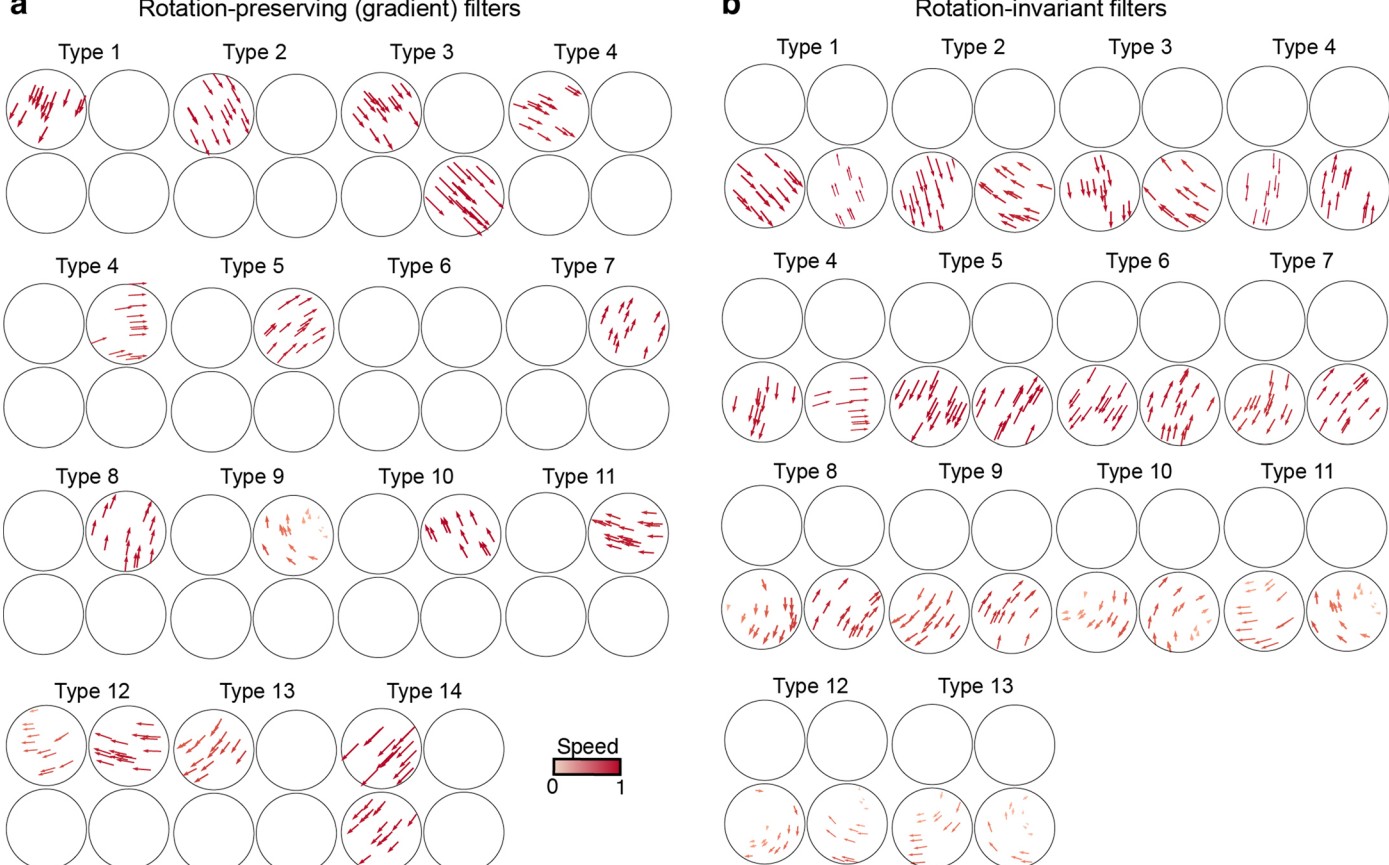

**Extended Data Fig. 2 | Effect of embedding-agnostic MARBLE representations on local flow fields.** Samples of local flow fields (LFFs), each drawn randomly from the corresponding MARBLE representation in Extended Data Fig. 1b-c.

The types correspond to the clusters. **a** For embedding-aware representations, LFFs rotate counter-clockwise as the cluster number increases. **b** For embedding-agnostic representations, LFFs contract as cluster number increases.

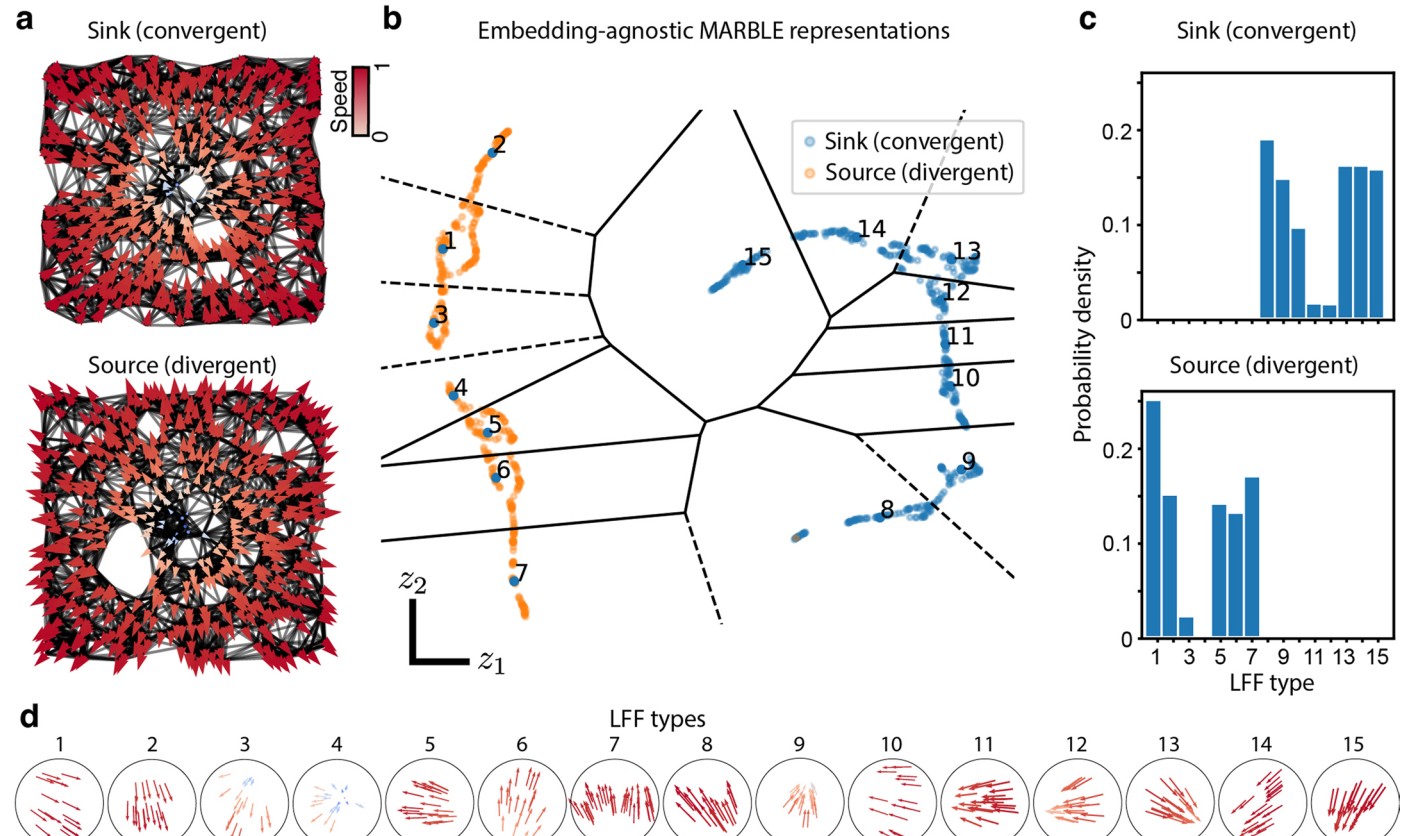

**Extended Data Fig. 3 | Embedding-invariant representations of convergent and divergent vector fields. a** Convergent and divergent vector fields sampled uniformly at random (n=512) in the interval [−1, 1]². **b** Embedding-agnostic MARBLE representations can distinguish the fields, even without rotational information. Black lines show k-means clustering (k=15). **c** Histogram of LFF types confirming the disjoint representations of the two flow fields. **d** One LFF drawn randomly from each cluster displays expanding LFFs for types 1−8 and contracting LFFs for types 9-15.

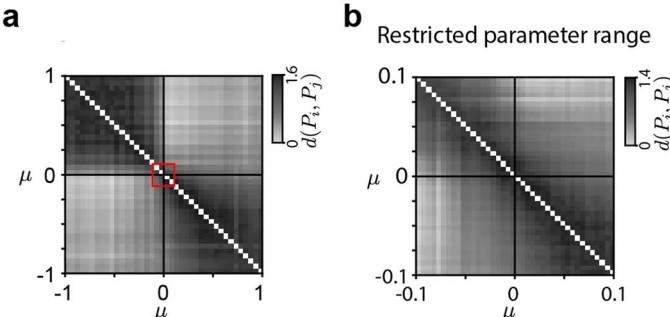

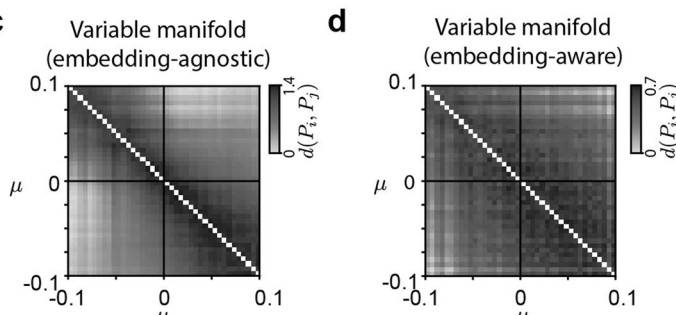

**Extended Data Fig. 4 | Embedding-agnostic MARBLE representations over deforming manifolds.** We illustrate the MARBLE representation of the Van der Pol oscillator example with 40 values of $\mu$. **a** For the parameter range $\mu \in [-1, 1]$ the dynamical regimes (stable/unstable) are clearly defined as clusters. **b** As in a, but for a restricted parameter regime $\mu \in [-0.1, 0.1]$ to highlight the definition of the cluster borders. **c** In embedding-agnostic mode, varying the curvature of the paraboloid $\beta(x^2 + y^2)$ by drawing $\beta$ uniformly at random from the internal $\beta \in [-0.2, 0.2]$ does not alter the MARBLE representation. **d** In embedding-aware mode, the same variation as in c destroys the cluster structure.

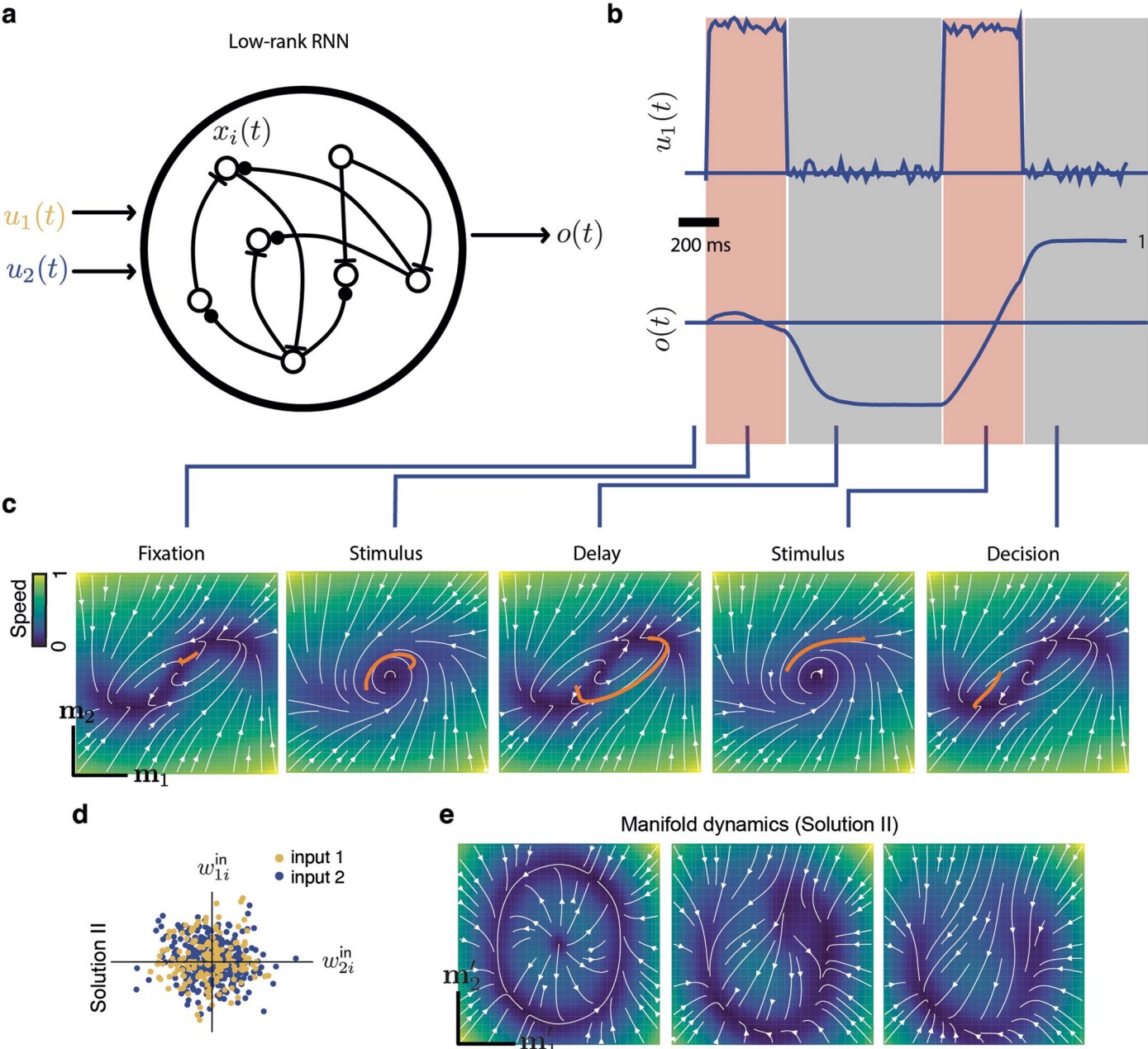

**Extended Data Fig. 5 | Structure and activity of a rank-2 RNN during the delayed-match-to-sample task. a** Schematic of a low-rank RNN, taking as input two stimuli and producing a decision variable as the output. Arrow endings represent inhibitory and excitatory connections. **b** Example input pattern for one of the stimuli and corresponding output pattern. Red and gray-shaded bands show stimulated and unstimulated periods. **c** Mean-field dynamics (heatmap and stream plot) superimposed with a sampled trajectory (orange) during one trial. **d** Space of input weights in a trained RNN. Unlike in Fig. 3d, here the network generalizes, that is, gives the same weighting to each input. Colors obtained by k-means clustering indicate two subpopulations specialized in one or the other input. **e** Mean-field dynamics for the low-rank RNN with input weights in d.

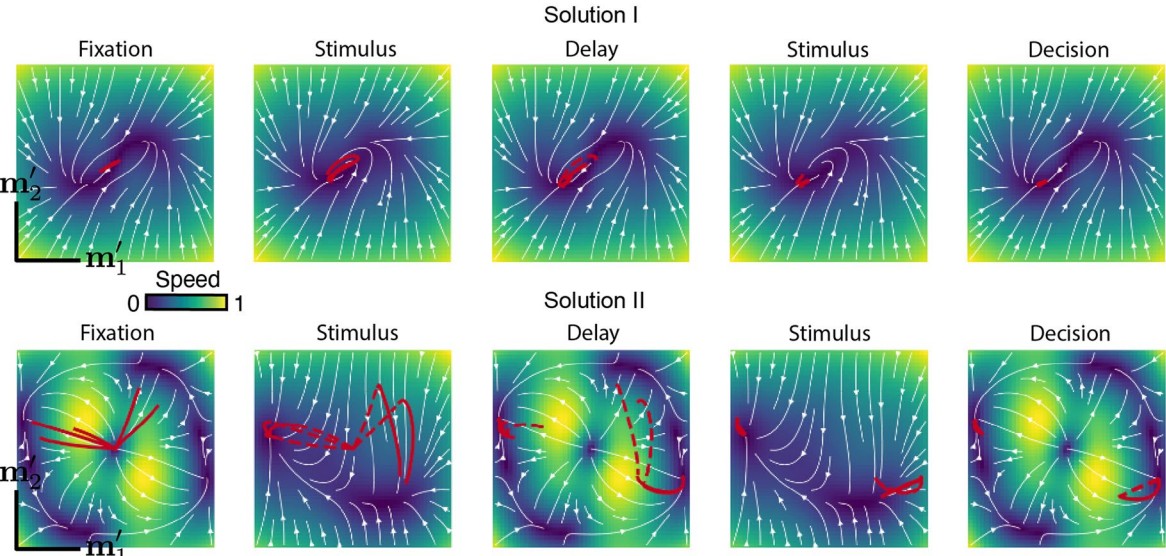

**Extended Data Fig. 6 | Qualitatively distinct RNN solutions to the DMS task.** Heatmaps and steam plots show the mean-field dynamics for the two independently trained RNN solutions, superimposed with five trajectories illustrating single-trial instances of the network dynamics.

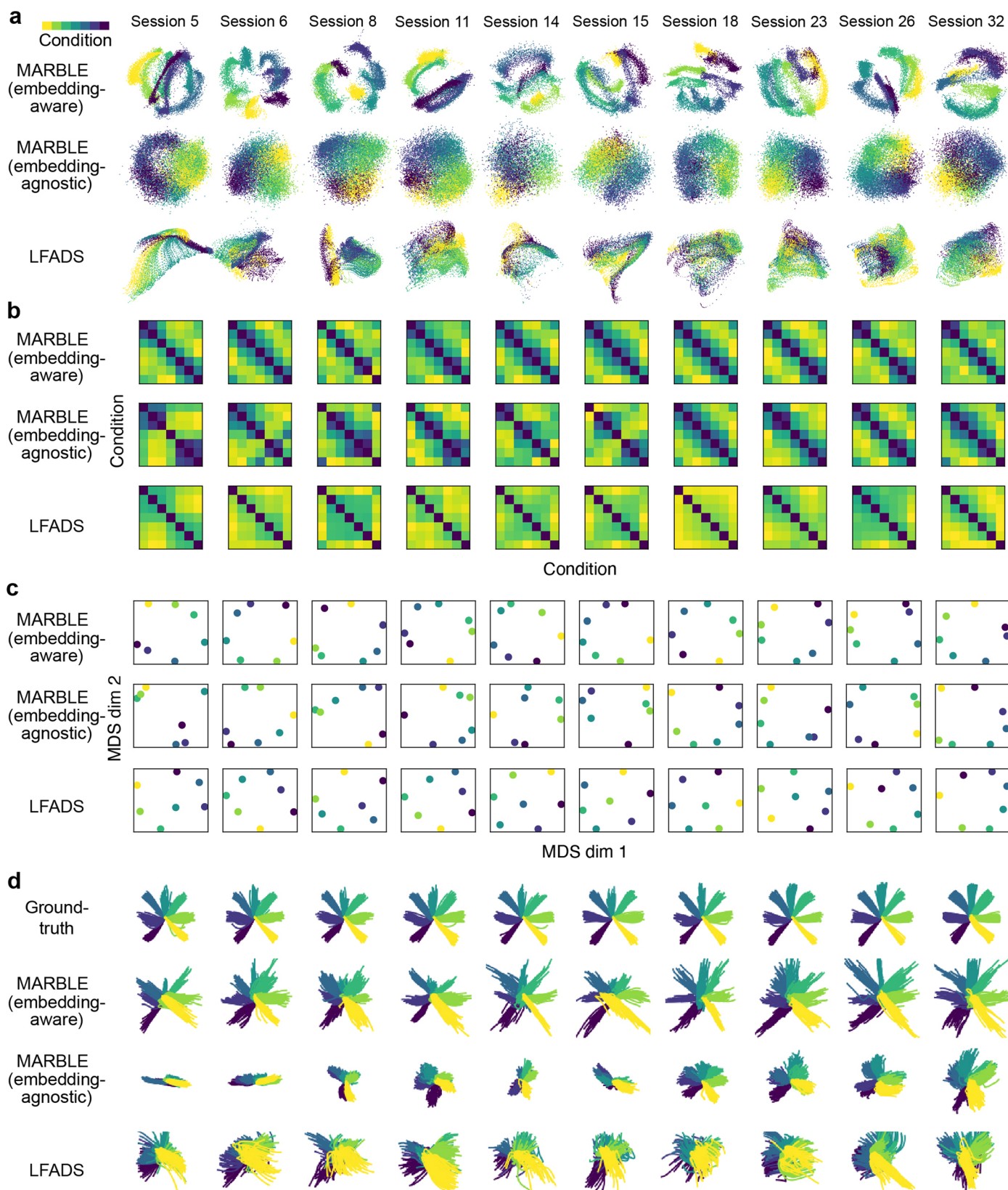

**Extended Data Fig. 7 | Comparison of MARBLE and LFADS for individual sessions of macaque reaching task. a** MARBLE representations (E=3) better reflect the arrangement of reaches in physical space when compared to LFADS. **b** The matrix of optimal transport distances between pairwise conditions within sessions shows a stronger periodic structure for MARBLE than LFADS. **c** MDS embedding of the distance matrix more consistently recovers the spatial arrangement of reaches for embedding-aware MARBLE, when compared to embedding-agnostic MARBLE and LFADS. **d** Hand trajectories linearly decoded from MARBLE representations show much stronger spatial correspondence to ground-truth kinematics than LFADS.

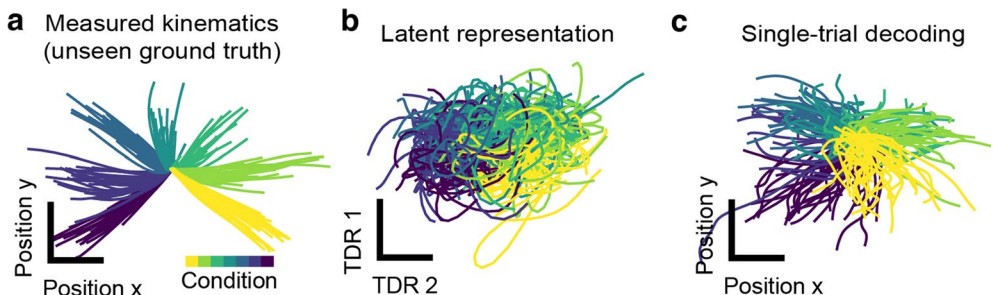

**Extended Data Fig. 8 | Representation and decoding of neural signals during macaque arm-reaching kinematics using supervised targeted dimensionality reduction (TDR).** This method regresses neural firing rates to subspaces defined by the different reaching directions. **a** Ground truth hand trajectories of a macaque in seven reach conditions. **b** Latent representation of neural data across conditions in a single session regressed to the corresponding TDR subspaces. **c** Linear decoding of hand trajectories from latent representations.

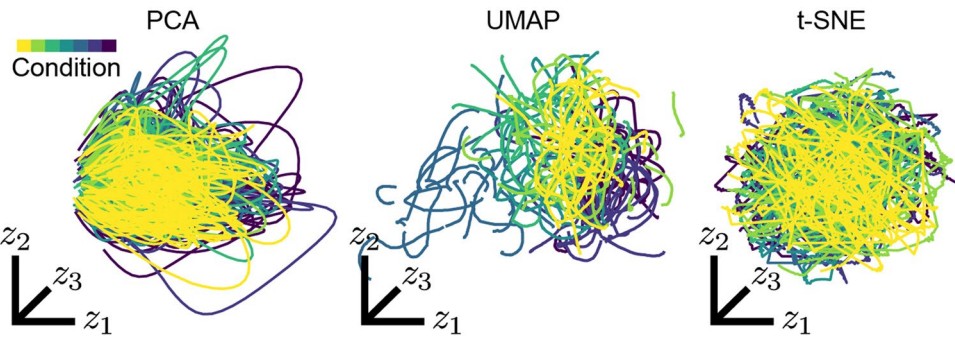

**Extended Data Fig. 9 | Latent representations of neural signals during macaque arm-reaching kinematics using unsupervised PCA, t-SNE and UMAP.**
These methods only map the neural states and do not explicitly account for the dynamics. Hence, they cannot reveal the geometric structure of latent dynamics.

# Reporting Summary

## Statistics

For all statistical analyses, confirm that the following items are present in the figure legend, table legend, main text, or Methods section.

| n/a | Confirmed | |
|---|---|---|
| ☐ | ☒ | The exact sample size (*n*) for each experimental group/condition, given as a discrete number and unit of measurement |
| ☐ | ☒ | A statement on whether measurements were taken from distinct samples or whether the same sample was measured repeatedly |
| ☐ | ☒ | The statistical test(s) used AND whether they are one- or two-sided<br>*Only common tests should be described solely by name; describe more complex techniques in the Methods section.* |
| ☐ | ☒ | A description of all covariates tested |
| ☐ | ☒ | A description of any assumptions or corrections, such as tests of normality and adjustment for multiple comparisons |
| ☐ | ☒ | A full description of the statistical parameters including central tendency (e.g. means) or other basic estimates (e.g. regression coefficient) AND variation (e.g. standard deviation) or associated estimates of uncertainty (e.g. confidence intervals) |
| ☐ | ☒ | For null hypothesis testing, the test statistic (e.g. *F*, *t*, *r*) with confidence intervals, effect sizes, degrees of freedom and *P* value noted<br>*Give P values as exact values whenever suitable.* |
| ☒ | ☐ | For Bayesian analysis, information on the choice of priors and Markov chain Monte Carlo settings |
| ☒ | ☐ | For hierarchical and complex designs, identification of the appropriate level for tests and full reporting of outcomes |
| ☒ | ☐ | Estimates of effect sizes (e.g. Cohen's *d*, Pearson's *r*), indicating how they were calculated |

*Our web collection on statistics for biologists contains articles on many of the points above.*

## Software and code

Policy information about availability of computer code

| Data collection | The data in this study partly constitutes experimental data from the literature and partly generated by custom code. The code used to generate the synthetic datasets can be found at https://github.com/agosztolai/MARBLE. |
|---|---|
| Data analysis | The code to carry out the simulations and analysis can be found at https://github.com/agosztolai/MARBLE. |

For manuscripts utilizing custom algorithms or software that are central to the research but not yet described in published literature, software must be made available to editors and reviewers. We strongly encourage code deposition in a community repository (e.g. GitHub). See the Nature Portfolio guidelines for submitting code & software for further information.

## Data

Policy information about availability of data

All manuscripts must include a data availability statement. This statement should provide the following information, where applicable:
- Accession codes, unique identifiers, or web links for publicly available datasets
- A description of any restrictions on data availability
- For clinical datasets or third party data, please ensure that the statement adheres to our policy

The data generated during the simulations is available here https://doi.org/10.7910/DVN/KTE4PC.

## Human research participants

Policy information about studies involving human research participants and Sex and Gender in Research.

| | |
|---|---|
| Reporting on sex and gender | Not applicable. |
| Population characteristics | Not applicable. |
| Recruitment | Not applicable. |
| Ethics oversight | Not applicable. |

Note that full information on the approval of the study protocol must also be provided in the manuscript.

# Field-specific reporting

Please select the one below that is the best fit for your research. If you are not sure, read the appropriate sections before making your selection.

☒ Life sciences      ☐ Behavioural & social sciences      ☐ Ecological, evolutionary & environmental sciences

For a reference copy of the document with all sections, see nature.com/documents/nr-reporting-summary-flat.pdf

# Life sciences study design

All studies must disclose on these points even when the disclosure is negative.

| | |
|---|---|
| Sample size | No explicit a priori sample size calculation was performed since we used publicly available datasets and synthetic data.<br><br>For the experiments involving synthetic data, the sample size was chosen to ensure that the dynamic changes of interest could be reliably detected and statistically evaluated. Specifically, we generated a sufficiently large number of samples (200 trajectories for RNN) to observe the variability of the system across different conditions (changes in gain) and to ensure that the trends were stable and not due to random fluctuations. Since we did not perform a statistical test here - no explicit sample size calculation was necessary.<br><br>For the macaque experimental dataset, we used all available data from the repository https://github.com/nplcode/lfads-neural-stitching-reproduce. The size of this dataset is inherently determined by the number of trials and recordings in the original experimental design. Given that this data has been widely used in the literature and is considered robust, we believe the sample size is sufficient to provide valid insights for our analyses.<br><br>Similarly, for the rat experimental dataset, we used the full dataset available from the repository https://crcns.org/data-sets/hc/hc-11/about-hc-11 As with the macaque data, this dataset represents a comprehensive set of trials designed to capture the behavioral and neural dynamics under study. Thus, the sample size reflects the original experiment's design, which is presumed to be sufficient for detecting the relevant neural or behavioral effects. |
| Data exclusions | No data was excluded. |
| Replication | MARBLE neural networks were trained five times. The results obtained on each attempt were consistent and reproducible. |
| Randomization | Randomization was not applicable to this study because the data used were derived from pre-existing datasets (synthetic data, macaque experimental data, and rat experimental data) collected under controlled experimental conditions. In these datasets, the experimental conditions and variables were already predetermined by the original study design. As such, our study did not involve any intervention or group assignment where randomization would be necessary. Instead, our focus was on analyzing the available data to observe specific neural and behavioral dynamics, which does not require further randomization of experimental groups. |
| Blinding | Blinding was not applicable to this study because the analysis was conducted on pre-existing datasets (synthetic data, macaque experimental data, and rat experimental data) that had already been collected under controlled conditions. The primary aim of the study was to analyze objective, quantitative data, such as neural and behavioral metrics, which are not subject to subjective interpretation or observer bias. Since the data analysis did not involve subjective assessments, treatment interventions, or human participants, there was no need to implement blinding procedures. |

# Reporting for specific materials, systems and methods

We require information from authors about some types of materials, experimental systems and methods used in many studies. Here, indicate whether each material, system or method listed is relevant to your study. If you are not sure if a list item applies to your research, read the appropriate section before selecting a response.

## Materials & experimental systems

| n/a | Involved in the study |
|---|---|
| ☒ | ☐ Antibodies |
| ☒ | ☐ Eukaryotic cell lines |
| ☒ | ☐ Palaeontology and archaeology |
| ☒ | ☐ Animals and other organisms |
| ☒ | ☐ Clinical data |
| ☒ | ☐ Dual use research of concern |

## Methods

| n/a | Involved in the study |
|---|---|
| ☒ | ☐ ChIP-seq |
| ☒ | ☐ Flow cytometry |
| ☒ | ☐ MRI-based neuroimaging |

