## [Peer Review File · Nature Methods]

Interpretable statistical representations of neural population dynamics and geometry

Corresponding Author: Dr Adam Gosztolai

A version of this paper was originally rejected for publication by Nature Methods, however that decision was reconsidered after appeal by the authors.

Version 0:

Decision Letter:

6th Oct 2023

Dear Adam,

Let me first sincerely apologize for the delays in the review process. While one of the reviews came in quite a while ago, the other two reviewers were very late, and one of them still has not provided any input. I am very sorry about this, especially since I don't have good news.

Your Article entitled "Interpretable statistical representations of neural population dynamics and geometry" has now been seen by two reviewers, whose comments are attached. In the light of their advice we have decided that we cannot offer to publish your manuscript in Nature Methods.

You will see that, while they find your work of some potential interest, the reviewers raise concerns about the advance your methodological approach represents over available methods and about its broad applicability at this stage. In addition to the concerns about the practical advance over established methods, the authors also question the comparisons to existing methods. We think that these criticisms are sufficiently important as to prevent publication of your work in Nature Methods.

Although we cannot publish your paper, it may be appropriate for another journal in the Nature Portfolio. If you wish to explore the journals and transfer your manuscript please use our manuscript transfer portal. You will not have to re-supply manuscript metadata and files, unless you wish to make modifications. For more information, please see our [manuscript transfer FAQ](http://www.nature.com/authors/author_resources/transfer_manuscripts.html?WT.mc_id=EMI_NPG_1511_AUTHORTRANSF&WT.ec_id=AUTHOR) page.

I am sorry that we cannot be more positive on this occasion but hope that you find the reviewers' comments helpful when preparing your paper for submission elsewhere. Again, I would like to apologize again for the unusual delays.

Best regards,
Nina

Nina Vogt, PhD
Senior Editor
Nature Methods

Reviewer Comments:

Reviewer #2 (Remarks to the Author):

In this manuscript, Gosztolai et al. introduce a novel method called MARBLE, which uses geometric deep learning to uncover latent dynamical structure in neural population activity. I found this work to be interesting and potentially useful for

neuroscientists working towards uncovering how neural populations might perform computations through dynamics. However, there are some aspects of the manuscript that I feel limit its utility for this audience.

Overall comments

My main concern is that this manuscript seems to be written in a way that targets people who already understand geometric deep learning, rather than neuroscientists who may have limited knowledge of the mathematical jargon. Primarily, it still seems unclear to me what MARBLE provides over other manifold identification or dynamics characterization techniques.

There's a paragraph in the introduction (lines 29-38) that attempts to outline where other methods fail, but despite reading it many times, I still don't quite understand the difference between previous dynamic models' "implicit dependence on the manifold geometry" (line 30-31--a negative aspect of previous work) and MARBLE representing "non-linear dynamics intrinsically on the manifold" (line 38 and 41--a positive aspect of the presented work). It seems that this difference might be key to what makes MARBLE special--if so, I believe it should be expanded upon, perhaps with a concrete example or a schematic figure showing the difference between a dynamic model depending on manifold geometry versus representing dynamics intrinsically on the manifold.

Likewise, I still don't quite understand the difference between geometry-aware and geometry-agnostic MARBLE embeddings, or when one should choose one over the other. In particular, I'm not sure what use geometry-aware embeddings have over other manifold characterization and dynamic model techniques. It does seem that geometry-agnostic embeddings allow for comparisons across many different instantiations of neural networks or neural populations performing the same task--if this works well, it is quite exciting, as it might provide another way to compare networks beyond geometric analysis like Canonical Correlation Analysis (CCA) and dynamic analysis like characterizing fixed point structure. This is especially poignant given previous results showing that artificial neural networks trained to perform behavioral tasks show a variety of geometric structures under CCA comparison, but a similar dynamic structure characterized by fixed points (see <https://doi.org/10.48550/arXiv.1907.08549>--I'm not sure this paper is cited in the manuscript, but if not, I believe it may be highly relevant). I think it would be good to see an expanded description of why each of these settings should be used and what they might differentially provide.

Other manuscript comments

Line 39: What does MARBLE stand for? I can't find a definition for it.

Line 44: It's not clear what this means--what is being compared here in contrastive learning? And between what is the latent space shared? It's true that these will be expanded upon later, but perhaps it would be good to give a description of the method with fewer unexplained pieces.

Line 75: What is subscript n ? I don't believe it was mentioned anywhere, but perhaps I missed it.

Line 78: What is a p-hop neighborhood? I Googled it, and it wasn't readily obvious what this means. Perhaps it would be better to give a less jargon-heavy description of the method in this section.

Line 79: If Z_c is a shared latent space across conditions, then what does subscript c refer to here? Is Z_c the latent activity of the given condition in a shared space? If so, it might be good to be explicit here.

Line 114: I don't believe the manuscript mentioned what the intrinsic and embedding dimensionalities were, which would be good to have. Also, this is trivial, but I believe this is a paraboloid, not a parabola--but I could be wrong.

Fig 2d: It's unclear what the different colors mean in this figure and how they relate to the colors in 2a.

Fig 2j is mislabeled (should be 2i, I think)

Line 138: Authors say previous methods only use static snapshots, but CCA matches neural trajectories across conditions--not point clouds. It has also been used to align neural activity (via dynamics of latent activity) across many days (<https://doi.org/10.1038/s41593-019-0555-4>) and even across individuals (<https://doi.org/10.1101/2022.09.26.509498>). However, in favor of the authors' point, the "universality" paper I cited earlier suggests different networks performing the same task may differ in CCA but not in fixed point structure. Similarly, this section has no comparison with any other methods of comparing RNN computations. Would those methods (e.g. CCA) show similar results? I would guess that CCA would show two separate types of computations here, like the "universality" paper.

Line 152: which two representative examples are shown? There are three panels here, so this is a little bit confusing.

Line 157: It's unclear why the authors would use PCA here first--is there an easy explanation for this?

Line 164: How do the authors determine whether there are subblocks present or if there are gradual changes in this distance matrix? It's not listed here, though the caption mentions hierarchical clustering. If hierarchical clustering, does one need to specify number of clusters to start with? And what does hierarchical clustering do with gradually changing distance matrices?

Line 173: An example from my previous point--the authors are switching to a geometry-agnostic MARBLE network, but it's unclear why.

Spike analysis section (lines 182-215): Unless I'm misunderstanding things, it seems a bit unfair to separate conditions for MARBLE network training first and then compare it to LFADS, which doesn't know anything about behavior except trial boundaries. This method seems more like MINT (<https://doi.org/10.1101/2023.04.05.535396>), where they build a library of neural dynamics with several conditions and then build a decoder off of that. It is impressive that MARBLE outperforms LFADS in neural decoding, but I would like to see how well it performs compared to MINT, which has a similar level of supervision.

Line 196: It seems unsurprising to find an embedding that reflects spatial geometry of reaches, especially since, as the authors note, neurons in PMd are sometimes directionally tuned. Even running PCA on motor cortex data sometimes reveals a similar spatial geometry.

Notes about codebase

The authors have helpfully made all the code and data easily available online. I found the code to be quite well documented (it also taught me what MARBLE stood for). Below are a few notes from my trying to get the code working on a couple different machines.

First, I tried running the code on my M2 Macbook Air, with limited success:

- The environment does not install out of the box easily on macOS (Macbook air M2). Had to strip package-specific version tags (as suggested by this Stack Overflow answer: <https://stackoverflow.com/a/56780297>). Even after that, there were several packages that needed to be replaced by pip binary packages because conda channels did not have arm64 versions of them. Specific changes:
 - Moved pytorch-* packages to pip as torch-*
 - Changed icu version to 67.1 as 58.2 is only available from win-32 architecture in conda-forge
 - There seem to be no mkl* packages that work for Apple silicon, so I removed them
 - Even after removing intel-based packages, environment did not resolve through conda. I resorted to installing packages one by one using pip in a conda environment for the specific notebooks I was trying to test.

After this, I tried running the code on my Windows computer through Windows Subsystem for Linux (WSL), which was far more successful:

- Environment sets up well on WSL out of the box, though I'm not sure it's actually using the GPU during training, despite CUDA and all being installed.
- Error when running `vanderpol.ipynb`: `MARBLE.construct_dataset(...)` doesn't run because `labels` isn't specified. It seems this code was changed in the last couple days but untested. I suggest not pushing code to the master branch until it is fully tested, or else create a git tag for a version that works to reproduce paper figures without errors. Reverting to the last commit that passed CI checks on GitHub resolved error
- Trying to run visualization notebook from spike analysis has errors because `ipympl`, `neo`, and `elephant` are not installed as part of environment. It would be good to go through all notebooks to make sure all the required modules are part of the environment.yml file.

References

Maheswaranathan, Niru, Williams, Alex H., Golub, Matthew D., Ganguli, Surya, Sussillo, David. Universality and individuality in neural dynamics across large populations of recurrent networks. arXiv. 2019-07-19. doi: 10.48550/arXiv.1907.08549

Gallego, Juan A., Perich, Matthew G., Chowdhury, Raaed H., Solla, Sara A., Miller, Lee E. Long-term stability of cortical population dynamics underlying consistent behavior. Nature Neuroscience. 2020-01. doi: 10.1038/s41593-019-0555-4

Safaie, Mostafa, Chang, Joanna C., Park, Junchol, Miller, Lee E., Dudman, Joshua T., Perich, Matthew G., Gallego, Juan Alvaro. Preserved neural population dynamics across animals performing similar behaviour. 2022-09. doi: 10.1101/2022.09.26.509498

Perkins, Sean M., Cunningham, John P., Wang, Qi, Churchland, Mark M. Simple decoding of behavior from a complicated neural manifold. bioRxiv. 2023-04. doi: 10.1101/2023.04.05.535396

Reviewer #4 (Remarks to the Author):

This paper introduces a novel dimensional reduction technique, called MARBLE, which focuses on local regions of the subspaces, and also utilizes an RNN to learn the latent vectors. The authors motivate this advance by discussing shortcomings of existing dimensionality reduction techniques in terms of generalization across subjects and poor fitting of complex geometries. The technique is then illustrated in one toy example and a delayed-match to sample task. It is then applied to a compiled dataset from a single macaque performing a center out reaching task.

This paper has serious flaws in its lack of comparison to other methods and the lack of a single compelling example of achieving a previously unseen level of performance for at least one task.

Specifically, at the first portion with a toy example, no other technique was introduced in comparison, and the capture of

complex rotational dynamics is not a new feature for these techniques. It would have been best to illustrate any specific advantages of this technique against others at this stage. Similarly, for the delayed match to sample, the authors simply report that a high amount of variance was explained, with no comparison to the state of the art.

Finally, the last section of the paper introduced a single neural dataset, which by itself does not address the motivated problem statement of generalization nor complex geometries. Success is claimed by showing that 7 trajectories emerge from the method. This is finally compared to a single technique (LFADS), whose performance does not appear analogous to its top performance in prior work, and which is not necessarily the only candidate for the existing state of the art. Many nonlinear techniques can reveal trajectories like Figure 3h from neural data of the type described, so it's not clear what is sufficiently novel here. Also, the general claims of this paper would tend to require testing with additional large-scale datasets, which are broadly available from multiple animal models.

Overall, I am not persuaded that this technique achieves best in class performance in revealing repeatable and generalizable dynamics, nor for revealing kinematic trajectories.

** For Nature Portfolio general information and news for authors, see <http://npg.nature.com/authors>.

Version 1:

Decision Letter:

22nd Dec 2023

Dear Dr Gosztolai,

Thank you for your letter asking us to reconsider our decision on your Article, "Interpretable statistical representations of neural population dynamics and geometry". After careful consideration we have decided that we are willing to consider a revised version of your manuscript.

Please make any additional changes as necessary and resubmit at the link provided below. At this point, I would also like to let you know that we may assign an additional reviewer.

- * include a point-by-point response to our referees and to any editorial suggestions
- * please underline/highlight any additions to the text or areas with other significant changes to facilitate review of the revised manuscript
- * address the points listed described below to conform to our open science requirements
- * ensure it complies with our general format requirements as set out in our guide to authors at www.nature.com/naturemethods
- * resubmit all the necessary files electronically by using the link below to access your home page

Link Redacted

We hope to receive your revised paper within four weeks. If you cannot send it within this time, please let us know. In this event, we will still be happy to reconsider your paper at a later date so long as nothing similar has been accepted for publication at Nature Methods or published elsewhere.

OPEN SCIENCE REQUIREMENTS

REPORTING SUMMARY AND EDITORIAL POLICY CHECKLISTS

When revising your manuscript, please submit reporting summary and editorial policy checklists.

DATA AVAILABILITY

CODE AVAILABILITY

Please include a "Code Availability" subsection in the Online Methods which details how your custom code is made available. Only in rare cases (where code is not central to the main conclusions of the paper) is the statement "available upon request" allowed (and reasons should be specified).

MATERIALS AVAILABILITY

ORCID

Nature Methods is committed to improving transparency in authorship. As part of our efforts in this direction, we are now requesting that all authors identified as 'corresponding author' on published papers create and link their Open Researcher and Contributor Identifier (ORCID) with their account on the Manuscript Tracking System (MTS), prior to acceptance. This applies to primary research papers only. ORCID helps the scientific community achieve unambiguous attribution of all scholarly contributions. You can create and link your ORCID from the home page of the MTS by clicking on 'Modify my Springer Nature account'. For more information please visit <http://www.springernature.com/orcid>.

Best regards,
Nina

Nina Vogt, PhD
Senior Editor
Nature Methods

Version 2:

Decision Letter:

2nd Apr 2024

Dear Dr Gosztolai,

Thank you for your patience. Your Article, "Interpretable statistical representations of neural population dynamics and geometry", has now been seen by two reviewers, one of the original ones and a new reviewer. As you will see from their comments below, although the reviewers find your work of considerable potential interest, they continue to raise a number of concerns. We are interested in the possibility of publishing your paper in Nature Methods, but would like to consider your response to these concerns before we reach a final decision on publication.

We therefore invite you to revise your manuscript to address these concerns. Importantly, the reviewers (while experts in the field) continue to find that the manuscript is not written in an accessible manner. Please make sure to address this as Nature Methods is read by a diverse audience. We ask that you also address the other concerns brought up by the reviewers.

Link Redacted

We hope to receive your revised paper within 2-3 months. If you cannot send it within this time, please let us know. In this event, we will still be happy to reconsider your paper at a later date so long as nothing similar has been accepted for publication at Nature Methods or published elsewhere.

OPEN SCIENCE REQUIREMENTS

REPORTING SUMMARY AND EDITORIAL POLICY CHECKLISTS

DATA AVAILABILITY

All novel DNA and RNA sequencing data, protein sequences, genetic polymorphisms, linked genotype and phenotype data, gene expression data, macromolecular structures, and proteomics data must be deposited in a publicly accessible database, and accession codes and associated hyperlinks must be provided in the "Data Availability" section.

CODE AVAILABILITY

Please include a "Code Availability" subsection in the Online Methods which details how your custom code is made available. Only in rare cases (where code is not central to the main conclusions of the paper) is the statement "available upon request" allowed (and reasons should be specified).

MATERIALS AVAILABILITY

ORCID

Best regards,
Nina

Nina Vogt, PhD
Senior Editor
Nature Methods

Reviewers' Comments:

Reviewer #2:

Remarks to the Author:

In this revision to their previously submitted manuscript introducing MARBLE, Gosztolai et al. have provided additional controls and datasets, as well as tried to make the description of the method clearer. In this, I believe they've largely succeeded--I find the manuscript to indeed be clearer than it was when I first read it. I'm also reasonably impressed with the new rat hippocampal data analysis. However, I still have some important concerns that I believe should be addressed.

Acknowledgement of previous results

After reading the revised manuscript, I believe I'm starting to understand something I didn't the first time around--this method finds a latent space representation of not just neural state but also empirical estimations of local flow fields. This is in contrast with most other latent factor analysis methods being used in neuroscience, which find a latent representation of only the neural state--this seems to be what the authors are calling "point cloud" methods.

First of all, it was the new clarity of the manuscript got me to this understanding, so assuming it is correct, the authors have done a better job getting their ideas across in this revision. However, it did take me some time to fully grasp the intention of this method, and I think part of the reason was how the authors were characterizing and treating computational modeling methods previously used in neuroscience, including so-called "point cloud" methods and dynamical systems models.

In particular, while reading the manuscript, I couldn't help but feel that the authors may be unaware of how previous methods have been used, and as such, it seemed like previous methods were being treated unfairly. For example, the authors showed that MARBLE applied to monkey center-out reaching data "reflect the spatial geometry of the physical reaches", and noted that this arrangement wasn't readily apparent in CEBRA, LFADS, PCA, t-SNE, or UMAP. However, this seems to ignore a number of neural population studies that have found this spatial geometry with simpler, linear methods. For a recent example, see Sun et al. 2022, which uses Targeted Dimensionality Reduction (TDR) to find a linear projection of neural population activity that separates out different reach directions. While it is true that such previous methods require more supervision than MARBLE does and that MARBLE might separate out the clusters better, the fact that this kind of spatial geometry exists in motor cortical activity is not surprising. In fact, this geometry is one of the main reasons that simple linear neural decoders function for most of the brain-machine interface work that has happened over the last decade.

Another example of this is the perspective the authors take on comparing neural population activity between multiple datasets. The authors contend that dynamical models "depend on the particular embedding of trajectories in state space and, in turn, the measured variables (i.e., neurons), hindering their ability to compare representations across experiments" (Page 2). They also note that "While it is possible to 'stitch' multiple datasets by fitting auxiliary transformations or aligning the neural dynamics to behaviour, this relies on the assumption that the respective neurons encode the same computation" (Page 8). These claims seem to discount previous work to the contrary.

First, dynamical models don't necessarily have to depend on the particular embedding, as exemplified by the original LFADS work--at the end of that paper, the authors showed that one can use the same LFADS dynamical model across multiple sessions, simply by changing the read-in and read-out weights from the model. Gosztolai et al. actually reference this ability to "stitch" in the second quote above, but then mention that it requires that neurons encode the same computation. This is simply not true--in theory, two neurons could have completely swapped roles, and the read-in and read-out weight would only have to swap to adjust accordingly.

Second, these statements seem to discount work with linear methods that don't even require the same neurons to be recorded to align neural trajectories across sessions. Gallego et al. 2020 shows that activity on the neural manifold during a particular behavior is consistent enough to be linearly aligned and decoded from with a single decoder over a period of up to two years--a period during which the neurons being recorded from are certainly not the same. Similarly, Safaie et al. 2023 (cited by this manuscript) showed the same thing but across monkeys, which clearly have different neurons.

It is likely the authors did not mean to discount such previous relevant work, but the way parts of the manuscript is written makes it come off this way. I propose that the authors rework some of these parts to better highlight how MARBLE adds to previous work.

Canonical correlation analysis

In my previous review, I mentioned that Canonical Correlation Analysis (CCA) would be provide a good comparison for some of the results, and I still believe that is the case. However, I'm thoroughly confused by how the authors have described and used CCA here. I'm concerned that there has been a critical misunderstanding, either in my reading of the manuscript and

rebuttal to my previous review, or in the authors' use of CCA.

In the rebuttal, the authors say:

"CCA compares two datasets by decomposing them into linear subspaces using PCA (i.e., SVD) and measures the correlation between the respective principal axes. Because each dataset is considered a point cloud, dynamical information is lost, and the correlation value is only caused by similarities/differences in sampling."

It is possible I am misinterpreting this, but as it reads, this description of CCA is wrong. CCA does not measure correlation between principal axes calculated through PCA--this would just be correlation. CCA finds a linear transformation for each dataset such that in the transformed, so-called "canonical" coordinates, the correlation is maximized. Importantly, the two datasets must be aligned point-to-point in some way for this to make sense. Thus, CCA is not just an analysis of point clouds--the order of the points in each dataset matters. In this way, CCA-based alignment and analysis have some measure of dynamical information beyond the simple point cloud distribution, even if it is not as explicit as MARBLE's embedding of local flow fields.

In the case of neural data, CCA alignment is usually done by averaging trajectories in neural state space over trials with the same behavioral conditions in each dataset. Then, finding the transformations that best aligns the neural trajectories allows one to project individual trials into the "canonical" space. As mentioned above and in my previous review, this type of analysis has allowed previous works to find consistent neural trajectory shapes in the same subjects over years (Gallego et al. 2020), and across different subjects (Safie et al 2023).

Later in the rebuttal, the authors state that CCA "will only pick up geometric differences between neural activations, which does not necessarily correspond to dynamical variation", citing the Maheswaranathan et al. 2019 paper. I'm not sure I understand this interpretation of that paper--the authors' reasoning here suggests that somehow CCA will not detect differences in dynamics, whereas the Maheswaranathan paper shows that CCA is more sensitive to changes in the specific implementation an RNN happens to use for a given task, while the fixed point analysis (an analysis of dynamical scaffolding) shows similarities across RNNs performing the same task. Thus, I would say that CCA would be more sensitive to changes in dynamics.

With that background, I am quite confused by the CCA results for the low-rank RNN analysis in the manuscript. The authors quite nicely show that there are two solutions for this one task that result in very different neural trajectories (similarly to what is shown in the Maheswaranathan paper). In a test of comparing two different iterations of Solution I and one iteration of Solution II, MARBLE nicely categorizes the two Solution I networks as more similar to each other than they are to the Solution II network. However, the CCA results are complete noise, despite the flow fields for the two Solution I networks being remarkably similar but oriented differently in the neural space. I'm left wondering why CCA couldn't align these two networks--is it because the authors' understanding of CCA is flawed (as evidenced by their description of CCA in the rebuttal)? I imagine part of it is because the authors used a random smattering of initial conditions that didn't actually align well across datasets, but I want some more information about this, and I couldn't find a good description in the methods. As it is, I'm concerned that this control doesn't make sense to me.

"Geometry"-aware vs. -agnostic

One of the things I found most confusing about the original manuscript was the distinction the authors made between the "geometry-aware" and "geometry-agnostic" modes of MARBLE. In this revision, I find that this problem is mostly taken care of, thanks to some rewriting and the addition of Figure 1d, which shows the differences between MARBLE results on simple constructed flow fields.

However, while I believe I now understand the difference between the two modes, I don't agree with the naming of it--I believe that is the piece that caused me the most confusion previously. What the authors start to describe as "geometry" for these purposes is revealed to be mainly the orientation of the local flow fields (page 4). Intuitively, this is not what I would call geometry--it seems to me to be more about how the flow field is embedded in the neural space. The authors even go on to say a few sentences later that the "geometry-agnostic mode is able to discount the arbitrary rotations in the LFFs induced by different embeddings" (page 4). Later on, the authors write "geometry-agnostic MARBLE can extract both abrupt and continuous dynamical variation while being invariant to manifold embedding at a slight loss of expressivity compared to the geometry-aware mode." Further, to justify a "geometry-agnostic" MARBLE in the low-rank RNN analysis, the authors write "Due to the arbitrary embedding of the manifolds across networks, we used geometry-agnostic MARBLE" (page 6). As such, it seems to me that the two modes here are really "embedding-aware" and "embedding-agnostic", and based on how the authors write about the two modes, I think they might agree with me.

Secondly, and related to my "previous work" and "CCA" comments above, the Gallego et al. 2020 paper using CCA to align neural trajectories across many sessions in the same monkey has a careful description of the hypothesis that neural activity on different days is simply a different embedding of the true underlying neural manifold. That paper follows that idea to build an embedding-agnostic decoding procedure using CCA. Later work from that group and others continue to use that terminology. I think this manuscript about MARBLE might do well to continue using the same terminology so as to keep the field's nomenclature consistent and to avoid alienating readers who might already be familiar with previous work. I realize that this may clash with the terminology used for the latent representation found by MARBLE, which the authors are also calling an embedding, but maybe the authors can separate these terms by only referring to the MARBLE embedding as a latent representation. I believe this would have alleviated much of my confusion.

Definition of "unsupervised" learning

As before, in the analysis of the monkey reaching dataset, the authors run MARBLE independently on each reach condition--this, I mentioned in my previous review, seems like it would provide an unfair advantage over LFADS, as this procedure provides some supervision during training. The authors mention in the rebuttal that this is because they don't want to assume consistent dynamics across different conditions (presumably to separate the local flow field computations). While that may be a good reason to train the model this way, it cannot be truly called "unsupervised" as the authors seem to be claiming.

Consider, for example, Linear Discriminant Analysis (LDA). During training, LDA would use only the condition labels (reach direction), but during testing, LDA simply embeds the neural state into the latent space for classification (or whatever else the user would like to do). The knowledge of condition labels at each step strikes me as identical to MARBLE's knowledge in this example, and yet I would consider LDA to be a supervised method, while the authors call this version of MARBLE "unsupervised". On the other hand, LFADS takes no information about conditions, and therefore assumes a single dynamical system over the whole session--to me, this is far more unsupervised. It may be that the authors consider the condition-splitting akin to something like self-supervision, or minimal supervision, but I find it quite misleading to call it "unsupervised".

As I mentioned before, MINT might be a better comparison with this method of analysis, but as the authors were unable to get the package to work, I think there are a couple alternative options here, both of which would be nice to see:

- 1) The authors could add a comparison with LDA or TDR (see Sun et al. 2022 for an example use), both of which seem like they may have a similar level of supervision to MARBLE. I have no doubt that MARBLE will exceed the capabilities of LDA or TDR, since LDA and TDR are relatively simple linear models, but it's sometimes useful to use a linear baseline. Furthermore, from previous work with linear methods (see again Sun et al. 2022 for an example), I believe that LDA and TDR should uncover some clustering of the seven reach conditions with just a couple latent factors.
- 2) The authors could run MARBLE unsupervised and completely blind to conditions, i.e., without splitting the trials by condition and finding a single manifold over all conditions. This seems to me to provide a fairer comparison with LFADS, which doesn't know about reach direction during training. I'm interested to see how getting rid of the condition labels might change the latent representations.

Minor comments

Fig 1b - I still find this flowchart confusing. What is the input to the diffusion box? And what is being represented by the graph with the overlaid flow field? Is this a separate input, or is this a representation of what the sample nodes are?

Fig 1d - As I said in the main comments, I greatly appreciate this addition. I have three suggestions/questions here: 1) it's hard to see what's behind the flow fields in the figure--perhaps it would help to make them more sparse? 2) What is each point in the latent space? A time point? A single trajectory? A local flow field? It would be good to make this clear early in the manuscript. 3) Since the agnostic version of MARBLE cares about expansive and contractive fields, it would have been nice to have a third set of flows comparing those and showing that they wouldn't overlap in the MARBLE latent representation.

Fig 1g (and others with MDS representations) - Is there a reason not to show the z-space directly? Or do you have to reduce dimensionality of the MARBLE latent representation for any of it to make sense? Might be good to specify this.

Page 6, bottom - "This also suggests that previous methods that used CCA to detect changes across RNNs13 or animals4,48 could have detected the compound effect of changing manifold curvature and different sampling of state space, without concluding dynamical differences." This is related to the CCA comments above, but my interpretation of all of the papers cited does not match this statement. To me, these papers were all using CCA to look for similarities (and found striking similarities, in the cases of citations 4 and 48, as well as the uncited Gallego et al. 2020 paper). I will concede, however that manifold curvature may indeed cause poorer CCA alignment though, as CCA is just a linear method.

Page 7, bottom - "Since a subset of neurons in PMd is directionally tuned...". The authors use directional tuning as a reason that neural representations would be sensitive to orientation on the manifold, but it seems much simpler (or at least less of a logical leap) to lean on previous findings that there's generally a geometric organization of neural states in population space that corresponds with reach direction.

Page 10, top - "we showed that temporal ordering naturally emerges from our similarity-preserving embeddings of local vector fields." I'm not sure I saw this explicitly mentioned before the discussion--what does this refer to? The monkey reach decoding? Rat hippocampus? Both?

Page 10, top - "This suggests a hypothesis that the neural readout into behaviour in biological brains might rely on the context of neural trajectories in a broader dynamical landscape." This statement seems vague, out of scope, and probably unnecessary. While this is an intriguing statement, I don't think anything in this manuscript really goes to support this or show what this really means.

Extra References

Gallego, Juan A., Perich, Matthew G., Chowdhury, Raed H., Solla, Sara A., Miller, Lee E. Long-term stability of cortical population dynamics underlying consistent behavior. *Nature Neuroscience*. 2020-01. doi: 10.1038/s41593-019-0555-4

Sun, Xulu, O'Shea, Daniel J., Golub, Matthew D., Trautmann, Eric M., Vyas, Saurabh, Ryu, Stephen I., Shenoy, Krishna V. Cortical preparatory activity indexes learned motor memories. *Nature* (2022)

None

Reviewer #5:

Remarks to the Author:

Gosztolai et al develop MARBLE, an unsupervised, data-driven approach for uncovering non-linear dynamics. MARBLE models non-linear dynamical systems on a manifold embedded within a high-dimensional space. It does this using geometric deep learning to create an embedding of local flow fields.

On the positive side, I found this work to be highly innovative. The model is a substantial departure from current approaches to infer dynamics from neuronal populations. Two particular innovations stood out: 1) the use of a proximity graph to approximate a manifold is to my knowledge new in the population dynamics space, and 2) the use of a graph convolutional network to uncover dynamics within the manifold, and to find an embedding set to describe those dynamics. The introduction of these methods to the set of approaches that attempt to model population dynamics is quite exciting. And the ability to compare embeddings across dynamical systems is very interesting.

However, as presented, there were several issues that made it less compelling and at present I do not feel the work is suitable for a broad readership journal like Nature Methods. Several major issues stood out:

1. While demonstrations on synthetic datasets were interesting, real data depart from idealized and synthetic data sets in many ways, particularly in having a lot of high-dimensional variability (spiking variability) that is unrelated to variables like kinematics, etc. Thus, demonstrations on real data are critical for proving the method's utility. However, the current demonstrations on real data were not compelling and did not seem rigorous.

Particularly problematic was the demonstration on macaque reaching data. Buried in the methods (and highlighted by one of the reviewers) was a note that each condition in this 7-target dataset was considered a different, separately-learned manifold. Thus the method received some knowledge of trial identity when learning the underlying manifold structure. This undermines the claims that the current method is unsupervised. Further, because the key result of this section is the accuracy in decoding reaching movements, this result seems trivial to achieve when the underlying representations are grouped by condition - most of the decoding accuracy can be achieved just by separating conditions and decoding the trial-averaged kinematic trajectory. Similarly, the visual separation of the trajectories may be a trivial result of the different conditions being learned on separate manifolds. (Related, it is unclear from the methods how the separate manifolds are being combined?) Though I do not believe the authors are being intentionally misleading, calling the application "unsupervised" is misleading and will cause confusion.

(Scientifically, the idea that different reaches are on separate manifolds and obey separate dynamics departs from current literature on manifolds underlying reaching movements, which I detail below.)

Related, there were particular choices made in the methods - heavy smoothing, PCA dimensionality choices, etc, that are presumably hand-tuned. It is unclear how robust the method is to different parameter/hyperparameter choices. For deep learning methods, demonstrating such robustness is crucial for a method to have broad applicability.

2. A higher level issue is the repeated claim in this manuscript that the method leads to interpretable representations, without any definition of what interpretability is. It is hard to evaluate any claim of a method being "more interpretable" without knowing what is meant by interpretability. Thus the notes e.g. that one method or the other "lacked interpretability" are not possible to substantiate. In some cases the authors seem to be conflating interpretability with decodability, but for neuroscientific applications, it is unclear why representations that lead to better decoding should be more "interpretable"- for example, because variability in neural population activity does not simply reflect representation of external variables. To even evaluate a claim of interpretability, the authors should be very clear from the outset what they mean.

3. I strongly agree with the previous reviewer that the manuscript as written – even with the inclusion of toy examples – is extremely difficult to follow for readers who are not experts in geometric deep learning. Even after reviewing the methods in detail, and having some familiarity with graph neural networks, it was still difficult to understand exactly what the authors were doing. To have this manuscript published in a broad readership journal like Nature Methods, I would strongly recommend that the manuscript be written in a way that breaks down the technical aspects of the method in a much more approachable fashion. Clearly outlining the different steps of the method and example outputs of different stages on real data would be critical. Particularly: 1) more clearly demonstrating how the nearest neighbors algorithm is used to create a manifold, 2) explicitly delineating how tangent spaces and moving between them works, 3) explaining how gradient filters are used and example outputs, 4) explaining how the method is sensitive to real world noise, and hyperparameter choices. Perhaps the authors might seek more detailed feedback from readers without geometric deep learning expertise to make the manuscript more approachable.

4. For a method that centers on modeling and extraction of latent dynamics – and may be quite innovative in this regard – noticeably little effort was spent trying to characterize or interpret the dynamics of real neural data (i.e., the flow fields themselves). There was little evidence presented that MARBLE is successfully modeling the dynamics of real neural data. E.g., applications to datasets with many conditions, and showing generalization of the learned model to unseen conditions, would be a powerful demonstration of the model's ability to learn dynamics on real data.

5. It appears that the manifold learning step does not have any sort of denoising process. Thus for real data applications which contain noise, the authors do a lot of (hand-tuned?) preprocessing of the data. This is in stark contrast to e.g. the LFADS point of comparison, which operates on the raw binned spike counts. Because noise in real neural data is a huge issue (and thus many methods resort to averaging across conditions or using a supervisory signal), it would be critical for the authors to

explore how sensitive MARBLE is to different choices of preprocessing when estimating the manifold. It seems like the current approach, with heavy smoothing, might destroy any high-frequency structure in the data.

Medium:

- For real data, the model chose as one point of comparison the multi-session macaque reaching dataset from the original LFADS paper. In the LFADS paper, the point of this dataset was that each session had very few channels, and thus it could highlight the ability to “stitch” together several sessions of separately-recorded neural data. Since the authors are not using it for that purpose, there are several other reaching datasets from the Shenoy lab [1,2] that would be much better for demonstrating manifold and dynamics discovery. If the authors are specifically trying to make a point about the effectiveness of MARBLE for limited-channel-count data, that should be explained more thoroughly - otherwise the current dataset is of questionable relevance for the author’s demonstration.

- Kudos to the authors for spending a lot of effort on the codebase, with examples and detailed installation instructions for multiple platforms. Similar to another reviewer, I had trouble installing the code on my M2 MacBook Pro. I did have success on a Mac Studio and on Linux. Similar to the other reviewer, I found some notebooks difficult to run because they had dependencies outside of those in the installation instructions.

- In a response to another reviewer regarding the issue of learning separate manifolds on the macaque reaching dataset, the authors say:

“We agree with the reviewer that MARBLE considers the trials under a given task condition to belong to an independent manifold. This is necessary because only under a given condition do we have good reason to believe that the dynamics are consistent.”

Perhaps there is a misunderstanding of the literature. For example, in the Churchland 2012 paper [3], the key point of the manuscript was that a consistent set of dynamics (rotational) describes a wide variety of conditions (100+ reaching conditions in some datasets), and all these conditions were present on the same manifold (estimated using smoothing and PCA). Similarly, in the LFADS paper [4] that the authors compare to, the method was used to learn a single manifold and set of dynamics across all conditions.

- In the methods there was a note about the optimal transport computation comparing conditions. Could this be expanded a bit? It was unclear whether this was still unsupervised or not.

- Other methods often have a recognition network that would allow inference to be performed on held out or unseen data. How is that done in the current case? How is cross-validation performed? Specifics and limitations in this regard should be clearly spelled out in the manuscript.

- It appears that the current formulation of the model relies on completely autonomous dynamics - i.e., all the temporal evolution of the data must be completely described by the learned vector field. While this may work well for carefully aligned data, it is unclear how this accommodates data with non-autonomous dynamics, i.e. where unexpected events occur. The authors should include some discussion of this issue.

- In the multi-session macaque reaching dataset, presumably the dynamics across all sessions should be the same. Demonstrating that MARBLE uncovers consistent dynamics (embeddings) across sessions would be compelling. It was unclear if/how this is shown in Fig. S9. (Also, this analysis would need to be done after addressing the fact that MARBLE’s application to this dataset is not truly unsupervised, though that is what is claimed in the manuscript.)

[1] <https://dandiarchive.org/dandiset/000070>

[2] <https://npsl.sites.stanford.edu/research/data-code>

[3] <https://pubmed.ncbi.nlm.nih.gov/22722855/>

[4] <https://pubmed.ncbi.nlm.nih.gov/30224673/>

Minor notes:

- Fig. 2b refers to “stimulation” patterns. In neuroscience “stimulation” often refers to e.g. electrical or optogenetic perturbation of a circuit. Consider using “stimulus” instead (paralleling Fig. 2a).

Version 3:

Decision Letter:

Our ref: NMETH-A52445C

30th Aug 2024

Dear Dr. Gosztolai,

Thank you for your patience. I am sorry for the delays. I was waiting for input from one of the reviewers but unfortunately did not hear from them.

Thank you for submitting your revised manuscript "Interpretable statistical representations of neural population dynamics and

geometry" (NMETH-A52445C). It has now been seen by one of the original referees and their comments are below. The reviewer finds that the paper has improved in revision, and therefore we'll be happy in principle to publish it in Nature Methods, pending minor revisions to satisfy the referees' final requests and to comply with our editorial and formatting guidelines.

Please do make sure to add a clarification about the unsupervised/minimally supervised nature of your approach early on.

TRANSPARENT PEER REVIEW

ORCID

Best regards,
Nina

Nina Vogt, PhD
Senior Editor
Nature Methods

Reviewer #2 (Remarks to the Author):

Gosztolai et al. present in this manuscript MARBLE, a novel computational method to analyze the low-dimensional, nonlinear dynamics of neural population activity. In this revision, the authors have added additional analyses to improve the rigor of the work, and they have added text to clarify misunderstandings from previous review cycles.

Overall, I am happy with this revision. The authors have addressed most, if not all of my concerns with their edits. Chiefly, the new CCA results make sense to me, and I find it intriguing that while CCA shows that dynamically similar RNN solutions are more alignable than dynamically different ones, they don't show the nuances that MARBLE does. This strikes me as an important result, and as impressed as I was with the MARBLE results before, the context brought by the CCA results give it some more impact.

Also, I think I now understand the authors' reasoning for calling this method "unsupervised". I agree with the authors that MARBLE is far less supervised than the methods that I mentioned in my previous review, and the authors agree with me that MARBLE can sometimes require more user input than a fully unsupervised algorithm, like LFADS. The paragraph in the discussion goes a long way towards making this clear, and I am happy with the change.

However, as I read the manuscript, I remained skeptical about the "unsupervised" claim. I'm not sure if this is from previous bias or from the idea not fully coming across until the discussion. For example, on the bottom of page 8, the authors wrote: "We asked whether this structure could emerge unsupervised from local dynamical features in MARBLE representations. We constructed a separate manifold from the firing rates for each reach condition where we expected the dynamics to be consistent..." This pair of sentences, back-to-back struck me as incredibly strange--the authors mentioned a feature that might "emerge unsupervised" but then mention constructing separate manifolds for each reach condition. While I now understand the reasoning, this seemingly inconsistent logical turnaround left a bad taste.

However, I wonder if the authors would agree that the level of supervision MARBLE uses is similar to that of running PCA on

neural trajectories averaged over each condition. If the only thing MARBLE uses the condition-splitting for is to estimate local flow fields to eventually embed together, then it seems to me that this is really just a trial aggregation pre-processing step, like trial averaging, but much more sophisticated. If the authors agree with this, then I think this provides a concrete comparison for readers familiar with more well-known methods--trial-averaged PCA is reasonably standard for neural state analysis. I think most people would not call trial-averaged PCA completely unsupervised, but they would not call it "supervised" either, and it may be a good way to situate MARBLE for the monkey dataset use case. If the authors can use this comparison, or one like it, this may situate the "unsupervisedness" of MARBLE more clearly for readers.

Raeed Chowdhury

Minor comments

Page 3, bottom: it's not stated anywhere what ρ is here--I believe it's the number of gradients taken for the flow field estimation, but I can't be sure (I may have missed it in the text, but if not, it should be clarified somewhere in the main body)

Page 4, top: "invariant different" I think there's a typo here

Page 4, bottom: Figure 2h-j does not exist. Also I couldn't quite tell what examples the authors refer to when they say "In both examples, note that...".

Page 6, figure 3b: typo in the figure ("vin")

Page 7, top: It's unclear to me how figure 3e shows the different fixed point landscapes of the two RNN solutions. Maybe a typo?

Page 7, bottom: I suggest removing the words "best-fit" from the description of CCA, since it could be confusing exactly what this means.

Page 18, S1.8 heading: change from "geometry invariance" to "embedding invariance"

Fig S1 and S6: captions missing for some panels. Overall, double check the figures, captions, and references to make sure they're correct and all there.

Reviewer #2 (Remarks on code availability):

I have only briefly glanced at the code, but I have ensured that at least the Van Der Pol oscillator notebook runs on my computer (MacBook Air M2). The code seems well designed, and the documentation online makes it easy to install and get set up.

My only comment on this is that I wonder how difficult it would be to get MARBLE to use Metal acceleration on M-series Macs, instead of using the CPU. I realize that the authors' main focus is on NVIDIA GPU support (as it should be), but given that at least PyTorch can use Metal acceleration, I wonder how difficult it would be to do this and how much of a speedup it would confer. That said, I'm not as well versed in PyTorch as the authors, so I will leave it up to their discretion.

Version 4:

Decision Letter:

26th Nov 2024

Dear Adam,

I am pleased to inform you that your Article, "MARBLE: Interpretable representations of neural population dynamics using geometric deep learning", has now been accepted for publication in *Nature Methods*. The received and accepted dates will be May 1st, 2023 and November 26th, 2024. This note is intended to let you know what to expect from us over the next month or so, and to let you know where to address any further questions.

Over the next few weeks, your paper will be copyedited to ensure that it conforms to *Nature Methods* style. Once your paper is typeset, you will receive an email with a link to choose the appropriate publishing options for your paper and our Author Services team will be in touch regarding any additional information that may be required. It is extremely important that you let us know now whether you will be difficult to contact over the next month. If this is the case, we ask that you send us the contact information (email, phone and fax) of someone who will be able to check the proofs and deal with any last-minute problems.

Please note that *Nature Methods* is a Transformative Journal (TJ). Authors may publish their research with us through the traditional subscription access route or make their paper immediately open access through payment of an article-processing charge (APC). Authors will not be required to make a final decision about access to their article until it has been accepted. ](https://www.springernature.com/gp/open-research/transformative-journals) Find out more about Transformative Journals

Best regards,
Nina

Nina Vogt, PhD
Senior Editor
Nature Methods

** Visit the Springer Nature Editorial and Publishing website at http://editorial-jobs.springernature.com?utm_source=ejP_NMeth_email&utm_medium=ejP_NMeth_email&utm_campaign=ejp_Nmeth for more information about our career opportunities. If you have any questions please click [here](mailto:editorial.publishing.jobs@springernature.com).**

Open Access This Peer Review File is licensed under a Creative Commons Attribution 4.0 International License, which permits use, sharing, adaptation, distribution and reproduction in any medium or format, as long as you give appropriate credit to the original author(s) and the source, provide a link to the Creative Commons license, and indicate if changes were made. In cases where reviewers are anonymous, credit should be given to 'Anonymous Referee' and the source. The images or other third party material in this Peer Review File are included in the article's Creative Commons license, unless indicated otherwise in a credit line to the material. If material is not included in the article's Creative Commons license and your intended use is not permitted by statutory regulation or exceeds the permitted use, you will need to obtain permission directly from the copyright holder.

Dear Reviewers,

Thank you for your time evaluating our work.

Below we provide detailed responses to your concerns with references to changes made in the paper. We also attached a diff.pdf file highlighting the changes.

Kind regards,

The Authors

Reviewer #2 (Remarks to the Author):

In this manuscript, Gosztolai et al. introduce a novel method called MARBLE, which uses geometric deep learning to uncover latent dynamical structure in neural population activity. I found this work to be interesting and potentially useful for neuroscientists working towards uncovering how neural populations might perform computations through dynamics. However, there are some aspects of the manuscript that I feel limit its utility for this audience.

We thank the reviewer for the positive evaluation of our work. Indeed, our primary audience is neuroscientists interested in the dynamical basis of neural computation. However, we also aim to appeal to computer scientists who study the dynamical processes underpinning computation in artificial neural networks such as RNNs or the similarity between biological and artificial computations.

Overall, we found the criticisms of the Reviewer very helpful. Taking on board the comments below, we have made significant improvements in the *exposition of our method* and the *comparisons against other methods*. We have also included a *new case study of hippocampal recordings in rats* (new animal system and brain region) to showcase the cross-animal consistency of MARBLE embeddings.

Overall comments

My main concern is that this manuscript seems to be written in a way that targets people who already understand geometric deep learning, rather than neuroscientists who may have limited knowledge of the mathematical jargon. Primarily, it still seems unclear to me what MARBLE provides over other manifold identification or dynamics characterization techniques.

We thank the reviewer for this comment highlighting the lack of clarity in the description of our method. We have added explanations in the Introduction and in the examples to clarify further the distinctiveness and capabilities of our method.

The specific advantages of MARBLE over existing methods are:

- 1) **Unsupervised representation-learning of neural dynamics.** This is a significant advantage in scientific discovery over previous supervised methods such as CEBRA that require user-defined labels (e.g., behavioural labels), which have to be provided by the user and are not available in many cases and can introduce bias in the analysis. Importantly, despite being fully unsupervised, MARBLE exhibits better interpretability and decoding performance than supervised methods in most cases we studied (Figs. 3-4).
- 2) **Unbiased comparison of dynamical processes across systems** (networks and animals). We show that we open new avenues in comparing computational strategies across systems (Fig. 2) and discovering consistent embeddings across animals (Fig. 4), which have significant theoretical and experimental implications, e.g., in brain-machine interface research. This is unlike dynamical systems models (LFADS, SLDS), which depend on the particular embedding of the trajectories (which change from experiment to experiment) and, in turn, the identity of measured neurons. This is made possible by MARBLE because it learns dynamics over the underlying manifold, which does not depend on the embedding.
- 3) **Representation of the dynamics over the manifold.** A key theoretical advance in MARBLE is to represent dynamics as a collection of local flow fields (LFFs) rather than a collection of points, as done by many manifold learning methods such as PCA, t-SNE, UMAP, etc. In doing so, these methods can only capture densities of events but cannot represent the dynamics of the system, and thus give significantly less interpretable representations (Fig. S8).

We have clarified this carefully in the Introduction and through specific comparisons of MARBLE to all of these methods.

There's a paragraph in the introduction (lines 29-38) that attempts to outline where other methods fail, but despite reading it many times, I still don't quite understand the difference between previous dynamic models' "implicit dependence on the manifold geometry" (line 30-31--a negative aspect of previous work) and MARBLE representing "non-linear dynamics intrinsically on the manifold" (line 38 and 41--a positive aspect of the presented work).

Thank you again for highlighting the lack of clarity here.

MARBLE depends on two key ideas:

- 1) Instead of learning the temporal information in single trajectories (trials), as in dynamical models, we focus on the information in ensembles of trajectories. This information is richer because it contains information about how the activities change when we perturb the system.
- 2) We decompose the manifold surface into local flow fields (LFFs) so that we can represent them *intrinsically*, i.e., depending only on local distances, so that we can finally map them to a shared latent space by learning a similarity-preserving mapping. This similarity-preserving property is important because this means that, distributionally, the local LFFs learn global information.

These innovations are necessary to allow advances 1-3) in the comment above. We have further clarified this in the updated text and have removed jargonistic terms.

It seems that this difference might be key to what makes MARBLE special--if so, I believe it should be expanded upon, perhaps with a concrete example or a schematic figure showing the difference between a dynamic model depending on manifold geometry versus representing dynamics intrinsically on the manifold.

Thank you for helping us to more clearly articulate the unique representations of MARBLE. We have now added Fig. 1d to provide a simple illustration of how MARBLE represents features. We have also reorganised our non-linear dynamical systems example (Fig. 1e,f), which illustrates how MARBLE can be stable under changes of manifold geometry.

These initial examples are used to build intuition on simple setups that can be more easily understood. Of course, the advantages of MARBLE are fully illustrated by our complex neuroscience use cases. Specifically, our approach makes it possible to analyse those complex data sets by uniquely enabling:

- a) Comparison of dynamics over deforming manifolds (Fig. 1e,f),
- b) Comparison of dynamics across differently initialised RNNs (Fig. 2h-j),
- c) Finding consistent embeddings across animals without auxiliary transformations (Fig. 4e).

The other examples in our paper, which deal with interpreting and decoding the dynamics in a single animal, can be analysed with other methods, and we use these examples to demonstrate that the ensemble viewpoint enabled by MARBLE is richer and achieves higher accuracy in decoding and interpretability than CEBRA or LFADS, even if MARBLE is fully unsupervised and does not use any user-provided labels for training.

Likewise, I still don't quite understand the difference between geometry-aware and geometry-agnostic MARBLE embeddings, or when one should choose one over the other.

We thank the reviewer for highlighting the important distinction between the 'Geometry-aware' and 'Geometry-agnostic' modes of MARBLE. These are two modes in which MARBLE can be used seamlessly, and they provide distinctive and complementary information.

As MARBLE represents the local flow field around each point, the key difference is that in 'Geometry-agnostic mode', the learnt latent representation is enforced to be rotation invariant. This means that manifolds that are oriented differently or even deformed – but carry the same dynamics – are represented by the same distribution of latent features. This feature uniquely allows the comparison of neural dynamics of different systems, as we demonstrate in Fig. 2h-j against CCA, and is useful when the embedding of dynamics is arbitrary.

When studying a given system (network, Fig. 2d-g or animal, Fig. 3, Fig. 4 b-d) or even across animals when the geometry of neural activities is important (Fig. 4e), we recommend using 'Geometry-aware mode', which has higher expressivity.

We have now expanded on this issue in our explanations in the text to clarify this distinction and the usefulness of both MARBLE modes when learning neural manifold representations.

In particular, I'm not sure what use geometry-aware embeddings have over other manifold characterization and dynamic model techniques.

We agree with the reviewer that we were not clear enough on the advantage of geometry-aware embeddings, and we have now corrected this in the text. In particular, we have commented on the following advantages:

- 1) Unsupervised learning. This is uniquely possible because of the local viewpoint.
- 2) More expressivity over pointwise methods (PCA, t-SNE, UMAP, etc.) by representing dynamics, not just neural activities, as points in space (Fig. S8).
- 3) Higher decoding accuracy and interpretability than dynamical modelling techniques (e.g. LFADS) and supervised representation-learning techniques due to the richer viewpoint of LFFs as opposed to points (Fig. 3-4).

It does seem that geometry-agnostic embeddings allow for comparisons across many different instantiations of neural networks or neural populations performing the same task--if this works well, it is quite exciting, as it might provide another way to compare networks beyond geometric analysis like Canonical Correlation Analysis (CCA) and dynamic analysis like characterizing fixed point structure. This is especially poignant given previous results showing that artificial neural networks trained to perform behavioral tasks show a variety of geometric structures under CCA comparison, but a similar dynamic structure characterized by fixed points (see <https://doi.org/10.48550/arXiv.1907.08549>--I'm not sure this paper is cited in the manuscript, but if not, I believe it may be highly relevant). I think it would be good to see an expanded description of why each of these settings should be used and what they might differentially provide.

We thank the reviewer for this very relevant comment.

To reinforce our point, we reproduced the results of Fig. 2 using CCA, showing a counterexample that CCA is unsuitable for detecting variation in the internal dynamics of the artificial neural networks. This is because CCA is a method that makes a distributional comparison between dynamical systems, looking at the collection of trajectories in a given system as a point cloud. In our RNN example, all dynamical variation takes place within a linear subspace, which can be detected by changes in local flow fields. By construction, the distributional variation between the conditions is dominated by sampling variability caused by different initial conditions across different trials. Hence, CCA is only detecting noise. In contrast, MARBLE is able to distinguish the

different dynamics in connection with the RNN solutions linked to the decision tasks. In particular, while fixed-point detection can find qualitative changes in the dynamics, e.g., the disappearance of a fixed point, it cannot compare dynamics qualitatively because it does not offer a parametrisation of the dynamics.

We did not cite this paper but cited instead Gallego et al. 2017. We have now also added this citation to the new manuscript.

Other manuscript comments

Line 39: What does MARBLE stand for? I can't find a definition for it.

MARBLE stands for Manifold Representation Basis Learning. We have added this to the new version.

Line 44: It's not clear what this means--what is being compared here in contrastive learning? And between what is the latent space shared? It's true that these will be expanded upon later, but perhaps it would be good to give a description of the method with fewer unexplained pieces.

Thank you for this observation. MARBLE represents the neural manifold as a graph. Local flow fields nearby on this graph will be more similar than local flow fields at random locations. This is the basis of unsupervised contrastive learning, which MARBLE is unique for.

We clarified this in the revised version.

Line 75: What is subscript n ? I don't believe it was mentioned anywhere, but perhaps I missed it.

n is the number of data points, i.e. neural activities across a set of trials. We have corrected this in the revised version.

Line 78: What is a p-hop neighborhood? I Googled it, and it wasn't readily obvious what this means. Perhaps it would be better to give a less jargon-heavy description of the method in this section.

P-hop neighbourhoods are points on the graph that are at most p neighbours away (i.e., in Hamming distance). We have made this more explicit in the text.

Line 79: If Z_c is a shared latent space across conditions, then what does subscript c refer to here? Is Z_c the latent activity of the given condition in a shared space? If so, it might be good to be explicit here.

Indeed, Z_c represents the latent representation for condition c . We have emphasised this point in the text.

Line 114: I don't believe the manuscript mentioned what the intrinsic and embedding dimensionalities were, which would be good to have. Also, this is trivial, but I believe this is a paraboloid, not a parabola--but I could be wrong.

We have changed it to paraboloid and have added the embedding dimensions in the text.

Fig 2d: It's unclear what the different colors mean in this figure and how they relate to the colors in 2a. Fig 2j is mislabeled (should be 2i, I think)

The colours in Fig 2d correspond to the colours in Fig 2a. They represent the subset of neurons in the RNN that specialises in the input with the corresponding colour. We determine this by clustering the network weights that relate the inputs to the respective nodes. We have clarified this in the caption of Fig 2.

Line 138: Authors say previous methods only use static snapshots, but CCA matches neural trajectories across conditions--not point clouds. It has also been used to align neural activity (via dynamics of latent activity) across many days (<https://doi.org/10.1038/s41593-019-0555-4>) and even across individuals (<https://doi.org/10.1101/2022.09.26.509498>). However, in favor of the authors' point, the "universality" paper I cited earlier suggests different networks performing the same task may differ in CCA but not in fixed point structure. Similarly, this section has no comparison with any other methods of comparing RNN computations. Would those methods (e.g. CCA) show similar results? I would guess that CCA would show two separate types of computations here, like the "universality" paper.

As we mentioned above, in the new version we have applied CCA to Fig 2i, following closely the procedure of Maheswaranathan et al. 2019. We found that CCA detects no variation in the underlying dynamics. By contrast, MARBLE detects a rich structure of continuous gain modulation and bifurcations.

CCA compares two datasets by decomposing them into linear subspaces using PCA (i.e., SVD) and measures the correlation between the respective principal axes. Because each dataset is considered a point cloud, dynamical information is lost, and the correlation value is only caused by similarities/differences in sampling.

The reviewer is correct in that CCA has been applied to compare neural dynamics, but as the paper of Maheswaranathan et al. 2019 also shows, this will only pick up geometric differences between neural activations, which does not necessarily correspond to dynamical variation. Instead, MARBLE is designed to compare dynamics on manifolds. We have clarified this in the paper.

Line 152: which two representative examples are shown? There are three panels here, so this is a little bit confusing.

The representative examples refer to two network solutions to the DMS task, as illustrated in Figs 2e and Fig S7e. We have now clarified this in the text.

Line 157: It's unclear why the authors would use PCA here first--is there an easy explanation for this?

In geometry-aware mode, MARBLE computes gradients (and higher-order features) to represent local flow fields. We performed PCA in the preprocessing step only to reduce the number of spatial dimensions for computing gradients and obtain a reduced number of parameters in the model.

Line 164: How do the authors determine whether there are subblocks present or if there are gradual changes in this distance matrix? It's not listed here, though the caption mentions hierarchical clustering.

This is a good question, and we apologise for the confusion. We detect the sub-blocks via hierarchical clustering. The continuous changes are visible as a gradual drop-off of the entries from the diagonal. However, to confirm this, we also performed an MDS embedding of the matrix of distributional distances between MARBLE representations of the respective conditions. We have previously only displayed this embedding in Fig 2j, which shows that the conditions are continuously varying along a line (1D manifold) for the respective network solutions.

We now include an additional figure to display the continuous variation corresponding to Fig 2f.

If hierarchical clustering, does one need to specify number of clusters to start with? And what does hierarchical clustering do with gradually changing distance matrices?

We simply took the two largest clusters appearing in hierarchical clustering since we were looking for the neural dynamical correlates of the loss of task performance, which is a binary effect. In more general scenarios, this can easily be replaced with a more sophisticated clustering algorithm, such as modularity maximisation.

We do not use hierarchical clustering to detect gradual changes, but only the block structure. To detect continuous changes within blocks, we use the MDS projection of matrices (see response above).

We have clarified this in the text.

Line 173: An example from my previous point--the authors are switching to a geometry-agnostic MARBLE network, but it's unclear why.

We switched to geometry-agnostic MARBLE because we were interested in comparing the dynamics across network solutions. Since the dynamics in different network solutions are confined to linear subspaces whose orientation is irrelevant to the task and is simply caused by different network initialisations, the geometry-agnostic mode makes MARBLE invariant to these random orientations and ensures that only the dynamical variations are learned.

Spike analysis section (lines 182-215): Unless I'm misunderstanding things, it seems a bit unfair to separate conditions for MARBLE network training first and then compare it to LFADS, which doesn't know anything about behavior except trial boundaries. This method seems more like MINT (<https://doi.org/10.1101/2023.04.05.535396>), where they build a library of neural dynamics with several conditions and then build a decoder off of that. It is impressive that MARBLE outperforms LFADS in neural decoding, but I would like to see how well it performs compared to MINT, which has a similar level of supervision.

We agree with the reviewer that MARBLE considers the trials under a given task condition to belong to an independent manifold. This is necessary because only under a given condition do we have good reason to believe that the dynamics are consistent.

This is not equivalent to supervised approaches. Trajectories under different conditions are only treated separately because within conditions dynamics evolve on manifolds, whose continuity we exploit during training to sample neighbouring flow fields which are going to be similar. At test time, MARBLE embeds individual flow fields and has no information about which condition it belongs to. It very often happens that local flow fields from different conditions are embedded nearby. See, for example, the toy example in Fig. 1d. This would not be expected from supervised methods.

Unfortunately, we could not compare against MINT due to difficulties we had in running their package. Instead, to compare with supervised methods, we performed a detailed comparison against CEBRA on two separate examples (Fig. 3-4) while being supervised with time labels only or with behavioural information. We find that MARBLE exceeds the performance of CEBRA in both examples and degrees of supervision on a single animal. Remarkably, we also find that MARBLE discovers consistent embeddings across animals (Fig. 4e). This is only possible with CEBRA when supervised by behavioural information, not when the labels are time only. This reinforces the fact that unsupervised MARBLE can not only be used for decoding but also for making unbiased comparisons between neural dynamics across animals.

Line 196: It seems unsurprising to find an embedding that reflects spatial geometry of reaches, especially since, as the authors note, neurons in PMd are sometimes directionally tuned. Even running PCA on motor cortex data sometimes reveals a similar spatial geometry.

In the revised version, we have now included a more extended comparison against PCA, t-SNE, UMAP (Fig. S8), LFADS (Fig. 3) and CEBRA (Fig. 4). We find that the embeddings these methods produce are significantly less interpretable and do not unfold the spatial geometry of reaches.

Notes about codebase

The authors have helpfully made all the code and data easily available online. I found the code to be quite well documented (it also taught me what MARBLE stood for). Below are a few notes from my trying to get the code working on a couple different machines.

We thank the reviewer for thoroughly testing our code. The original code was available on Ubuntu and Macs with Intel processors, as we indicated in the GitHub repository.

We have now provided precompiled environments to work with Windows and Apple Silicon (M1, M2, M3).

First, I tried running the code on my M2 Macbook Air, with limited success:

- The environment does not install out of the box easily on macOS (Macbook air M2). Had to strip package-specific version tags (as suggested by this Stack Overflow answer: <https://stackoverflow.com/a/56780297>). Even after that, there were several packages that needed to be replaced by pip binary packages because conda channels did not have arm64 versions of them. Specific changes:

- Moved pytorch-* packages to pip as torch-*
- Changed icu version to 67.1 as 58.2 is only available from win-32 architecture in conda-forge
- There seem to be no mkl* packages that work for Apple silicon, so I removed them
- Even after removing intel-based packages, environment did not resolve through conda. I resorted to installing packages one by one using pip in a conda environment for the specific notebooks I was trying to test.

After this, I tried running the code on my Windows computer through Windows Subsystem for Linux (WSL), which was far more successful:

- Environment sets up well on WSL out of the box, though I'm not sure it's actually using the GPU during training, despite CUDA and all being installed.
- Error when running `vanderpol.ipynb`: `MARBLE.construct_dataset(...)` doesn't run because `labels` isn't specified. It seems this code was changed in the last couple days but untested. I suggest not pushing code to the master branch until it is fully tested, or else create a git tag for a version that works to reproduce paper figures without errors. Reverting to the last commit that passed CI checks on GitHub resolved error

We agree with the reviewer. I think this must have been an error on our side, which we spotted too late.

- Trying to run visualization notebook from spike analysis has errors because `ipyml`, `neo`, and `elephant` are not installed as part of environment. It would be good to go through all notebooks to make sure all the required modules are part of the environment.yml file.

These packages are not part of MARBLE and are necessary only for preprocessing the neural spiking dataset. We have now added 'pip' commands at the beginning of the notebook to install these packages automatically.

References

Maheswaranathan, Niru, Williams, Alex H., Golub, Matthew D., Ganguli, Surya, Sussillo, David. Universality and individuality in neural dynamics across large populations of recurrent networks. arXiv. 2019-07-19. doi: 10.48550/arXiv.1907.08549

Gallego, Juan A., Perich, Matthew G., Chowdhury, Raed H., Solla, Sara A., Miller, Lee E. Long-term stability of cortical population dynamics underlying consistent behavior. Nature Neuroscience. 2020-01. doi: 10.1038/s41593-019-0555-4

Safaie, Mostafa, Chang, Joanna C., Park, Junchol, Miller, Lee E., Dudman, Joshua T., Perich, Matthew G., Gallego, Juan Alvaro. Preserved neural population dynamics across animals performing similar behaviour. 2022-09. doi: 10.1101/2022.09.26.509498

Perkins, Sean M., Cunningham, John P., Wang, Qi, Churchland, Mark M. Simple decoding of behavior from a complicated neural manifold. bioRxiv. 2023-04. doi: 10.1101/2023.04.05.535396

Reviewer #4 (Remarks to the Author):

This paper introduces a novel dimensional reduction technique, called MARBLE, which focuses on local regions of the subspaces, and also utilizes an RNN to learn the latent vectors.

We regret if there has been a misunderstanding. MARBLE does *not* use an RNN (recurrent neural network). Our method combines geometric deep learning (GDL) ideas with empirical dynamical modelling (Sugihara et al., Science 2012) to learn dynamical processes. Therefore, MARBLE introduces the theoretical and computational developments to allow the use of GDL to learn dynamical processes and, in turn, neural data.

We believe this provides a significant advance in neuroscience methodology and machine learning, as we demonstrate through an expanded set of comparisons and a new case study on rat hippocampal recordings.

The authors motivate this advance by discussing shortcomings of existing dimensionality reduction techniques in terms of generalization across subjects and poor fitting of complex geometries.

We thank the reviewer for raising the issue of motivating the advances provided by MARBLE. We recognise this might not have been as clear as it could have been in the original paper, so we have expanded and corrected our motivation in the corrected manuscript.

Please let us clarify that “generalization across subjects” is only one advantage of MARBLE, and “fitting of complex geometries” is a feature of MARBLE but not our theoretical advance. This problem is well-studied in manifold learning, which we use for manifold approximation.

In summary, the specific advantages over existing methods provided by the new mathematical and computational framework underpinning MARBLE are:

1. **Unsupervised representation-learning of neural dynamics.** This is a significant advantage in scientific discovery over previous supervised methods such as CEBRA that require user-defined labels. Despite being unsupervised, MARBLE exhibits *better interpretability and decoding performance than supervised methods* in most cases we studied (Figs. 3-4)
2. **Unbiased comparison of dynamical processes across systems** (networks and animals). We show that MARBLE enables *comparing computational strategies across systems* (Fig. 2) and *discovering consistent embeddings across animals* (Fig. 4), which have significant theoretical and experimental implications, e.g., in brain-machine interface research. This is unlike dynamical systems models (LFADS, SLDS), which depend on the embedding of the trajectories (which usually change across experiments) and, in turn, the identity of measured neurons. This is possible as MARBLE learns dynamics over the manifold, which does not depend on the embedding.

3. **Representation of dynamics over the manifold.** A key theoretical advance is to represent dynamics as a collection of local flow fields (LFFs) rather than a collection of points, as done by many manifold learning methods such as PCA, t-SNE, UMAP, etc. Specifically, these methods do not represent the dynamics and thus give significantly less interpretable representations (Fig. S8).

The technique is then illustrated in one toy example and a delayed-match to sample task. It is then applied to a compiled dataset from a single macaque performing a center out reaching task.

Our choice of examples followed from our endeavour to introduce the properties of the method through motivating examples through complex computational examples to experimental data sets.

Firstly, to build up intuition, and as pointed out by the reviewer, we have used two mathematical examples: (i) rotational/linear vector fields and (ii) van der Pol on a paraboloid). Specifically, each of these examples is used to explain key properties of the method: the importance of geometry-aware and geometry-agnostic learning in (i); and the relevance of nonlinearity in the dynamics and nonlinearity in the manifold in (ii). We have clarified this in our revised version.

Secondly, given its interest in computational neuroscience and machine learning, we have used the RNN example of a delayed-match to sample task. As noted in the response to Reviewer 2, we have added further comparisons with CCA to highlight the advantages of MARBLE in the analysis of RNN dynamics in connection with task prediction.

Finally, another focus of the work is the use cases from experimental recordings. To add further evidence, we have now included another experimental dataset to the macaque reaching task recordings. Specifically, we have now applied MARBLE to neural firings in the rat hippocampus during navigation of a linear maze (Fig. 4). Through this new use case, we show that MARBLE representations are significantly more interpretable and decodable in single sessions than the competing CEBRA method, both supervised with temporal labels or behavioural labels (animal position and direction).

Furthermore, we also showed that MARBLE embeddings are consistent across animals (Fig. 4e) without using behavioural labels (animal position and direction) as required by CEBRA. This is a significant advance because it allows neural dynamics in a given task to be compared across animals to test for similarity in cognitive strategies. This also reinforces our finding in RNNs applied to the delayed-match-to-sample task that MARBLE provides a well-defined metric across neural networks. Comparison across systems is a distinctive and important advance, which opens the door to studying the cross-animal aspect of neural computations.

This paper has serious flaws in its lack of comparison to other methods and the lack of a single compelling example of achieving a previously unseen level of performance for at least one task.

Although a direct comparison of MARBLE to other methods is not always feasible, due to its unique capabilities, we have expanded whenever possible the comparisons to other methods. Specifically, we present extensive comparisons as follows:

1. **Comparison of dynamics across systems.** We demonstrate this in Fig. 2h-j on RNNs trained on the delayed-match-to-sample task. We compare this to Canonical Correlation Analysis (CCA), showing that CCA is unsuitable for detecting variation in the internal dynamics of the artificial neural networks. This is because CCA is a method that makes a distributional comparison between dynamical systems, looking at the collection of trajectories in a given system as a point cloud. By contrast, MARBLE represents the data as a collection of local flow fields (as opposed to points), which represents the dynamical system.
2. **Decoding across animals.** In Fig. 4e, we compare MARBLE against CEBRA in finding consistent embeddings across animals. We find that while CEBRA needs behaviour (animal position and direction) as labels, MARBLE can detect consistent embeddings from neural dynamics alone. This means it is suitable for comparing cognitive strategies across animals.
3. **Interpretability and decoding within a single animal.** In Fig. 3 and 4a-d, we perform a detailed comparison against CEBRA, supervised and self-supervised and LFADS. In Fig. S8, we also compared against PCA, t-SNE and UMAP. We find that in recordings from the macaque premotor cortex and rat hippocampus MARBLE embeddings are significantly more interpretable and decodable than competitors.

Specifically, at the first portion with a toy example, no other technique was introduced in comparison, and the capture of complex rotational dynamics is not a new feature for these techniques. It would have been best to illustrate any specific advantages of this technique against others at this stage.

We apologise for any lack of clarity. It was not our intention to imply that MARBLE is designed to capture rotational features. Rather, as mentioned above, this example serves the purpose of building intuition about MARBLE representations and the fact that MARBLE can be operated in two modes (geometry-aware and geometry-agnostic) to gain different representations addressing different needs. This simple example allows for a direct understanding of this unique feature in-built in MARBLE, whereby the local directionality of the flows can be either used or effectively ignored so that the analysis can focus on the manifold alone or on the directed flows and the manifold together. We have clarified this issue in our updated version.

Note that in all other examples, Figs. 2-4, where we study much more complex non-linear dynamics consisting of multiple fixed points, line attractors, limit cycles, etc, we perform a detailed comparison against current widely-used methods.

Similarly, for the delayed match to sample, the authors simply report that a high amount of variance was explained, with no comparison to the state of the art.

We thank the reviewer for this suggestion. To our knowledge, no other method is capable of comparing dynamics across RNNs based on a sparse set of trajectories. At the request of the reviewers, we now add a comparison against CCA, showing that it cannot detect informative variation. This is because CCA uses a point cloud representation, which is the same across all networks, as the dynamics evolve on the same plane but with different orientations. However, CCA misses within-manifold dynamical variations picked up by MARBLE.

Finally, the last section of the paper introduced a single neural dataset, which by itself does not address the motivated problem statement of generalization nor complex geometries. Success is claimed by showing that 7 trajectories emerge from the method.

With this example, we highlight both the geometric interpretability of the unsupervised MARBLE representations, which match the reach configurations based on neural data alone without any labels, and its superior decoding performance. By comparison, PCA, t-SNE, UMAP (Fig. S8), LFADS, Fig. 3) and CEBRA (Fig. 3-4) performed significantly worse in both interpretability and decoding, even with supervision.

To address the important issue of generalisation, we now also demonstrate the application of MARBLE to another case study of the rat hippocampus (Fig. 4). Here, we show that MARBLE generalises across animals to reveal the geometry of spatial positions and direction without the need for user-defined labels. The results of unsupervised MARBLE for this use case also compare favourably with CEBRA (supervised and self-supervised).

This is finally compared to a single technique (LFADS), whose performance does not appear analogous to its top performance in prior work, and which is not necessarily the only candidate for the existing state of the art.

The LFADS results reported are numerically identical to the original published paper (Fig. 4e,f in Pandarinath et al. Nat Methods, 2018) Indeed, we have obtained the LFADS model results from the authors themselves, and we have discussed with them to ensure their results are reproduced properly.

Following the recent publication of CEBRA, we have now performed a detailed comparison against CEBRA in two different case studies (Fig. 3, 4).

Many nonlinear techniques can reveal trajectories like Figure 3h from neural data of the type described, so it's not clear what is sufficiently novel here.

We are not aware of any other non-linear technique that can reveal such trajectories from sparsely sampled data under different initialisations. We have made extensive comparisons against t-SNE, UMAP, LFADS and CEBRA, which are state-of-the-art

nonlinear methods, which could not reveal the geometric correspondences discovered by MARBLE.

Also, the general claims of this paper would tend to require testing with additional large-scale datasets, which are broadly available from multiple animal models.

We now provide a comparison in two case studies, using large-scale datasets, both in the macaque premotor cortex (Fig. 3) and the rat hippocampus (Fig. 4). We hope these serve to exemplify the applicability of the method.

Overall, I am not persuaded that this technique achieves best in class performance in revealing repeatable and generalizable dynamics, nor for revealing kinematic trajectories.

We hope we have provided additional evidence for the repeatability of our results and to support the claims of the paper.

Response to reviews.

We thank the Reviewers for the constructive criticism of our article. In the revised manuscript, we provide additional controls, sensitivity analyses, and clarifications to the text and figures to support our findings.

We believe the revised article is substantially improved and demonstrates MARBLE's immediate practical relevance to analyse neural recordings in broad settings.

Before we address the individual concerns, we address two points raised by both Reviewers.

1. Improved presentation

As the Reviewers remarked, MARBLE introduces a novel viewpoint for representing neural dynamics by leveraging the low-dimensional geometric structure of neural states and developing differential geometric and geometric deep learning techniques to learn dynamical flow fields.

We have thoroughly revised the manuscript by

- presenting an improved discussion of previous methods in the Introduction,
- illustrating the methodology in greater detail on extended Figs. 1, 2 and S1,
- performing additional controls against linear methods (CCA, Fig. 3i,j and TDR, Figs. 4g, S11),
- providing depiction of the training data (Fig. S9) to illustrate why methods such as CCA can fail to distinguish dynamics in a general setting
- performing sensitivity analysis showing that MARBLE retains state-of-the-art performance for a broad range of hyperparameters (Figs. S13, S14).

2. Unsupervised learning of MARBLE

The Reviewers raised the concern that MARBLE receives user input - specifically, information about the experimental conditions, which inform which dynamical trajectories are dynamically consistent - and, as a result, it is not unsupervised.

The term 'supervised' in machine learning generally refers to introducing correlations between the inputs (in our case, the neural states) and the outputs (their corresponding latent states). Hence the presence of user input does not automatically make a method supervised.

To illustrate this point, consider first a supervised method, e.g., CEBRA-behaviour. In the macaque reaching example, when using behavioural conditions as labels, this method imposes that a neural state belongs to a certain class and, as a result, clumps neural states into clusters (Fig. 4e, middle), losing temporal information as a result.

By contrast, in MARBLE, condition labels do not mean class assignment, but a notion of adjacency, which allows extracting local flow field features (LFFs). Crucially, unlike neural states, LFFs are broadly shared between conditions - they simply encode the spatial gradient (Fig. S1e, S2) of the local field, quantifying the local variation of the dynamics in neural space. Therefore learning an LFF around a neural state does not impose a class assignment of the neural state itself. An exception is when two flow fields are very different (Fig. S6), i.e., they share no LFFs, in which case it makes sense to

separate them in latent space. In contrast to CEBRA-behaviour, MARBLE preserves the temporal structure in the macaque reaching neural data (Fig. 4e, left) and thus allows accurate decoding of velocity (Fig. 4g, right).

Let us also use an analogy to spectral clustering - an unsupervised algorithm (but note that beyond this analogy, the purpose of spectral clustering and MARBLE are completely different, the former being clustering and the latter being representation learning). Both methods use a notion of adjacency between data points as user input. This adjacency graph is used to extract features, Laplacian eigenvectors in spectral clustering, and LFFs in MARBLE (Fig. 1d), which as we explain above, do not carry class assignment information.

In spectral clustering, features are then fed into an unsupervised algorithm, k-means clustering. In MARBLE, this is an inductive, unsupervised contrastive learning algorithm based on GraphSAGE (Hamilton et al. 2017). Crucially, in both spectral clustering and MARBLE, adjacency does not predispose points to be mapped close; points that are close in (neural) space can be mapped far apart in embedding space and vice versa. For example, this would happen when applying spectral clustering to points generated from two overlapping Gaussian distributions. Likewise, MARBLE will find disjoint clusters (Fig. 4e) or identify similarities and continuous structures between datasets (Fig. 2b, 2d,e, 3f, 3i,j). For example, in Fig 2b, rather than separating flow fields into four clusters, MARBLE identifies the 1D manifold in latent space that parametrises the angular and radial variation in the vector fields (note how LFFs far away can be mapped close together and vice versa). This behaviour is emergent; *a priori* all of these datasets were treated as different conditions, yet continuous, overlapping structure emerges instead of clusters. This can happen because in MARBLE the same multilayer perceptron maps all LFFs independently. This yields parameter sharing, which allows MARBLE to map two LFFs in entirely different parts of the dataset close together.

Several of these points are addressed below in more detail in response to the issues raised by the reviewers. We have also taken due care to explain the feature extraction and learning steps clearly in the manuscript.

We hope that our revisions have sufficiently addressed the Reviewers' concerns.

Reviewer #2:

Remarks to the Author:

In this revision to their previously submitted manuscript introducing MARBLE, Gosztolai et al. have provided additional controls and datasets, as well as tried to make the description of the method clearer. In this, I believe they've largely succeeded--I find the manuscript to indeed be clearer than it was when I first read it. I'm also reasonably impressed with the new rat hippocampal data analysis. However, I still have some important concerns that I believe should be addressed.

We thank the Reviewer for the positive feedback on our revised manuscript and the new rat case study. We have now implemented additional revisions to address any remaining concerns. See below.

Acknowledgement of previous results

After reading the revised manuscript, I believe I'm starting to understand something I didn't the first time around--this method finds a latent space representation of not just neural state but also empirical estimations of local flow fields. This is in contrast with most other latent factor analysis methods being used in neuroscience, which find a latent representation of only the neural state--this seems to be what the authors are calling "point cloud" methods.

We are pleased to hear this now comes across clearly. MARBLE represents the local flow field (LFF) around every neural state, which first 'lifts' the state into a much higher dimensional space--from d dimensions to $d^{(p+1)}$ -dimensional feature vectors describing the LFFs, where the p is the order of the features ($p=2$ in all case studies). Then, it maps these vectors to a low-dimensional latent space. Thus, MARBLE represents both neural states and dynamics, which is richer than previous methods that explicitly only represent neural states.

See the following answers for more details.

First of all, it was the new clarity of the manuscript got me to this understanding, so assuming it is correct, the authors have done a better job getting their ideas across in this revision. However, it did take me some time to fully grasp the intention of this method, and I think part of the reason was how the authors were characterizing and treating computational modeling methods previously used in neuroscience, including so-called "point cloud" methods and dynamical systems models.

By labelling previous methods as "point cloud" and "dynamical systems" methods, we intended to classify them based on the type of information they use as inductive bias in learning.

We now removed the term 'point cloud' to avoid confusion. Instead, we now distinguish previous dimensionality reduction methods by whether they explicitly model time information in consecutive neural states (dynamical systems methods such as RNNs, LFADS, symbolic dynamics, etc.) or not (PCA, UMAP, t-SNE, etc.).

We contrast these methods to MARBLE, which uses information on the spatial variation of the flows in the neighbourhood of neural states based on nearby trajectories. Note that this also contains time information because temporally consecutive points are typically also close over the neural manifold.

In the revised manuscript, we clarified the description of previous methods in the Introduction.

In particular, while reading the manuscript, I couldn't help but feel that the authors may be unaware of how previous methods have been used, and as such, it seemed like previous methods were being treated unfairly.

While we aimed to highlight MARBLE's advantages over other methods, we are striving for a fair comparison as our extensive benchmarking shows. For example, we used CEBRA both self-supervised (time only) and supervised (condition labels) to provide the most stringent benchmarks.

For example, the authors showed that MARBLE applied to monkey center-out reaching data "reflect the spatial geometry of the physical reaches", and noted that this arrangement wasn't readily apparent in CEBRA, LFADS, PCA, t-SNE, or UMAP. However, this seems to ignore a number of neural population studies that have found this spatial geometry with simpler, linear methods. For a recent example, see Sun et al. 2022, which uses Targeted Dimensionality Reduction (TDR) to find a linear projection of neural population activity that separates out different reach directions. While it is true that such previous methods require more supervision than MARBLE does and that MARBLE might separate out the clusters better, the fact that this kind of spatial geometry exists in motor cortical activity is not surprising. In fact, this geometry is one of the main reasons that simple linear neural decoders function for most of the brain-machine interface work that has happened over the last decade.

We agree that the spatial geometry in motor cortical activity is intuitively expected given the task structure. Yet, discovering it without supervision is highly challenging.

We also agree that TDR can separate out reach conditions as clusters in latent space as we now demonstrate in Fig. S11b. However, the performance of TDR is far surpassed by MARBLE (compare Fig. S11b and 4e) despite the fact that TDR requires both temporally aligned trajectories (as regression coefficients are computed point-by-point), and physical reach coordinates at inputs to define regression subspaces. By contrast, MARBLE only assumes that reaches in a specific direction are dynamically consistent (driven by similar inputs), which allows feature extraction as we describe in the preamble. This is not a sufficient condition to find spatially separated and ordered (by condition and time) latent states.

Rather, we discover this spatial arrangement as an emergent property of MARBLE embeddings. More specifically, the spatial arrangement of reaches in latent space reveals that the manifolds for individual reach conditions are, in fact, submanifolds of a global manifold of all reaches, which was evidenced in previous studies (e.g., Churchland et al. 2012, Sun et al. 2022) using supervised methods.

We have now cited Sun et al. 2022 and the TDR method (Mante et al. 2013) and further clarified the differences between MARBLE and how supervised methods such as TDR find representations.

Another example of this is the perspective the authors take on comparing neural population activity between multiple datasets. The authors contend that dynamical models "depend on the particular embedding of trajectories in state space and, in turn, the measured variables (i.e., neurons), hindering their ability to compare representations across experiments" (Page 2). They also note that "While it is possible to 'stitch' multiple datasets by fitting auxiliary transformations or aligning the neural dynamics to behaviour, this relies on the

assumption that the respective neurons encode the same computation" (Page 8). These claims seem to discount previous work to the contrary.

A trajectory of neural states depends on which set of neurons are being recorded and other details (e.g., modality) of the measurement. In non-linear dynamical theory, these 'nuisance variables' are formalised as a measurement function, an unknown non-linear relationship that provides a unique embedding for each session. Models that learn the temporal variation of trajectories will, by definition, depend on the particular embedding, which in turn depends on the details of the recording.

We clarified this in the Introduction.

First, dynamical models don't necessarily have to depend on the particular embedding, as exemplified by the original LFADS work--at the end of that paper, the authors showed that one can use the same LFADS dynamical model across multiple sessions, simply by changing the read-in and read-out weights from the model. Gosztolai et al. actually reference this ability to "stitch" in the second quote above, but then mention that it requires that neurons encode the same computation. This is simply not true--in theory, two neurons could have completely swapped roles, and the read-in and read-out weight would only have to swap to adjust accordingly.

We apologise for the confusion, which we believe was due to using the word "computation". We did not mean that neurons "individually" encode the same computation but rather that they encode the same latent dynamics as a population.

Consider, for example, our RNN example in Fig. 3h (also Figs. S8 and S9), where we demonstrate two RNNs that possess different internal dynamics yet solve the same computation (a contextual decision-making task). In LFADS, it would be impossible to relate these dynamics by changing the read-in and read-out weights; non-linear latent dynamics with different fixed point structures cannot be matched by linear transformations.

However, we do agree that changing the read-in and read-out matrices can be sufficient to create a mapping between sessions that encode the same population dynamics. This is not only possible in LFADS but also in MARBLE as shown by our rat hippocampus example (Fig. 5e).

We have rephrased this statement in our revised manuscript without the words "computation" and "stitching" to avoid confusion and added an additional clarification.

Second, these statements seem to discount work with linear methods that don't even require the same neurons to be recorded to align neural trajectories across sessions. Gallego et al. 2020 shows that activity on the neural manifold during a particular behavior is consistent enough to be linearly aligned and decoded from with a single decoder over a period of up to two years--a period during which the neurons being recorded from are certainly not the same. Similarly, Safaie et al. 2023 (cited by this manuscript) showed the same thing but across monkeys, which clearly have different neurons.

It was not our intention to discount linear subspace alignment methods, such as CCA, for aligning sessions--we apologise for this impression. We agree that linear methods can be powerful in aligning latent representations,

as Gallego et al. 2020 and Safaie et al. 2023 show and we have reinforced this point further in the updated manuscript.

However, as our control experiment involving CCA shows (Fig 3i,j), a high degree of alignment using linear methods such as CCA is not a sufficient condition for having consistent dynamics. In our extended and corrected CCA results reported below, we show that CCA induces similar consistency between dynamical flow fields that actually differ due to gain modulation (Fig. 3i,j, right). These subtle dynamical differences are reliably detected by MARBLE (Fig. 3i,j, left, and middle for negative control). Our results suggest that CCA is sensitive to sample variation caused by the randomness between trials across conditions.

Our results suggest that in Gallego et al. 2020 and Safaie et al. 2023 CCA succeeded because the dynamics under a given task condition evolve in bundles in neural space (e.g., see figure below, Fig. 4c in the text), which means that trial-averaged dynamics -approximate well the single-trial dynamics. In this special case, distributional differences detected by CCA reflect dynamical differences.

This highly regular state-space structure need not be the case, e.g., for the highly nonlinear dynamics during cognitive tasks that we study (see figure below, Fig. S9 in the text). Here, the trial-averaged dynamics is not a good approximation of single-trial dynamics. Thus, distributional shifts detected by CCA can indicate both dynamical variation and sample variation.

We have now cited Gallego et al. 2020 in addition to Safaie et al. 2023 (already cited) and have clarified this ambiguity in the Introduction, Discussion and when presenting our CCA control experiment.

It is likely the authors did not mean to discount such previous relevant work, but the way parts of the manuscript is written makes it come off this way. I propose that the authors rework some of these parts to better highlight how MARBLE adds to previous work.

Thank you for the constructive suggestions that have helped clarify our explanations and our thinking on this important issue. We hope that the revised manuscript text improves the comparison with previous methods.

Canonical correlation analysis

In my previous review, I mentioned that Canonical Correlation Analysis (CCA) would be provide a good comparison for some of the results, and I still believe that is the case. However, I'm thoroughly confused by how the authors have described and used CCA here. I'm concerned that there has been a critical misunderstanding, either in my reading of the manuscript and rebuttal to my previous review, or in the authors' use of CCA.

Regrettably, there was indeed a small bug in our implementation of CCA, caused by not transposing one of the feature matrices, and the high sample variance from condition to condition in our data (Fig. S9) had led us to believe (incorrectly) that the noise in the CCA control experiment was a valid result.

We have now corrected this bug and repeated this control experiment, as well as an additional negative control (see below). We have checked the results using two separate implementations of CCA: one as previously implemented using the scikit-learn library (but with the bug fixed) and a custom code that was obtained from the Matlab `ccacorr()` function (as used in Safaie et al. 2023), obtaining identical results.

Crucially, the corrected CCA control experiment now provides a more refined comparison, allowing us to reinforce the differences between MARBLE and CCA. We thank the Reviewer for this question that has led to this improvement.

The new Fig. 3i,j allows us to make the following conclusions:

1. MARBLE representations are sensitive to continuous changes in the dynamical flow field during gain modulation, which otherwise preserve the fixed point structure.

This is shown by the progressive decrease of off-diagonal entries in the distance matrix (Fig 3.i, left), and corroborated by the MDS embedding of the distance matrix (Fig 3.j, left), which shows that shifts in MARBLE latent representations capture the continuous variation of the underlying gain parameter in all three networks simultaneously.

CCA could not detect these dynamical changes, as shown by the approximately constant distance matrix (Fig 3.i, right) and mixed-up labels in the corresponding MDS embedding (Fig 3.j, right).

2. MARBLE representations detect no dynamical variation during unstimulated epochs (Fig 3.i, middle).

This further confirms that the variation in (Fig 3.i,j left) is due to gain modulation. CCA agrees with MARBLE within a given block representing a network (Fig. 3i, middle - MARBLE, right - CCA) but was not sensitive to discriminate differences across networks (Fig 3.i,j, right).

3. Both MARBLE and CCA are able to capture the similarity across different RNNs with similar fixed-point structures (two-block structure in Fig 3i,j left and right).

As argued above, our conclusions are in line with Gallego et al. 2020, Safaie et al. 2023, which show that CCA can, under some circumstances, find similarities between neural dynamics whose underlying latent dynamics are the same. Yet, the examples in those works focused on motor control problems during simple tasks, where trajectories typically evolve in bundles in neural space (see figure and explanation in response above). Based

on our analysis of dynamical landscapes with higher nonlinearity, hence containing multiple fixed points and bifurcations, we find that CCA is not sensitive enough to detect dynamical changes across systems in a general setting. In cognitive decision-making problems, such as the ones studied herein, these fundamentally nonlinear variations have key behavioural relevance.

We have rewritten the corresponding section in the manuscript to clarify these points.

In the rebuttal, the authors say:

"CCA compares two datasets by decomposing them into linear subspaces using PCA (i.e., SVD) and measures the correlation between the respective principal axes. Because each dataset is considered a point cloud, dynamical information is lost, and the correlation value is only caused by similarities/differences in sampling."

It is possible I am misinterpreting this, but as it reads, this description of CCA is wrong. CCA does not measure correlation between principal axes calculated through PCA--this would just be correlation. CCA finds a linear transformation for each dataset such that in the transformed, so-called "canonical" coordinates, the correlation is maximized.

We apologise for the lack of clarity in describing CCA in the rebuttal. Our numerical implementation and description of CCA in the manuscript agree with the Reviewer's description.

Briefly, we consider feature matrices A and B, representing ensembles of time-ordered neural states obtained from two RNNs. Then, we implement the following steps using standard scikit-learn libraries:

- 1) Find the linear subspace description of A and B using singular value decomposition (SVD)
- 2) Finding the (unique) best linear transformation that maximises the correlation between the ordered set of right singular vectors.
- 3) Return the average of residual correlation values.

Importantly, the two datasets must be aligned point-to-point in some way for this to make sense. Thus, CCA is not just an analysis of point clouds--the order of the points in each dataset matters. In this way, CCA-based alignment and analysis have some measure of dynamical information beyond the simple point cloud distribution, even if it is not as explicit as MARBLE's embedding of local flow fields.

We agree that the point-to-point correspondence required by CCA is likely why it has been so powerful in detecting similarities across animals in previous studies, such as Gallego et al. 2020 and Safaie et al. 2023, where trajectories evolve in spatially compact bundles (see above).

We also agree that CCA can encode temporal information, albeit only implicitly in the ordering of feature matrices. However, this hinges upon the alignment of single-trial trajectories across conditions (which does not hold in general and MARBLE does not require).

However, as we argue above, in our case study, such point-to-point correspondence between datasets loses its meaning. In particular, in the RNNs we study, trajectories explore different portions of the state space (Fig

S9) because of the noise term (equation S20) and the highly nonlinear flows (Fig 3e, Fig. S8). This is especially true when the RNNs implement altogether different non-linear dynamical processes (Fig S8e, S9).

We now elaborate in the text that, because MARBLE detects flow field features, it does not require point-to-point alignment and can compare much more general dynamical systems and that these comparisons are robust to sample variation (Fig. 3f) and robust to differently embedded manifolds (Fig. 3i,j).

In the case of neural data, CCA alignment is usually done by averaging trajectories in neural state space over trials with the same behavioral conditions in each dataset. Then, finding the transformations that best aligns the neural trajectories allows one to project individual trials into the "canonical" space. As mentioned above and in my previous review, this type of analysis has allowed previous works to find consistent neural trajectory shapes in the same subjects over years (Gallego et al. 2020), and across different subjects (Safie et al 2023).

As the reviewer points out, condition-averaging typically precedes the application of CCA. Yet, as we mentioned above, this is a valid operation when the trajectories under a given condition evolve in bundles in neural space (see figure and explanation in response above), but not when the trajectories are derived from a highly non-linear flow field having multiple fixed points (Figs. 3c,e,h, S8c,e, S9). In the latter case, trial-averaging would lead to the destruction of the dynamical information.

We have clarified this point in our manuscript.

Later in the rebuttal, the authors state that CCA "will only pick up geometric differences between neural activations, which does not necessarily correspond to dynamical variation", citing the Maheswaranathan et al. 2019 paper. I'm not sure I understand this interpretation of that paper--the authors' reasoning here suggests that somehow CCA will not detect differences in dynamics, whereas the Maheswaranathan paper shows that CCA is more sensitive to changes in the specific implementation an RNN happens to use for a given task, while the fixed point analysis (an analysis of dynamical scaffolding) shows similarities across RNNs performing the same task. Thus, I would say that CCA would be more sensitive to changes in dynamics.

Maheswaranathan et al. 2019 finds that CCA detects differences in flow fields when fixed point detection does not. We agree that one interpretation of these results is that CCA is more sensitive than fixed point detection in discriminating flow fields. However, these findings can also be explained by geometric differences between the corresponding manifolds, which otherwise carry the same dynamical flow field, i.e., same fixed point structure. In fact, RNNs impose no constraint on the geometry of the neural manifold, apart from smoothness (which MARBLE exploits). This is what Maheswaranathan et al. reports: "We find the geometry of the RNN representations can be highly sensitive to different network architectures".

Let us thus reinforce our point above that dynamical systems can be distinguished by (1) different fixed-point structures, (2) continuous (within-manifold) differences within the flow fields that preserve the fixed point structure and (3) the embedding of these flow fields into neural state space.

In Maheswaranathan et al. the differences in dynamical systems detected by CCA was from (3), which is corroborated by our results in Fig. 3ij. This figure shows that CCA is not sensitive enough to detect within-manifold differences in the dynamics (point 2, caused by gain modulation in our example).

Our results are thus in agreement with the results of Maheswaranathan et al., and portrait a more refined picture of differences between dynamical systems. We have clarified this in our manuscript.

With that background, I am quite confused by the CCA results for the low-rank RNN analysis in the manuscript. The authors quite nicely show that there are two solutions for this one task that result in very different neural trajectories (similarly to what is shown in the Maheswaranathan paper). In a test of comparing two different iterations of Solution I and one iteration of Solution II, MARBLE nicely categorizes the two Solution I networks as more similar to each other than they are to the Solution II network. However, the CCA results are complete noise, despite the flow fields for the two Solution I networks being remarkably similar but oriented differently in the neural space. I'm left wondering why CCA couldn't align these two networks--is it because the authors' understanding of CCA is flawed (as evidenced by their description of CCA in the rebuttal)? I imagine part of it is because the authors used a random smattering of initial conditions that didn't actually align well across datasets, but I want some more information about this, and I couldn't find a good description in the methods. As it is, I'm concerned that this control doesn't make sense to me.

We hope that the above explanations, additional examples and modifications in the text have successfully addressed the concerns raised.

"Geometry"-aware vs. -agnostic

One of the things I found most confusing about the original manuscript was the distinction the authors made between the "geometry-aware" and "geometry-agnostic" modes of MARBLE. In this revision, I find that this problem is mostly taken care of, thanks to some rewriting and the addition of Figure 1d, which shows the differences between MARBLE results on simple constructed flow fields.

We thank the Reviewer for the positive feedback.

However, while I believe I now understand the difference between the two modes, I don't agree with the naming of it--I believe that is the piece that caused me the most confusion previously. What the authors start to describe as "geometry" for these purposes is revealed to be mainly the orientation of the local flow fields (page 4). Intuitively, this is not what I would call geometry--it seems to me to be more about how the flow field is embedded in the neural space. The authors even go on to say a few sentences later that the "geometry-agnostic mode is able to discount the arbitrary rotations in the LFFs induced by different embeddings" (page 4). Later on, the authors write "geometry-agnostic MARBLE can extract both abrupt and continuous dynamical variation while being invariant to manifold embedding at a slight loss of expressivity compared to the geometry-aware mode." Further, to justify a "geometry-agnostic" MARBLE in the low-rank RNN analysis, the authors write "Due to the arbitrary embedding of the manifolds across networks, we used geometry-agnostic MARBLE" (page 6). As such, it seems to me that the two modes here are really "embedding-aware" and "embedding-agnostic", and based on how the authors write about the two modes, I think they might agree with me.

We thank the reviewer for raising this issue. Indeed, we had used geometry and embedding interchangeably, which caused the confusion. We have now changed 'geometry-aware/agnostic' to 'embedding-aware/agnostic',

while at the same time changing 'MARBLE embedding' to 'MARBLE latent representation', which we agree is a more accurate terminology.

Secondly, and related to my "previous work" and "CCA" comments above, the Gallego et al. 2020 paper using CCA to align neural trajectories across many sessions in the same monkey has a careful description of the hypothesis that neural activity on different days is simply a different embedding of the true underlying neural manifold. That paper follows that idea to build an embedding-agnostic decoding procedure using CCA. Later work from that group and others continue to use that terminology. I think this manuscript about MARBLE might do well to continue using the same terminology so as to keep the field's nomenclature consistent and to avoid alienating readers who might already be familiar with previous work. I realize that this may clash with the terminology used for the latent representation found by MARBLE, which the authors are also calling an embedding, but maybe the authors can separate these terms by only referring to the MARBLE embedding as a latent representation. I believe this would have alleviated much of my confusion.

Thank you for the suggestion. As mentioned above, we have changed the terminology to adhere to the language in Gallego et al 2020 and other works.

Definition of "unsupervised" learning

As before, in the analysis of the monkey reaching dataset, the authors run MARBLE independently on each reach condition--this, I mentioned in my previous review, seems like it would provide an unfair advantage over LFADS, as this procedure provides some supervision during training. The authors mention in the rebuttal that this is because they don't want to assume consistent dynamics across different conditions (presumably to separate the local flow field computations). While that may be a good reason to train the model this way, it cannot be truly called "unsupervised" as the authors seem to be claiming.

Although MARBLE receives condition labels to gain structural knowledge (adjacency between neural states), this does not introduce correlation between input and output; hence, one would not label it as 'supervised' in a statistical learning sense. Please see the preamble to this document for a detailed discussion.

We think the confusion is perhaps summarised by the reviewer's statement "the authors run MARBLE independently on each reach condition". This is not the case. MARBLE is trained on all conditions *at once* contrasting LFFs drawn at random from each condition. Yet, because MARBLE only learns variation within LFFs, it shares parameters and can find similar LFFs across multiple conditions. We have now clarified this in the manuscript.

We agree with the Reviewer that MARBLE typically (but not always – see rat example in Fig. 5) receives more user input than LFADS. To make this transparent, we have now clarified the user inputs, feature extraction and unsupervised learning of MARBLE and differences against LFADS in the Discussion. However, the user input to MARBLE is fairly minimal: we only ask for multiple trials that have been collected under similar conditions, which are almost always available. MARBLE will then infer how the dynamics across conditions are related in latent space based as an emergent property of training.

Consider, for example, Linear Discriminant Analysis (LDA). During training, LDA would use only the condition labels (reach direction), but during testing, LDA simply embeds the neural state into the latent space for

classification (or whatever else the user would like to do). The knowledge of condition labels at each step strikes me as identical to MARBLE's knowledge in this example, and yet I would consider LDA to be a supervised method, while the authors call this version of MARBLE "unsupervised". On the other hand, LFADS takes no information about conditions, and therefore assumes a single dynamical system over the whole session--to me, this is far more unsupervised. It may be that the authors consider the condition-splitting akin to something like self-supervision, or minimal supervision, but I find it quite misleading to call it "unsupervised".

We think this might be a misunderstanding, as the way MARBLE uses its labels is very different from LDA.

In LDA, labels identify linear combinations of features that optimally segregate classes within a dataset. Thus, in LDA, the features are specifically tailored to a task (e.g., classification), thus introducing correlations between input (neural states) and output (latent states).

In MARBLE, on the other hand, the labels serve to introduce structural knowledge, i.e., adjacency information between some (but not all) neural states. We need this because during feature extraction, which relies on finding LFFs, we need to ensure that the trajectories are non-intersecting. This occurs when the trajectories are dynamically consistent, i.e., come from the same dynamical system under similar conditions. Therefore, LFFs typically contain no information about the condition label - they simply encode the local variation of the field, and are broadly shared across conditions. We explain this in detail in the preamble.

However, we do agree with the Reviewer that MARBLE typically (but not always – see rat example in Fig. 5) requires user input, and hence we have clarified this in several points in the manuscript (Abstract, main text and Discussion).

As I mentioned before, MINT might be a better comparison with this method of analysis, but as the authors were unable to get the package to work, I think there are a couple alternative options here, both of which would be nice to see:

Thank you for the suggestions. Because of our difficulties to use MINT we benchmarked our method against CEBRA, which to our knowledge is still the state-of-the-art method. We used CEBRA as a supervised method in both the macaque (Fig. 4) and rat (Fig. 4) examples, in order to provide the strongest possible benchmark for MARBLE.

1) The authors could add a comparison with LDA or TDR (see Sun et al. 2022 for an example use), both of which seem like they may have a similar level of supervision to MARBLE. I have no doubt that MARBLE will exceed the capabilities of LDA or TDR, since LDA and TDR are relatively simple linear models, but it's sometimes useful to use a linear baseline. Furthermore, from previous work with linear methods (see again Sun et al. 2022 for an example), I believe that LDA and TDR should uncover some clustering of the seven reach conditions with just a couple latent factors.

As requested, we have now benchmarked MARBLE against TDR, with the (x,y) coordinate of reach directions in kinematic space as regressors (Fig. 4g, Fig. S11). We were positively impressed by TDR's performance, which was comparable to LFADS in decoding reach direction. However, MARBLE significantly exceeded this performance.

We would like to remark, however, that the level of supervision in TDR and MARBLE is not the same. Whilst TDR regresses the neural states against reach directions, which is the decoding variable itself, MARBLE receives no information about the reach direction in physical space. Thus, the geometric arrangement of conditions in latent space is completely emergent. Moreover, as explained in the preamble, these conditions allow feature extraction but do not impose input-output correlation.

2) The authors could run MARBLE unsupervised and completely blind to conditions, i.e., without splitting the trials by condition and finding a single manifold over all conditions. This seems to me to provide a fairer comparison with LFADS, which doesn't know about reach direction during training. I'm interested to see how getting rid of the condition labels might change the latent representations.

Providing no condition labels would be an incorrect way to use MARBLE, as in general feature extraction can be inaccurate when locality is not well defined. This is why in all case studies we partition the datasets into dynamically consistent trajectories (or subtrajectories, Fig. 3b) and we then let MARBLE discover any relationships between them in latent space.

In other words, applying MARBLE 'blindly' would require a manifold learning step that identifies the single manifold over *all* conditions. Instead, the key insight of MARBLE (inspired by differential geometry) is that this step is not necessary, as this single manifold emerges from the learning of the local flow fields.

We have rephrased our statements throughout and, to describe better the comparison to LFADS, we mention in the Abstract, main text and Discussion that MARBLE typically requires user input, allowing the extraction of dynamical features, which are then learnt in an unsupervised way.

Minor comments

Fig 1b - I still find this flowchart confusing.

We apologise for the confusing diagram. We have now completely revised Fig. 1 and included a pictorial representation of the algorithmic steps. We hope that this new representation helps understand better the steps of the MARBLE algorithm.

What is the input to the diffusion box?

The role of the diffusion is to smooth the vector field (via vector diffusion utilising parallel transport over the manifold). Therefore, the input to the diffusion box is the whole vector field (Fig. 1b,c). Downstream the MARBLE pipeline, batches of neural states are randomly sampled from all conditions to train the algorithm.

And what is being represented by the graph with the overlaid flow field? Is this a separate input, or is this a representation of what the sample nodes are?

The graph under the flow field represents the proximity graph used for feature extraction (gradient features). This graph is constructed directly from the flow field based on the proximity of data points to other data points.

We fit one graph to neural states from a given condition. The graph injects structural knowledge, i.e., the adjacency relationship between some (but not all) neural states, and is used for feature extraction.

We have clarified this in the caption of the figure.

Fig 1d - As I said in the main comments, I greatly appreciate this addition. I have three suggestions/questions here:

Thank you for the suggestions.

1) it's hard to see what's behind the flow fields in the figure--perhaps it would help to make them more sparse?

We have made the flow fields sparser.

2) What is each point in the latent space? A time point? A single trajectory? A local flow field? It would be good to make this clear early in the manuscript.

Each point is a local flow field. We have now clarified this in the text and Fig. 1 and 2.

3) Since the agnostic version of MARBLE cares about expansive and contractive fields, it would have been nice to have a third set of flows comparing those and showing that they wouldn't overlap in the MARBLE latent representation.

We added this third set of flow fields in Fig. S6, where we show that the latent representations of expansive and contractive fields do not overlap. Thanks for this suggestion.

Fig 1g (and others with MDS representations) - Is there a reason not to show the z-space directly? Or do you have to reduce dimensionality of the MARBLE latent representation for any of it to make sense? Might be good to specify this.

We do not show the z-space of the MDS because Fig. 2e (previously Fig. 1g) indicates that the latent variable across conditions varies on a 1D manifold, which can be well-illustrated on a 2D plot.

Page 6, bottom - "This also suggests that previous methods that used CCA to detect changes across RNNs¹³ or animals^{4,48} could have detected the compound effect of changing manifold curvature and different sampling of state space, without concluding dynamical differences." This is related to the CCA comments above, but my interpretation of all of the papers cited does not match this statement. To me, these papers were all using CCA to look for similarities (and found striking similarities, in the cases of citations 4 and 48, as well as the uncited Gallego et al. 2020 paper). I will concede, however that manifold curvature may indeed cause poorer CCA alignment though, as CCA is just a linear method.

We thank you for raising this point, which we have addressed above in detail. Briefly, we claim that CCA cannot in general resolve within-manifold continuous differences in the dynamics that have otherwise the same

fixed point structure. This is because for strongly non-linear dynamics the underlying trial-to-trial variation in trajectories prevents point-to-point alignment of trajectories across conditions, which is a requirement for CCA.

Page 7, bottom - "Since a subset of neurons in PMd is directionally tuned...". The authors use directional tuning as a reason that neural representations would be sensitive to orientation on the manifold, but it seems much simpler (or at least less of a logical leap) to lean on previous findings that there's generally a geometric organization of neural states in population space that corresponds with reach direction.

Thank you for the suggestion. We have modified the corresponding sentence to: "Previous supervised approaches revealed a global geometric structure of latent states spanning different reach conditions (Churchland et al. 2012, Sun et al. 2022). We asked whether this structure could emerge unsupervised from local dynamical features in MARBLE representations."

Page 10, top - "we showed that temporal ordering naturally emerges from our similarity-preserving embeddings of local vector fields." I'm not sure I saw this explicitly mentioned before the discussion--what does this refer to? The monkey reach decoding? Rat hippocampus? Both?

Temporal ordering is encoded in the similarity between LFFs because temporally consecutive samples are adjacent over the manifold and their LFFs are typically highly similar (except at a fixed point). Correspondingly, in both the monkey and rat case studies, we discover that the latent representations preserve the arrangement of points over time.

We have clarified this in the manuscript.

Page 10, top - "This suggests a hypothesis that the neural readout into behaviour in biological brains might rely on the context of neural trajectories in a broader dynamical landscape." This statement seems vague, out of scope, and probably unnecessary. While this is an intriguing statement, I don't think anything in this manuscript really goes to support this or show what this really means.

We have rephrased this sentence "This suggests that neural flow fields in different animals can be viewed as a projection of common latent dynamics and can be reconstructed as an emergent property of the similarity-preserving embedding of local flow fields."

Extra References

Gallego, Juan A., Perich, Matthew G., Chowdhury, Raed H., Solla, Sara A., Miller, Lee E. Long-term stability of cortical population dynamics underlying consistent behavior. *Nature Neuroscience*. 2020-01. doi: 10.1038/s41593-019-0555-4

Sun, Xulu, O'Shea, Daniel J., Golub, Matthew D., Trautmann, Eric M., Vyas, Saurabh, Ryu, Stephen I., Shenoy, Krishna V. Cortical preparatory activity indexes learned motor memories. *Nature* (2022)

We have now cited these references.

Reviewer #5:

Remarks to the Author:

Gosztolai et al develop MARBLE, an unsupervised, data-driven approach for uncovering non-linear dynamics. MARBLE models non-linear dynamical systems on a manifold embedded within a high-dimensional space. It does this using geometric deep learning to create an embedding of local flow fields.

On the positive side, I found this work to be highly innovative. The model is a substantial departure from current approaches to infer dynamics from neuronal populations. Two particular innovations stood out: 1) the use of a proximity graph to approximate a manifold is to my knowledge new in the population dynamics space, and 2) the use of a graph convolutional network to uncover dynamics within the manifold, and to find an embedding set to describe those dynamics. The introduction of these methods to the set of approaches that attempt to model population dynamics is quite exciting. And the ability to compare embeddings across dynamical systems is very interesting.

Thank you for the positive summary of our work. We agree that the introduction of graph neural networks to neural data is one of the key methodological novelties in our paper.

However, we would like to add that another key theoretical insight is the possibility of representing a non-linear dynamical system by learning only the variation within local flow fields (LFFs) and how these LFFs relate to each other locally over the proximity graph. This decomposition is allowed by the manifold structure of the data, as a manifold is mathematically an atlas of local patches (i.e., LFFs), a key insight from differential geometry. Learning only the variation within LFFs forces parameter sharing, which permits identifying overlaps between LFFs across datasets. For example, in Fig. 2b MARBLE finds overlaps between the dynamical flows. In Fig. 2c-e and Fig. 3 finding these similarities and differences translates into being able to compare dynamical processes across parameter conditions and across different networks. In other words, the local learning by MARBLE circumvents assuming relationships between datasets and discover their relationship in latent space as an emergent property.

We have clarified this in the main text as we believe this concept is crucial for understanding the learning paradigm in MARBLE.

However, as presented, there were several issues that made it less compelling and at present I do not feel the work is suitable for a broad readership journal like Nature Methods. Several major issues stood out:

1. While demonstrations on synthetic datasets were interesting, real data depart from idealized and synthetic data sets in many ways, particularly in having a lot of high-dimensional variability (spiking variability) that is unrelated to variables like kinematics, etc. Thus, demonstrations on real data are critical for proving the method's utility. However, the current demonstrations on real data were not compelling and did not seem rigorous.

We agree that high-dimensional information arising from spike timing variability is an important component of neural coding, as for example was shown in the visual cortex (e.g., Stringer et al. Nature 2017). However, high-dimensional variability corresponds to a fractal-like, rather than manifold-like distribution of population

states. Specifically, the manifold assumption asserts that neural states evolve in a low-dimensional subspace of neural state space. While the manifold assumption is certainly not universal, it has been corroborated by numerous studies in motor control, cognition, navigation and other tasks (see, e.g., [4-12]).

Although we agree that studying only dynamics constrained to the low-dimensional manifold loses potentially important information about the high-dimensional variability in spiking data, our article demonstrates that the manifold assumption gives access to hitherto unexploited structure in neural data.

In this regard, we respectfully disagree and would like to argue that our demonstrations are compelling and rigorous. Let us detail why we believe our findings are of direct practical relevance to a broad readership.

- 1) (Although not a real data example), our RNN case study (Fig. 3) represents a rapidly growing research direction in neuroscience for using RNNs as surrogate models of brain activity [15,43-45]. This case study demonstrates a workflow for detecting continuous and abrupt changes in the population dynamics of RNNs, even across different RNNs.

Compelling: Our findings make MARBLE of direct practical relevance as a model-free method for the comparison of non-linear dynamical systems in neuroscience and other fields.

Rigorous: As our benchmarking illustrates (Fig. 3i,j), these results remain robust for sparsely sampled and highly non-linear dynamical flow fields arising in cognitive decision-making tasks.

- 2) Our macaque arm-reaching example demonstrates that MARBLE can discover global geometric arrangements in multi-condition motor cortical data.

Compelling: This dataset is now a well-accepted benchmark of representation-learning methods, which allowed us to demonstrate MARBLE's significantly better capacity to provide interpretable and decodable representations than competing methods (CEBRA, LFADS, etc., see Fig. 4).

Rigorous: We obtain robust representations across multiple sessions (Fig. S11). We have also performed sensitivity analysis against preprocessing hyperparameters that our results continue to hold (Fig. S13).

- 3) Our rat hippocampus example demonstrates that MARBLE embeddings are consistent across animals.

Compelling: As Reviewer 2 also noted, this example is compelling because it requires no supervision signal (behavioural data) to find consistent embeddings. Yet, in this regard, it obtains comparable results to supervised methods (Cebra-behaviour) that uses behaviour as a template to match embeddings.

Rigorous: At the request of the Reviewer, we have performed sensitivity analysis to show that our results are robust to varying preprocessing parameters (Fig. S14).

Particularly problematic was the demonstration on macaque reaching data. Buried in the methods (and highlighted by one of the reviewers) was a note that each condition in this 7-target dataset was considered a different, separately-learned manifold.

We apologise for this confusion. We aimed to be clear that MARBLE typically requires user input (but not always – see rat example in Fig. 5) to define sets of trials which are dynamically consistent, and thus lie on the same manifold. See the second paragraph: “We may equivalently treat the trajectories from a set of trials under a given experimental condition c...”. See also the caption of Fig. 4 “(using reach conditions as labels)”.

We have now further clarified this aspect in the macaque case study and elsewhere in the text.

Thus the method received some knowledge of trial identity when learning the underlying manifold structure. This undermines the claims that the current method is unsupervised.

Thank you for bringing up this subtle but important point. In the preamble of the rebuttal, we discuss in detail why we describe MARBLE as an unsupervised algorithm in the statistical learning sense.

Briefly, MARBLE requires labels as structural knowledge that defines adjacency between some neural states, namely, those that are dynamically consistent. Knowing the neighbourhood of each point allows defining LFFs and performing feature extraction using gradient filters. These spatial gradients around a neural state encode only the local variation in the flow field. Importantly, however, they do not typically carry information about which condition the LFF belongs to. An exception is when the flow fields are very different to begin with (Fig. S6), in which case it is meaningful to cluster them in latent space. Thus, the user input does not, in general, predispose the data labelled by different conditions to get disjoint (clustered) latent representations (e.g., Fig. 2b). To the contrary, MARBLE is designed to find overlaps between LFFs in latent space by embedding only local flow fields (LFFs) with the same network that shares parameters. This is exemplified by Fig. 2b (and Fig. S4c) where the latent representations of flow fields with differently assigned labels overlap in a way that reveals relationships between datasets. Note in the insets how LFFs far away can be mapped close together and vice versa.

In other words, in supervised methods, labels are used to impose a correlation between the input (neural state with a given label) and the corresponding latent space coordinate. In contrast, MARBLE learns variation with LFFs, which contain only local information and are typically broadly shared between conditions.

We have clarified this in the main text of the manuscript.

Further, because the key result of this section is the accuracy in decoding reaching movements, this result seems trivial to achieve when the underlying representations are grouped by condition - most of the decoding accuracy can be achieved just by separating conditions and decoding the trial-averaged kinematic trajectory.

We agree with the Reviewer that accurate decoding of reaching movements is one of the key results of this section. The other key result is that the geometric arrangement (ordering) between reach conditions and between timepoints emerges in latent space from learning only local information in LFFs.

However, significant decoding accuracy cannot be achieved directly decoding the firing rates. To show this, we performed three (two new) benchmarks.

1. (new) In the first benchmark, we trained a linear decoder to decode firing rates directly to kinematics without prior latent space embedding (by MARBLE or another non-linear method). We find poor performance for both position and velocity decoding, showing that firing rate trajectories cannot be accurately decoded into arm kinematics (see new Fig. 4g).
2. (new) In the second benchmark, we used Targeted Dimensionality Reduction (TDR, Mante et al. Nature 2013), which regresses the neural state trajectories to a linear subspace per timepoint corresponding to each arm reach direction. This benchmark partially confirmed the Reviewer's intuition that final reach conditions can be decoded to an accuracy comparable to LFADS, but significantly lower than MARBLE (Figs. 4g and S11). However, decoding the velocity of reaches showed poor performance, highlighting the importance of accurate dynamical representation of MARBLE.

We agree that further decoding accuracy could be achieved by more complex non-linear decoders, for example, a k-means decoder. However, in our study, the simple linear decoder serves the purpose of demonstrating that MARBLE can 'unfold' the latent dynamics governing the typically entangled non-linear dynamics in neural space.

3. To demonstrate that the decoding performance is allowed by MARBLE representations, in our third benchmark, we used CEBRA as a supervised algorithm, labelling trajectories by condition, which in CEBRA creates a grouped representation (Fig. 4e). Fig. 4g shows that while CEBRA representation provides accurate decoding of the reach directions (condition), it leads to inaccurate decoding of the reach velocities.

All of these benchmarks corroborate the fact that the decoding performance is not a trivial result of grouping conditions together. They also confirm that to achieve high decoding accuracy, the latent representation must capture information both about the reaching condition, which would allow decoding of the reach direction, and the temporal dynamics during the reach. MARBLE excels because it captures the full spatiotemporal dynamics of reaches.

We have clarified this confusion in the main text.

Similarly, the visual separation of the trajectories may be a trivial result of the different conditions being learned on separate manifolds. (Related, it is unclear from the methods how the separate manifolds are being combined?) Though I do not believe the authors are being intentionally misleading, calling the application "unsupervised" is misleading and will cause confusion.

While in the macaque arm-reaching example neural representations are indeed 'visually separated', this is not a consequence of the different conditions being learnt over separate manifolds. As we clarify above, these manifolds merely define a scaffold on which one can define a spatial gradient to describe a local flow field – they are not used by the learning algorithm. For example, in Fig. 2b, we observe that rather than separating flow fields into four clusters, MARBLE identifies the 1D manifold in latent space that parametrises the angular and radial variation in the vector fields.

In general, MARBLE is built on parameter sharing between local flow fields across conditions, and is designed to find overlaid, rather than clustered representations. For example, we show that it can detect continuous structures between datasets (Fig. 2b, 2d,e, 3f, 3i,j).

(Scientifically, the idea that different reaches are on separate manifolds and obey separate dynamics departs from current literature on manifolds underlying reaching movements, which I detail below.)

Our assumption of providing local descriptions of the data which are then combined consistently during learning does not contradict the current scientific consensus that all reaches evolve on a single global manifold. In fact, we show that condition-manifolds emerge as submanifolds of this global manifold in MARBLE latent space.

If we a priori knew this global manifold prior to applying MARBLE, this would trivialise our results because we would know that each reach condition is necessarily close in latent space to nearby conditions (e.g., that “right” is close to “down-right” or “up-right” but not “left”). This is precisely what supervised methods that used behaviour as auxiliary signal do. In contrast, in MARBLE, our only assumption is that trajectories within a given condition are likely related, which provides information about the local structure of the data. This means that the global manifold we infer - specifically the continuity of reach conditions along the manifold - is an emergent property.

We have now clarified this important point in the manuscript.

Related, there were particular choices made in the methods - heavy smoothing, PCA dimensionality choices, etc, that are presumably hand-tuned. It is unclear how robust the method is to different parameter/hyperparameter choices. For deep learning methods, demonstrating such robustness is crucial for a method to have broad applicability.

We agree that the robustness of the method to data preprocessing is essential. Our method has two preprocessing steps involving hyperparameters and the results are robust to their tuning. We have carried out extended robustness checks on them, as detailed below.

As MARBLE takes firing rates as input, we used a Gaussian convolution kernel to convert spike trains to rates. We have now performed sensitivity analysis against the length scale of the kernel. This result is shown on Fig. S13 for the macaque data and Fig. S14 for the rat data. These results show that decoding performance remains state-of-the-art for a broad range of the kernel scale.

We also performed PCA to reduce the size of the dataset prior to applying MARBLE. As MARBLE uses vector diffusion to denoise the data (Fig. 1c, see response below for more details), which requires the whole dataset to be loaded on a GPU. Lower data size yields in faster training, which is practically advantageous. We have now performed sensitivity analysis, showing that our results are robust against of the number of principal components (Fig. S13 for the macaque data and Fig. S14 for the rat data).

Further, the decoding accuracy for the macaque case study can be marginally improved compared to the values we reported by increasing the number of principal components (Fig. S13).

Prompted by the Reviewer's comment and suggestion, we now include Tables 1 and 2 to display all hyperparameters used during training. Most of these hyperparameters are standard in deep learning, while others are used to define the model architecture. We expect most of these will not need tuning for new applications as we have also kept them constant throughout this paper. However, we have now included instructions on how to tune these hyperparameters if the user does not obtain satisfactory results with the default parameters. Note that this is standard practice in deep learning models, including CEBRA and LFADS.

2. A higher level issue is the repeated claim in this manuscript that the method leads to interpretable representations, without any definition of what interpretability is. It is hard to evaluate any claim of a method being "more interpretable" without knowing what is meant by interpretability. Thus the notes e.g. that one method or the other "lacked interpretability" are not possible to substantiate. In some cases the authors seem to be conflating interpretability with decodability, but for neuroscientific applications, it is unclear why representations that lead to better decoding should be more "interpretable"- for example, because variability in neural population activity does not simply reflect representation of external variables. To even evaluate a claim of interpretability, the authors should be very clear from the outset what they mean.

This is an important question and we thank the reviewer for bringing it up. In our paper, we define interpretability as the property of latent representations of neural states and dynamics to unveil latent variables that explain the variability in the data. More specifically, we claim that the latent representations produced by a method are interpretable when:

1. They visually reflected the task's structure. In the case of macaque reaching, this corresponds to both the spatial ordering of reach conditions and the preservation of the temporal ordering of latent vectors.
2. They can be linearly decoded with high accuracy. This test reflects that the latent representations simplified the dynamics to low-dimensional latent variables, which can be efficiently used for downstream tasks. Note that a non-linear decoder would not serve this purpose because we would not know if accurate decoding resulted from the fidelity of the representations or the decoder itself.
3. They reveal the variation of latent states across conditions. In experiments where only a few parameters are being manipulated (e.g., the direction of a reach or the change of a gain), we expect that comparing the latent embeddings of systems across conditions reveals a low-dimensional structure that parametrises the change of conditions.

We have clarified these points in the main text.

3. I strongly agree with the previous reviewer that the manuscript as written – even with the inclusion of toy examples – is extremely difficult to follow for readers who are not experts in geometric deep learning. Even after reviewing the methods in detail, and having some familiarity with graph neural networks, it was still difficult to understand exactly what the authors were doing. To have this manuscript published in a broad readership journal like Nature Methods, I would strongly recommend that the manuscript be written in a way that breaks down the technical aspects of the method in a much more approachable fashion. Clearly outlining the different steps of the method and example outputs of different stages on real data would be critical. Particularly: 1) more clearly demonstrating how the nearest neighbors algorithm is used to create a manifold, 2) explicitly delineating

how tangent spaces and moving between them works, 3) explaining how gradient filters are used and example outputs, 4) explaining how the method is sensitive to real world noise, and hyperparameter choices. Perhaps the authors might seek more detailed feedback from readers without geometric deep learning expertise to make the manuscript more approachable.

We greatly appreciate both Reviewers' feedback, which helped us improve the clarity of our manuscript, including new figures and explanations We acknowledge that MARBLE provides a novel viewpoint on the representation learning of neural data, drawing on fields such as graph neural networks and differential geometry that are currently not in neuroscientists' standard toolbox.

We followed the suggestions and split Figure 1 into two figures. Figure 1 now expands upon the steps in the MARBLE pipeline, including

1. Manifold learning - How the nearest neighbour graph approximates the manifold
2. Trainable vector field smoothing - Obtaining a smoothed vector field that preserves the fixed point structure of the vector field. This operation has a trainable parameter that automatically tunes the balance between smoothness and feature expressivity.
3. Feature extraction and unsupervised representation learning - We depict how nearest neighbour graphs are used to inject structural knowledge, i.e., locality (and not labels), and how training proceeds by contrasting local flow fields defined by all graphs.

Figure 2 now provides more intuition about the MARBLE latent representations and their relationship with local flow fields.

We also extended Figure S1 with new parts d and e to illustrate alignment of tangent spaces and computation of gradient filters. We included this in the SI since gradient filters are just one particular choice and are not instrumental for understanding the MARBLE algorithm or how it is used.

We have also rewritten the text to match the improved depiction of the workflow.

4. For a method that centers on modeling and extraction of latent dynamics – and may be quite innovative in this regard – noticeably little effort was spent trying to characterize or interpret the dynamics of real neural data (i.e., the flow fields themselves). There was little evidence presented that MARBLE is successfully modeling the dynamics of real neural data. E.g., applications to datasets with many conditions, and showing generalization of the learned model to unseen conditions, would be a powerful demonstration of the model's ability to learn dynamics on real data.

This is a very interesting application. As the Reviewer suggests, the techniques that MARBLE introduces will allow addressing several important problems, such as the ability to decode from unseen neural conditions based on the assumption that these conditions relate to those in the training set based on some underlying global latent geometry. We also believe that MARBLE will allow studying the fine-grained changes in dynamics during learning, which purely geometric methods such as CCA are not expressive enough to capture. We are currently working to explore these directions. However, in order to keep the focus of the present article, we leave this for future work.

For this manuscript, we have now provided insets in Fig. 2b to illustrate the variation of the fields along latent manifolds.

We have now mentioned some of these directions in the Discussion.

5. It appears that the manifold learning step does not have any sort of denoising process.

We believe this could be a misunderstanding. The first step in our pipeline is a learnable vector diffusion operation (Fig. 1c) that denoises the vector field globally while preserving the fixed point structure. This vector diffusion is different from scalar (heat) diffusion in that it does not act channel-wise on the vectors but utilises parallel transport over (the tangent bundle of) the manifold to align the flow fields between neighbouring trajectories. It is not hand-tuned but uses a learnable parameter which ensures that an optional balance is reached between smoothing and the expressivity of local flow fields.

Thus for real data applications which contain noise, the authors do a lot of (hand-tuned?) preprocessing of the data. This is in stark contrast to e.g. the LFADS point of comparison, which operates on the raw binned spike counts.

We agree that spikes can encode important information. However, smooth firing rates often provide a great deal of insight, as our case studies corroborate. Thus, working with continuous variables is a design choice that we made in order to be able to use the additional spatial inductive bias of the data in the relative spatial position of ensembles of nearby neural trajectories. This would not be possible by learning only the temporal structure in single trajectories, as done in LFADS. This explains MARBLE's higher interpretability and decodability than LFADS (Fig. 4e-g). In addition, working with continuous variables (such as firing rates) makes MARBLE applicable to a broad range of dynamical systems, even outside neuroscience.

In sum, we agree that the high-frequency structure of neural dynamics are important in some applications and is one that LFADS is possibly more adept at capturing. However, the strong emphasis on the 'manifold assumption' in current neuroscience literature (e.g., Refs. [4-12]) provides substantial evidence that low spatial-frequency, spatially-structured dynamics are prevalent in neural systems, which MARBLE is specifically designed to capture.

We have clarified these technical points in the main text and the distinction with LFADS in the Discussion.

Because noise in real neural data is a huge issue (and thus many methods resort to averaging across conditions or using a supervisory signal), it would be critical for the authors to explore how sensitive MARBLE is to different choices of preprocessing when estimating the manifold. It seems like the current approach, with heavy smoothing, might destroy any high-frequency structure in the data.

We agree with the reviewer that robustness to preprocessing hyperparameters is essential. As we detailed in the response above, we have performed robustness analysis for both the macaque and rat examples against the number of PCA dimensions and Gaussian smoothing kernel scale. The results are shown on Fig. S13 for the macaque data and Fig. S14 for the rat data. These results show that decoding performance remains state-of-the-art when changing the number of principal component and the kernel scale within a broad range.

Further, we also find that the decoding accuracy for the macaque case study can be marginally improved compared to the values we reported by increasing the number of principal components (Fig. S13), so our reported results are in fact conservative.

Medium:

- For real data, the model chose as one point of comparison the multi-session macaque reaching dataset from the original LFADS paper. In the LFADS paper, the point of this dataset was that each session had very few channels, and thus it could highlight the ability to “stitch” together several sessions of separately-recorded neural data. Since the authors are not using it for that purpose, there are several other reaching datasets from the Shenoy lab [1,2] that would be much better for demonstrating manifold and dynamics discovery. If the authors are specifically trying to make a point about the effectiveness of MARBLE for limited-channel-count data, that should be explained more thoroughly - otherwise the current dataset is of questionable relevance for the author’s demonstration.

We do not limit ourselves to low channel count data. For example, while the macaque dataset had 24 channels, the rat dataset contained 48-120 channels. Rather, we assume that the population dynamics evolve on a low-dimensional manifold.

Each case study we provide has been carefully chosen to illustrate a specific advantage point of MARBLE;

- RNN example: illustrates MARBLE’s ability to reveal discrete and continuous low-dimensional variation across conditions (Figs. 3f-j)
- Macaque example: illustrates MARBLE’s ability to find a geometric representation of reaches in macaque neural recordings and state-of-the-art decoding thereof using a simple decoder
- Rat example: illustrates that MARBLE embeddings are consistent across animals.

While demonstrations on additional datasets are certainly possible, and we have already done so for independent manuscripts in preparation, for this manuscript, we have selected datasets that have been previously used by existing works, namely, LFADS and CEBRA, to enable reproducible benchmarking and transparent comparison.

- Kudos to the authors for spending a lot of effort on the codebase, with examples and detailed installation instructions for multiple platforms. Similar to another reviewer, I had trouble installing the code on my M2 MacBook Pro. I did have success on a Mac Studio and on Linux. Similar to the other reviewer, I found some notebooks difficult to run because they had dependencies outside of those in the installation instructions.

We thank the reviewer for thoroughly testing our codebase. We have now fixed the issue with the installation on M1/M2/M3 Macs, which we think was caused by the Pytorch Geometric library, whose developers recently changed the installation protocol.

Please note that the data processing relies on libraries that are not part of the MARBLE package but, indeed, they should automatically install when running the individual notebooks. We have now run all of the code again to confirm that all examples download and run automatically.

We are striving to maintain the MARBLE code in the foreseeable future and we will efficiently deal with any issues the users might encounter.

- In a response to another reviewer regarding the issue of learning separate manifolds on the macaque reaching dataset, the authors say:

“We agree with the reviewer that MARBLE considers the trials under a given task condition to belong to an independent manifold. This is necessary because only under a given condition do we have good reason to believe that the dynamics are consistent.”

Perhaps there is a misunderstanding of the literature. For example, in the Churchland 2012 paper [3], the key point of the manuscript was that a consistent set of dynamics (rotational) describes a wide variety of conditions (100+ reaching conditions in some datasets), and all these conditions were present on the same manifold (estimated using smoothing and PCA). Similarly, in the LFADS paper [4] that the authors compare to, the method was used to learn a single manifold and set of dynamics across all conditions.

We thank the Reviewer for pointing out this important issue, which was not clearly articulated in our article. As we described above, MARBLE requires local structural knowledge of the data. We took individual reach directions and the smallest dynamical unit that belongs to the same manifold. This does not contradict the possibility that all these manifolds are submanifolds of a common manifold, as Churchland 2012 and the LFADS paper showed. However, our aim was to show that this manifold emerges from solely local information.

We have clarified this point in the manuscript.

- In the methods there was a note about the optimal transport computation comparing conditions. Could this be expanded a bit? It was unclear whether this was still unsupervised or not.

We use optimal transport to compare MARBLE latent representations across conditions (Figs. 2d, 3f, S11b) and networks (Fig. 3i). One of the claims of interpretability (see above) stems from the idea that MARBLE representations reveal low-dimensional latent variables. This means that across conditions, distributional changes of MARBLE embeddings should reflect variation in these variables.

We chose the optimal transport distance to compute this distributional distance because it accounts for both density differences and the metric structure of the latent space.

As this is a postprocessing step, this does not introduce supervision into the learning.

We have clarified this in the manuscript.

- Other methods often have a recognition network that would allow inference to be performed on held out or unseen data. How is that done in the current case? How is cross-validation performed? Specifics and limitations in this regard should be clearly spelled out in the manuscript.

MARBLE is an inductive algorithm, meaning that it learns to embed LFFs point-wise and, at test time, it can be evaluated on LFFs around unseen (and unlabeled) points. Thus, network training is successful when the

multilayer perceptron generalises to unseen LFFs. We test this by splitting the neural states into training/validation/test sets (80/10/10 %) and testing if the loss on held-out data (validation and test sets) is comparable to the loss on the training set. The loss is an unsupervised loss, which tests whether, in the validation and test sets, neighbouring LFFs are as close relative to non-neighbouring ones in the given subset of data as it would be expected from the training dataset.

We have clarified this in the manuscript.

- It appears that the current formulation of the model relies on completely autonomous dynamics - i.e., all the temporal evolution of the data must be completely described by the learned vector field. While this may work well for carefully aligned data, it is unclear how this accommodates data with non-autonomous dynamics, i.e. where unexpected events occur. The authors should include some discussion of this issue.

This is correct. We rely on “dynamical consistency” within a condition, which means that the inputs are the same for the respective trials. This means that inputs, which can be time-varying along the trial, can be absorbed and modelled as augmented variables of an autonomous dynamical system. Across conditions, we make no assumptions regarding the variation of inputs.

This shows that one way to use MARBLE is to partition the dataset into subsets where one can assume “dynamical consistency” across trials. For example, in the RNN case study, we split the dataset into stimulated and unstimulated subsets (Fig. 3b, S8b,c), in the macaque case study, we split the dataset into reach conditions (Fig. 4c), whereas in the rat hippocampus dataset, we did not perform any splitting.

We clarified this in the manuscript and Discussion.

- In the multi-session macaque reaching dataset, presumably the dynamics across all sessions should be the same. Demonstrating that MARBLE uncovers consistent dynamics (embeddings) across sessions would be compelling. It was unclear if/how this is shown in Fig. S9. (Also, this analysis would need to be done after addressing the fact that MARBLE’s application to this dataset is not truly unsupervised, though that is what is claimed in the manuscript.)

[1] <https://dandiarchive.org/dandiset/000070>

[2] <https://npsl.stanford.edu/research/data-code>

[3] <https://pubmed.ncbi.nlm.nih.gov/22722855/>

[4] <https://pubmed.ncbi.nlm.nih.gov/30224673/>

In Fig. S9 (S11 in the revised manuscript), we assess the across-session consistency of embeddings in three ways: (1) Fig. S11a visually reveals similar geometric arrangements up to a rotational degree of freedom, (2) Fig. S11b quantifies this similarity by the diagonal structure of the condition-to-condition distance matrix, whose rows are ordered by the angular order of reaches (Fig. S11b) and (3) Fig. S11c further visually corroborates the emergent angular order of reaching conditions in the latent representations that is consistent across sessions.

Minor notes:

- Fig. 2b refers to “stimulation” patterns. In neuroscience “stimulation” often refers to e.g. electrical or optogenetic perturbation of a circuit. Consider using “stimulus” instead (paralleling Fig. 2a).

We have now changed “stimulation” to “stimulus” in Fig. 2b. Thanks.

Reviewer #2:

Remarks to the Author:

Gosztolai et al. present in this manuscript MARBLE, a novel computational method to analyze the low-dimensional, nonlinear dynamics of neural population activity. In this revision, the authors have added additional analyses to improve the rigor of the work, and they have added text to clarify misunderstandings from previous review cycles.

Overall, I am happy with this revision. The authors have addressed most, if not all of my concerns with their edits. Chiefly, the new CCA results make sense to me, and I find it intriguing that while CCA shows that dynamically similar RNN solutions are more alignable than dynamically different ones, they don't show the nuances that MARBLE does. This strikes me as an important result, and as impressed as I was with the MARBLE results before, the context brought by the CCA results give it some more impact.

We thank the Reviewer for his previous remarks and his substantial effort into making suggestions for improvements, all of which have significantly increased the rigour and clarity of our study.

Also, I think I now understand the authors' reasoning for calling this method "unsupervised". I agree with the authors that MARBLE is far less supervised than the methods that I mentioned in my previous review, and the authors agree with me that MARBLE can sometimes require more user input than a fully unsupervised algorithm, like LFADS. The paragraph in the discussion goes a long way towards making this clear, and I am happy with the change.

However, as I read the manuscript, I remained skeptical about the "unsupervised" claim. I'm not sure if this is from previous bias or from the idea not fully coming across until the discussion. For example, on the bottom of page 8, the authors wrote: "We asked whether this structure could emerge unsupervised from local dynamical features in MARBLE representations. We constructed a separate manifold from the firing rates for each reach condition where we expected the dynamics to be consistent..." This pair of sentences, back-to-back struck me as incredibly strange--the authors mentioned a feature that might "emerge unsupervised" but then mention constructing separate manifolds for each reach condition. While I now understand the reasoning, this seemingly inconsistent logical turnaround left a bad taste.

We agree with the Reviewer that the term 'unsupervised' could cause confusion in this context. We have rephrased the sentence to clarify that the geometric configurations in MARBLE embeddings is an emergent property of the dynamics and not the user input.

However, I wonder if the authors would agree that the level of supervision MARBLE uses is similar to that of running PCA on neural trajectories averaged over each condition. If the only thing MARBLE uses the condition-splitting for is to estimate local flow fields to eventually embed together, then it seems to me that this is really just a trial aggregation pre-processing step, like trial averaging, but much more sophisticated. If the authors agree with this, then I think this provides a concrete comparison for readers familiar with more well-known methods--trial-averaged PCA is reasonably standard for neural state analysis. I think most people would not call trial-averaged PCA completely unsupervised, but they would not call it "supervised" either, and it may be a good way to situate MARBLE for the monkey dataset use case. If the authors can use this comparison, or one like it, this may situate the "unsupervisedness" of MARBLE more clearly for readers.

We predominantly agree with this description. MARBLE first requires aggregating ensembles of trials that allow feature extraction and then learning to embed these features via an unsupervised algorithm. This is conceptually similar to the frequent approach of applying PCA to condition-averaged (i.e., trial-averaging per condition) trajectories.

However, it is very important to stress that MARBLE does not average across trials for a given condition. In fact, in a follow-up work with the Jazayeri lab, we show that in decision-making problems PCA on condition-averaged data can blur together different conditions, which can be falsely interpreted as a delayed decision. By contrast, MARBLE can

assign trials to their corresponding decision-manifolds, which does not necessarily correspond to the ones initially defined by the user.

More generally, as we described in the previous rebuttal, this parallel with condition averaging only makes sense when the neural activities evolve closely for different trials. We expect that as neuroscience tasks become progressively more naturalistic and complex, we will increasingly encounter situations where activity will follow complex dynamical patterns where trial averaging loses its meaning altogether.

We have now added a remark to the Discussion to better situate MARBLE relative to existing methods such as trial-averaged PCA.

Raeed Chowdhury

Minor comments

Thank you for the comments below. We have corrected the remaining typos.

Page 3, bottom: it's not stated anywhere what p is here--I believe it's the number of gradients taken for the flow field estimation, but I can't be sure (I may have missed it in the text, but if not, it should be clarified somewhere in the main body)

The integer p is the order of the function that locally approximates the vector field. It is also equal to the size of the local flow field, which is defined as the neighbourhood at most p hops away from a given vector. We have clarified this now in the main text.

Page 4, top: "invariant different" I think there's a typo here

It should read *invariant of* different. We have now fixed this typo.

Page 4, bottom: Figure 2h-j does not exist. Also I couldn't quite tell what examples the authors refer to when they say "In both examples, note that...".

This should refer to Fig 3h-j. In the following sentence, we meant to say "In both embedding-aware and -agnostic examples, note that". We corrected these confusing references.

Page 6, figure 3b: typo in the figure ("vin")

Corrected.

Page 7, top: It's unclear to me how figure 3e shows the different fixed point landscapes of the two RNN solutions. Maybe a typo?

Fig 3e shows different fixed point landscapes for one RNN solution. We have changed the sentence to "These two solutions exhibit qualitatively different fixed point landscapes (three fixed points for zero gain and one limit cycle for large enough gain), which cannot be aligned via continuous (linear or non-linear) transformations."

Page 7, bottom: I suggest removing the words "best-fit" from the description of CCA, since it could be confusing exactly what this means.

Removed.

Page 18, S1.8 heading: change from "geometry invariance" to "embedding invariance"

Corrected.

Fig S1 and S6: captions missing for some panels. Overall, double check the figures, captions, and references to make sure they're correct and all there.

We have revised each figure caption carefully.